# Agonist-mediated switching of ion selectivity in TPC2 differentially promotes lysosomal function

Susanne Gerndt[1,2†], Cheng-Chang Chen[1†], Yu-Kai Chao[2†], Yu Yuan[3†], Sandra Burgstaller[4], Anna Scotto Rosato[2], Einar Krogsaeter[2], Nicole Urban[5], Katharina Jacob[2], Ong Nam Phuong Nguyen[1], Meghan T Miller[1,6], Marco Keller[1], Angelika M Vollmar[1], Thomas Gudermann[2], Susanna Zierler[2], Johann Schredelseker[1,2], Michael Schaefer[1,5], Martin Biel[1], Roland Malli[4], Christian Wahl-Schott[7]*, Franz Bracher[1]*, Sandip Patel[3]*, Christian Grimm[2]*

[1]Department of Pharmacy – Center for Drug Research, Ludwig-Maximilians-Universität, Munich, Germany; [2]Walther Straub Institute of Pharmacology and Toxicology, Faculty of Medicine, Ludwig-Maximilians-Universität, Munich, Germany; [3]Department of Cell and Developmental Biology, University College London, London, United Kingdom; [4]Molecular Biology and Biochemistry, Gottfried Schatz Research Center, Medical University of Graz, Graz, Austria; [5]Rudolf-Boehm-Institute for Pharmacology and Toxicology, Universität Leipzig, Leipzig, Germany; [6]Pharma Research and Early Development (pRED), Roche Innovation Center Basel, F. Hoffmann-La Roche, Basel, Switzerland; [7]Institute for Neurophysiology, Hannover Medical School, Hannover, Germany

*For correspondence:
wahl-schott.christian@mh-hannover.de (CW-S);
franz.bracher@cup.uni-muenchen.de (FB);
patel.s@ucl.ac.uk (SP);
Christian.Grimm@med.uni-muenchen.de (CG)

†These authors contributed equally to this work

Competing interests: The authors declare that no competing interests exist.

**Abstract** Ion selectivity is a defining feature of a given ion channel and is considered immutable. Here we show that ion selectivity of the lysosomal ion channel TPC2, which is hotly debated (Calcraft et al., 2009; Guo et al., 2017; Jha et al., 2014; Ruas et al., 2015; Wang et al., 2012), depends on the activating ligand. A high-throughput screen identified two structurally distinct TPC2 agonists. One of these evoked robust $Ca^{2+}$-signals and non-selective cation currents, the other weaker $Ca^{2+}$-signals and $Na^+$-selective currents. These properties were mirrored by the $Ca^{2+}$-mobilizing messenger, NAADP and the phosphoinositide, $PI(3,5)P_2$, respectively. Agonist action was differentially inhibited by mutation of a single TPC2 residue and coupled to opposing changes in lysosomal pH and exocytosis. Our findings resolve conflicting reports on the permeability and gating properties of TPC2 and they establish a new paradigm whereby a single ion channel mediates distinct, functionally-relevant ionic signatures on demand.

## Introduction

When ion channels open in response to a given stimulus, they allow the flow of a specific set of ions. The textbook view is that ion selectivity for a given ion channel is a hallmark feature that cannot be changed. But evidence has accrued suggesting that this may not always be the case. P2X receptors have long been proposed to change their ion selectivity in real time upon prolonged activation (*Khakh and Lester, 1999*) albeit controversially (*Li et al., 2015*). And in TRPV channels, the selectivity filter is thought to form a second gate thereby coupling channel opening with changes in ion selectivity (*Cao et al., 2013*). Other channels such as Orai channels (*McNally et al., 2012*) and the mitochondrial uniporter MCU1 (*Kamer et al., 2018*) appear to alter their ion selectivity depending on protein partners.

Two-pore channels (TPC1-3) are ancient members of the voltage-gated ion channel superfamily (*Patel, 2015*). TPCs are expressed throughout the endo-lysosomal system where they regulate the trafficking of various cargoes (*Grimm et al., 2014*; *Ruas et al., 2010*; *Sakurai et al., 2015*). Loss- and gain-of function of TPCs is implicated in a number of diseases including NAFLD (non-alcoholic fatty liver disease) and Parkinson's (*Patel, 2015*; *Patel and Kilpatrick, 2018*). Despite their patho-physiological importance, both the ion selectivity and activating ligand(s) of TPCs are equivocal. Initial studies characterized TPCs as non-selective $Ca^{2+}$-permeable channels activated by NAADP (nicotinic acid adenine dinucleotide phosphate) (*Brailoiu et al., 2009*; *Brailoiu et al., 2010*; *Calcraft et al., 2009*; *Grimm et al., 2014*; *Pitt et al., 2010*; *Ruas et al., 2015*; *Schieder et al., 2010*). But other studies indicate that TPCs are highly-selective $Na^+$ channels that similar to endo-lysosomal TRP mucolipins (TRPMLs), are activated directly by $PI(3,5)P_2$ (phosphatidylinositol 3,5-bisphosphate), and/or by voltage (*Boccaccio et al., 2014*; *Cang et al., 2014*; *Guo et al., 2017*; *She et al., 2018*; *Wang et al., 2012*). Co-regulation of TPCs by these disparate stimuli has been noted in some instances (*Jha et al., 2014*; *Ogunbayo et al., 2018*; *Pitt et al., 2014*; *Rybalchenko et al., 2012*).

Functional characterization of TPCs is challenging due to their intracellular location. The anionic nature of NAADP and $PI(3,5)P_2$ prevents sufficient plasma membrane permeability, making them less suitable for experimental use in intact cells. On the other hand, studies of isolated lysosomes may result in loss of crucial accessory factors such as NAADP-binding proteins (*Lin-Moshier et al., 2012*). To circumvent these issues, we screened for membrane-permeable small molecule activators of TPCs. Here, we report two agonists that surprisingly switch the ion selectivity of TPC2 with different consequences for lysosomal function and integrity.

## Results

### Identification of novel small molecule agonists of TPC2

Mutation of the N-terminal endo-lysosomal targeting motif in human TPC2 redirects it to the plasma membrane where TPC2 reportedly mediates robust $Ca^{2+}$ entry upon activation with NAADP (*Brailoiu et al., 2010*). We took advantage of this property to screen a cell line stably expressing TPC2$^{L11A/L12A}$ with a library of 80000 natural and synthetic small molecules using a FLIPR-based $Ca^{2+}$ assay (*Figure 1A and B*). Two structurally independent hits were identified, termed TPC2-A1-N and TPC2-A1-P (*Figure 1C and D*), which reproducibly evoked $Ca^{2+}$ signals. The signals evoked by the compounds showed different kinetics whereby the TPC2-A1-N response reached its plateau faster than TPC2-A1-P (*Figure 1E and F*). The structures and activities of these hits were confirmed by independent chemical syntheses and subsequent retesting. Full concentration-effect relationships for the plateau response indicated $EC_{50}$ values of 7.8 μM and 10.5 μM, for TPC2-A1-N and TPC2-A1-P, respectively (*Figure 1G and H*). We thus identified novel small molecules that activate TPC2 and confirm its $Ca^{2+}$-permeability in intact cells.

### TPC2 agonists differentially evoke cytosolic $Ca^{2+}$ signals in live cells

To validate the hits further, we performed ratiometric $Ca^{2+}$ imaging of single cells transiently transfected with TPC2$^{L11A/L12A}$ or a 'pore-dead' version (TPC2$^{L11A/L12A/L265P}$). As shown in *Figure 1I and J*, TPC2-A1-N evoked $Ca^{2+}$ signals in cells expressing TPC2$^{L11A/L12A}$ but not TPC2$^{L11A/L12A/L265P}$. These data confirm that TPC2-A1-N evoked $Ca^{2+}$ influx through the TPC2 pore. In accord, the responses to TPC2-A1-N were selectively blocked by the recently identified TPC2 blockers tetrandrine (Tet), raloxifene (Ral), and fluphenazine (Flu) (*Penny et al., 2019*; *Sakurai et al., 2015*; *Figure 1—figure supplement 1*) and by removal of extracellular $Ca^{2+}$ (*Figure 1—figure supplement 2*). TPC2-A1-P also induced $Ca^{2+}$ signals in cells expressing TPC2 in the presence (*Figure 1K and L*) but not absence (*Figure 1—figure supplement 2*) of extracellular $Ca^{2+}$. However, the responses were smaller and delayed compared to TPC2-A1-N (*Figure 1I and K*), consistent with the results obtained in cells stably expressing TPC2$^{L11A/L12A}$ (*Figure 1E and F*). TPC2-A1-P failed to induce $Ca^{2+}$ signals in cells expressing 'pore-dead' TPC2$^{L11A/L12A/L265P}$ (*Figure 1K and L*). Both TPC2-A1-N and TPC2-A1-P also failed to evoke $Ca^{2+}$ signals in cells expressing human TRPML1 re-routed to the plasma membrane (TRPML1$^{\Delta NC}$) (*Grimm et al., 2010*; *Yamaguchi et al., 2011*; *Figure 1M–N* and *Figure 1—figure supplement 1A–B*). Similar negative results were obtained with the agonists in

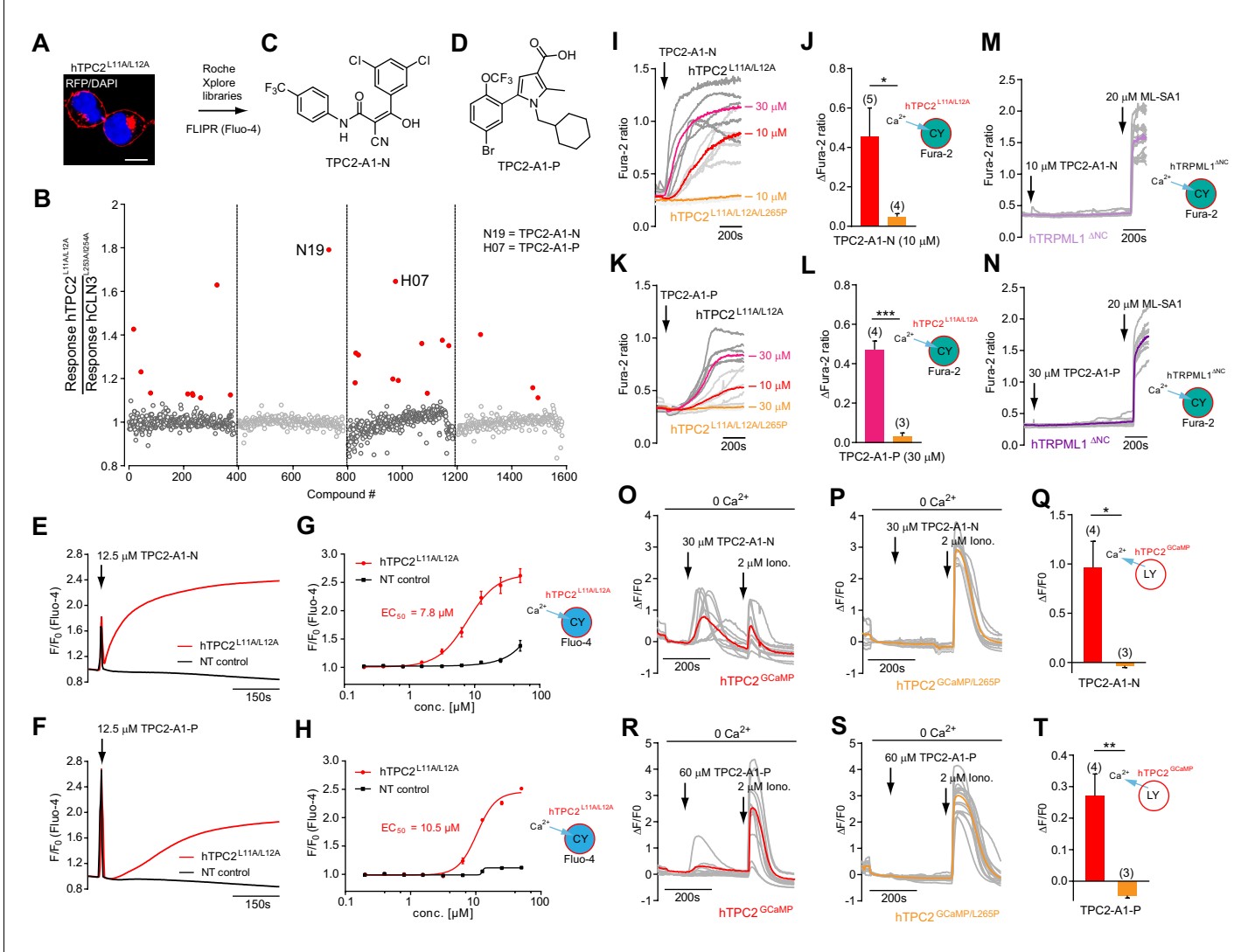

**Figure 1.** Identification of novel small molecule agonists of TPC2. (A) The chemical screening was performed using cells stably expressing human TPC2 re-routed to the plasma membrane (hTPC2$^{L11A/L12A}$) and loaded with the fluorescent Ca$^{2+}$ indicator, Fluo-4 (CY = cytosol). For counter-screening, a stable cell line expressing cell surface human CLN3 (hCLN3$^{L253A/I254A}$) was used. (B) A library comprising 80.000 compounds was screened. Shown are the primary screen results of four plates (with 384 compounds, each) where the response to a given compound in cells expressing hTPC2$^{L11A/L12A}$ is expressed relative to that in cells expressing hCLN3$^{L253A/I254A}$. Primary hits are shown as red spots. Subsequent confirmatory and follow-up assays led to the identification of the two hit compounds, N19 (TPC2-A1-N) and H07 (TPC2-A1-P). (C–D) Figures of TPC2-A1-N and TPC2-A1-P. (E–F) Representative FLIPR-generated Ca$^{2+}$ signals (Fluo-4) in TPC2$^{L11A/L12A}$–expressing cells (red lines) or non-transfected (NT) control cells (black) after addition of TPC2-A1-N (E) or TPC2-A1-P (F). (G–H) Concentration-effect relationships for Ca$^{2+}$ increases (Fluo-4) in response to different concentrations of TPC2-A1-N (G) and TPC2-A1-P (H). Each concentration was tested in duplicates in two to four independent experiments, each. (I) Representative Ca$^{2+}$ signals recorded from HeLa cells loaded with the ratiometric Ca$^{2+}$ indicator, Fura-2 and stimulated with the indicated concentration of TPC2-A1-N. Cells were transiently transfected with plasma-membrane-targeted human TPC2 (hTPC2$^{L11A/L12A}$) or a pore dead mutant (TPC2$^{L11A/L12A/L265P}$). Colored lines represent the mean response from a population of cells. Traces in grey represent responses of single cells. (J) Statistical analysis of the maximal change in Fura-2 ratio (mean ± SEM) with the number of independent transfections in parentheses. An unpaired t-test was applied. *p<0.05. (K–L) Similar to I and J except that cells were stimulated with TPC2-A1-P. ***p<0.001. Traces in grey represent single cells responding to TPC2-A1-P. (M–N) Representative Ca$^{2+}$ signals recorded from HeLa cells transiently transfected with human TRPML1$^{ΔNC}$. Cells were sequentially stimulated with TPC2-A1-N (M) or TPC2-A1-P (N) and the TRPML agonist ML-SA1. (O–T) Ca$^{2+}$ signals recorded from HeLa cells transiently transfected with GCaMP6(s) fused to human TPC2 (hTPC2$^{GCaMP}$) or a pore-dead mutant (hTPC2$^{GCaMP/L265P}$). Cells were sequentially stimulated with TPC2-A1-N (O and P) or TPC2-A1-P (R and S) and the Ca$^{2+}$ ionophore, ionomycin in the absence of extracellular Ca$^{2+}$. Statistical analysis of the change in fluorescence intensity (mean ± SEM; Q and T). An unpaired t-test was applied. *p<0.05 and **p<0.01. CY = cytosol, LY = lysosome.

The online version of this article includes the following source data and figure supplement(s) for figure 1:

**Source data 1.** Identification of TPC2 hit compounds.

*Figure 1 continued on next page*

**Figure supplement 1.** Effect of different TPC2 channel blockers on TPC2-A1-N and ML-SA1 mediated changes in intracellular calcium.
**Figure supplement 1—source data 1.** Effect of different TPC2 channel blockers on TPC2-A1-N and ML-SA1 mediated changes in intracellular calcium.
**Figure supplement 2.** TPC2-A1-N and TPC2-A1-P mediated changes in intracellular calcium in the absence of extracellular calcium.
**Figure supplement 2—source data 1.** TPC2-A1-N and TPC2-A1-P mediated changes in intracellular calcium in the absence of extracellular calcium.
**Figure supplement 3.** Effects of TPC2-A1-N and TPC2-A1-P on TRPML1, 2, and 3.
**Figure supplement 3—source data 1.** Effects of TPC2-A1-N and TPC2-A1-P on TRPML1, 2, and 3.
**Figure supplement 4.** Synthesis and functional analysis of TPC2-A1-N analogs.
**Figure supplement 4—source data 1.** Functional analysis of TPC2-A1-N analogs.
**Figure supplement 5.** Structures of TPC2-A1-N and analogs.
**Figure supplement 6.** Synthesis and functional analysis of TPC2-A1-P analogs.
**Figure supplement 6—source data 1.** Functional analysis of TPC2-A1-P analogs.
**Figure supplement 7.** Effect of TPC2-A1-N and TPC2-A1-P on TPC1.
**Figure supplement 7—source data 1.** Effects of TPC2-A1-N and TPC2-A1-P on TPC1 in endo-lysosomal patch-clamp experiments.

cells expressing TRPML2 or TRPML3 (*Figure 1—figure supplement 3C and D*), indicating a selective, pore-dependent action of both agonists on TPC2.

The consensus logP (octanol/water; SwissADME) values for TPC2-A1-N (4.56) and TPC2-A1-P (5.35) predict that they are cell permeable. To determine whether TPC2-A1-N and TPC2-A1-P were able to release $Ca^{2+}$ from lysosomes, we expressed TPC2 fused to the genetically encoded $Ca^{2+}$ indicator, GCaMP6(s) both with (TPC2$^{GCaMP}$) and without (TPC2$^{GCaMP/L265P}$) an intact pore (*Figure 1O–T*) in order to measure global $Ca^{2+}$ signals. TPC2-A1-N and TPC2-A1-P evoked $Ca^{2+}$ signals in cells expressing intracellular TPC2$^{GCaMP}$ but not TPC2$^{GCaMP/L265P}$. Again the responses to TPC2-A1-P were modest compared to TPC2-A1-N.

Systematic structure modifications were performed on both screening hits. Surprisingly, none of the modified versions of TPC-A1-N or TPC2-A1-P showed significantly increased efficacies (*Figure 1—figure supplements 4* and *5*). TPC2-A1-P analogues missing the trifluoromethoxy residue were completely inactive (*Figure 1—figure supplement 6*).

Collectively, our data show that TPC2-A1-N and TPC2-A1-P activate TPC2 with differential effects on $Ca^{2+}$ mobilization.

## TPC2 agonists differentially evoke Na$^+$ currents in isolated endo-lysosomes

To determine more directly whether TPC2-A1-N and TPC2-A1-P activate TPC2, endo-lysosomal patch-clamp experiments were performed (*Figure 2*). TPC2-A1-N elicited currents using Na$^+$ as the major permeant ion, in vacuolin-enlarged endo-lysosomes isolated from TPC2-expressing cells (*Figure 2A*). The currents were inhibited by ATP as reported previously (*Cang et al., 2013*). In contrast, no activation was found in cells expressing TPC1 (*Figure 1—figure supplement 7*). Endo-lysosomes isolated from cells expressing a gain-of-function variant of TPC2 (TPC2$^{M484L}$) (*Chao et al., 2017*) showed larger currents compared to the wild-type isoform upon application of TPC2-A1-N (*Figure 2B*). TPC2-A1-P also evoked currents in endo-lysosomes isolated from cells expressing TPC2 and TPC2$^{M484L}$ (*Figure 2C and D*). As with TPC2-A1-N, the currents were potentiated by the gain-of-function variant (*Figure 2D*). Surprisingly, the currents evoked by TPC2-A1-P were significantly larger than those evoked by TPC2-A1-N (*Figure 2E–G*) in both wild-type and gain-of-function variant. Full concentration-effect relationships for the plateau response in endo-lysosomal patch-clamp experiments indicated EC$_{50}$ values of 0.6 μM for both TPC2-A1-N and TPC2-A1-P (*Figure 2—figure supplement 1A*).

Increased Na$^+$ currents in the face of reduced $Ca^{2+}$ signals (*Figure 1*) upon TPC2-A1-P activation prompted us to examine the effects of Na$^+$ removal on intracellular $Ca^{2+}$ elevation. For these experiments, we used cells stably expressing TPC2$^{L11A/L12A}$ and agonist concentrations that evoked similar responses. As shown in *Figure 2H and J*, the amplitude of $Ca^{2+}$ signals evoked by TPC2-A1-N were unaffected by removal of Na$^+$ from the extracellular solution, although the rate of rise was increased. In contrast, both the amplitude and rate of rise of the responses to TPC2-A1-P were markedly enhanced (*Figure 2I and J*). These data indicate that TPC2 is activated by TPC2-A1-N and TPC2-A1-P but that its permeability to Na$^+$ differs upon activation.

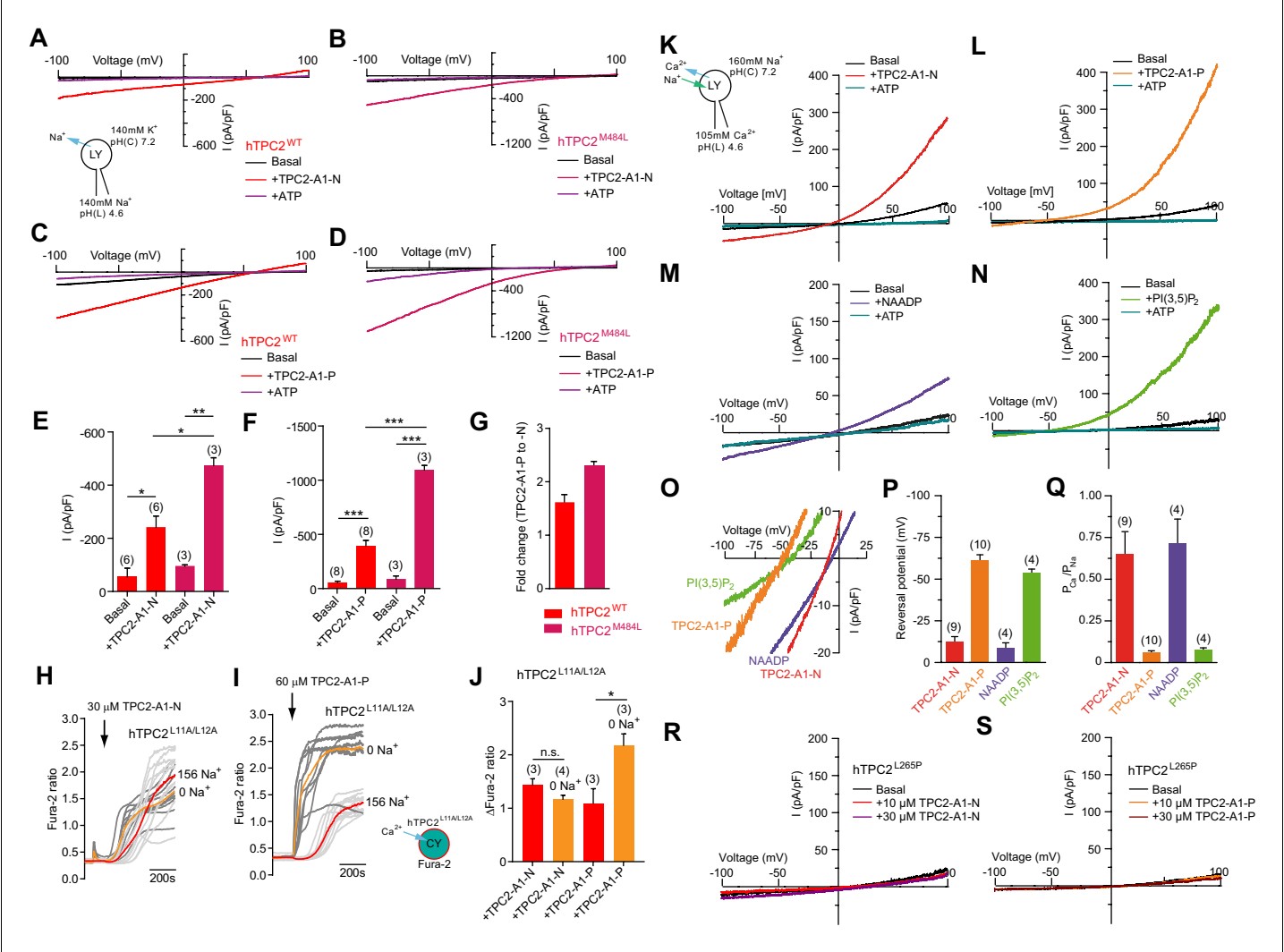

**Figure 2.** TPC2 agonists alter $Ca^{2+}/Na^+$ permeability of TPC2. (A–B) Representative TPC2-A1-N-evoked currents from enlarged endo-lysosomes isolated from HEK293 cells transiently expressing human TPC2 (hTPC2) (A) or a gain-of-function variant, (hTPC2$^{M484L}$) (B). Currents were obtained before and after addition of 10 µM TPC2-A1-N in the absence or presence of 1 mM ATP. Recordings were carried out using standard bath and pipette solutions and applying ramp protocols (−100 mV to +100 mV over 500 ms) every 5 s at a holding potential of −60 mV. (C–D) Similar to A and B except that TPC2-A1-P (10 µM) was used. (E–G) Statistical analysis of current densities (mean ± SEM) recorded at −100 mV for endo-lysosomes expressing TPC2 or TPC2$^{M484L}$ for TPC2-A1-N (E) and TPC2-A1-P (F), and the fold-change in current evoked by TPC2-A1-P versus TPC2-A1-N in the two variants (G). Red bars are WT and maroon bars are mutant. An unpaired t-test was applied. *p<0.05, **p<0.01, and ***p<0.001 (n > 10 endo-lysosomes per condition). (H–I) $Ca^{2+}$ signals evoked by TPC2-A1-N (H) or TPC2-A1-P (I) in Fura-2-loaded HEK293 cells stably expressing hTPC2$^{L11A/L12A}$. Recordings (mean responses from three to four independent experiments) were obtained in standard or Na$^+$-free extracellular solution. (J) Statistical analysis of the maximal change in Fura-2 ratio (mean ± SEM). An unpaired t-test was applied. *p<0.05. (K–N) Agonist-evoked cation currents from enlarged endo-lysosomes isolated from HEK293 cells stably expressing human TPC2 under bi-ionic conditions: 160 mM Na$^+$ in the cytosol (bath) and 105 mM $Ca^{2+}$ in the lumen (pipette). Representative current-voltage curves before and after stimulation with either 10 µM TPC2-A1-N (K), 10 µM TPC2-A1-P (L), 50 nM NAADP (4 recordings out of 8 attempts) (M), or 1 µM PI(3,5)P$_2$ (N). ATP (1 mM) was added at the end of each experiment. Recordings were carried out using ramp protocols (−100 mV to +100 mV over 500 ms) every 5 s at a holding potential of 0 mV. (O) Expanded views of K-N, showing the reversal potentials (E$_{rev}$) of currents evoked by the indicated agonist. (P–Q) Statistical analyses of E$_{rev}$ (P) and the calculated relative cationic permeability ratios ($P_{Ca}/P_{Na}$) (Q). Shown are mean values ± SEM. (R–S) Representative TPC2-A1-N- (R) or TPC2-A1-P- (S) evoked currents from endo-lysosomes isolated from HEK293 cells transiently expressing hTPC2$^{L265P}$ under bi-ionic conditions.

The online version of this article includes the following source data and figure supplement(s) for figure 2:

**Source data 1.** TPC2 agonists alter Ca/Na permeability of TPC2.

**Figure supplement 1.** Dose response curves for TPC2-A1-N and TPC2-A1-P in endo-lysosomal patch-clamp experiments and endo-lysosomal patch-clamp analysis of compound effects under different conditions.

*Figure 2 continued on next page*

*Figure 2 continued*

**Figure supplement 1—source data 1.** Dose response curves for TPC2-A1-N and TPC2-A1-P in endo-lysosomal patch-clamp experiments and endo-lysosomal patch-clamp analysis of compound effects under different conditions.

## TPC2 agonists alter $Ca^{2+}/Na^+$ permeability of TPC2

The above analyses raised the intriguing possibility that the selectivity of TPC2 to $Ca^{2+}$ and $Na^+$ might not be fixed. To directly test this, we measured cation conductance evoked by the two compounds under bi-ionic conditions (luminal side: 105 mM $Ca^{2+}$, pH 4.6, cytosolic side: 160 mM $Na^+$, pH 7.2) (*Figure 2K and L*). For these experiments, we used cells stably expressing TPC2. Based on the reversal potential ($E_{rev}$), the $P_{Ca}/P_{Na}$ permeability ratio of TPC2-A1-N-evoked currents ($I_{TPC2-A1-N}$) was $0.65 \pm 0.13$ (*Figure 2O–Q*). In marked contrast, $E_{rev}$ values for TPC2-A1-P-evoked currents ($I_{TPC2-A1-P}$) were more negative such that the $P_{Ca}/P_{Na}$ permeability ratio was $0.04 \pm 0.01$ (*Figure 2O–Q*) and thus $Na^+$-selective. These ratios are highly reminiscent of those obtained for endogenous NAADP-mediated currents in planar patch-clamp analyses (*Ruas et al., 2015*) and for recombinant TPC2 activated by $PI(3,5)P_2$ measured using endo-lysosomal patch-clamp experiments where the channel appeared insensitive to NAADP (*Wang et al., 2012*). We thus examined the effects of NAADP and $PI(3,5)P_2$ in parallel. NAADP (50 nM) evoked robust currents with an $E_{rev}$ similar to TPC2-A1-N ($-8.5$ mV for NAADP versus $-12$ mV for TPC2-A1-N). The $P_{Ca}/P_{Na}$ for NAADP ($0.73 \pm 0.14$) was thus similar to TPC2-A1-N (*Figure 2M–Q*). In contrast, the $E_{rev}$ for $PI(3,5)P_2$ ($-53$ mV) was similar to TPC2-A1-P ($-61$ mV) and the previously reported $E_{rev}$ for $PI(3,5)P_2$ ($-68$ mV) (*Wang et al., 2012*), corresponding to a $P_{Ca}/P_{Na}$ ratio of $0.08 \pm 0.01$ (*Figure 2M–Q*). No significant outward $Na^+$ or inward $Ca^{2+}$ conductances were observed in endo-lysosomes isolated from HEK293 cells expressing 'pore-dead' $TPC2^{L265P}$ in the presence of TPC2-A1-N or TPC2-A1-P (*Figure 2R–S* and *Figure 2—figure supplement 1B–C*). Moreover, the agonists induced similar shifts in selectivity using luminal solution containing 114 mM $Na^+$ and 30 mM $Ca^{2+}$ and a bath solution containing 160 mM $Na^+$ (*Figure 2—figure supplement 1D–F*) confirming that the measured $P_{Ca}/P_{Na}$ permeability ratio is independent of the luminal $Ca^{2+}$ concentration. These data indicate that TPC2-A1-N and TPC2-A1-P not only differentially affect current amplitudes but also influence the $P_{Ca}/P_{Na}$ permeability ratio (*Figure 2Q*). TPC2-A1-N-activated TPC2 shows a higher relative $Ca^{2+}$ permeability, similar to NAADP-activated TPC2. In contrast, TPC2-A1-P-activated TPC2 shows a lower relative $Ca^{2+}$ permeability, similar to $PI(3,5)P_2$-activated TPC2. Ion permeation through TPC2 is thus ligand-dependent providing an explanation for conflicting evidence that TPC2 is a NAADP-activated $Ca^{2+}$ release channel (*Brailoiu et al., 2010*; *Grimm et al., 2014*; *Pitt et al., 2010*; *Ruas et al., 2015*; *Schieder et al., 2010*) or a $PI(3,5)P_2$ gated $Na^+$ channel (*Boccaccio et al., 2014*; *Guo et al., 2017*; *Wang et al., 2012*).

## TPC2 agonists activate TPC2 through distinct sites

The $PI(3,5)P_2$ binding site in TPC2 has recently been identified (*She et al., 2019*). To gain mechanistic insight into agonist action, we mutated K204 which is required for activation of TPC2 by $PI(3,5)P2$ (*Kirsch et al., 2018*; *She et al., 2019*) and examined its effect on responses to TPC2-A1-N and TPC2-A1-P. Intracellular $Ca^{2+}$ elevation mediated by TPC2-A1-N was similar in cells expressing plasma membrane targeted TPC2 with and without the K204A mutation (*Figure 3A and C*). In contrast, the responses to TPC2-A1-P were reduced approximately two-fold (*Figure 3B and C*). Similar selective inhibition was observed in GCaMP assays (*Figure 3G*). Thus, whereas responses to TPC2-A1-N were comparable for the wild-type and mutant channel, the responses to TPC2-A1-P were reduced by the mutation. Furthermore, currents evoked by TPC2-A1-N were largely unaffected by the K204A mutation (*Figure 3H and J*). In contrast, those to TPC2-A1-P were reduced (*Figure 3I and J*). Collectively, these data suggest K204 is required for activation of TPC2 by TPC2-A1-P and $PI(3,5)P_2$ but not TPC2-A1-N. Molecular determinants for agonist action in TPC2 thus differ.

## TPC2 agonists differentially affect lysosomal pH

Lysosomal function and integrity critically depend on the pH in the lysosomal lumen. Previously, it has been reported that NAADP induced pH changes in the lumen of acidic $Ca^{2+}$ stores, leading to an alkalinization (*Cosker et al., 2010*; *Morgan and Galione, 2007*) but the role of TPC2 in

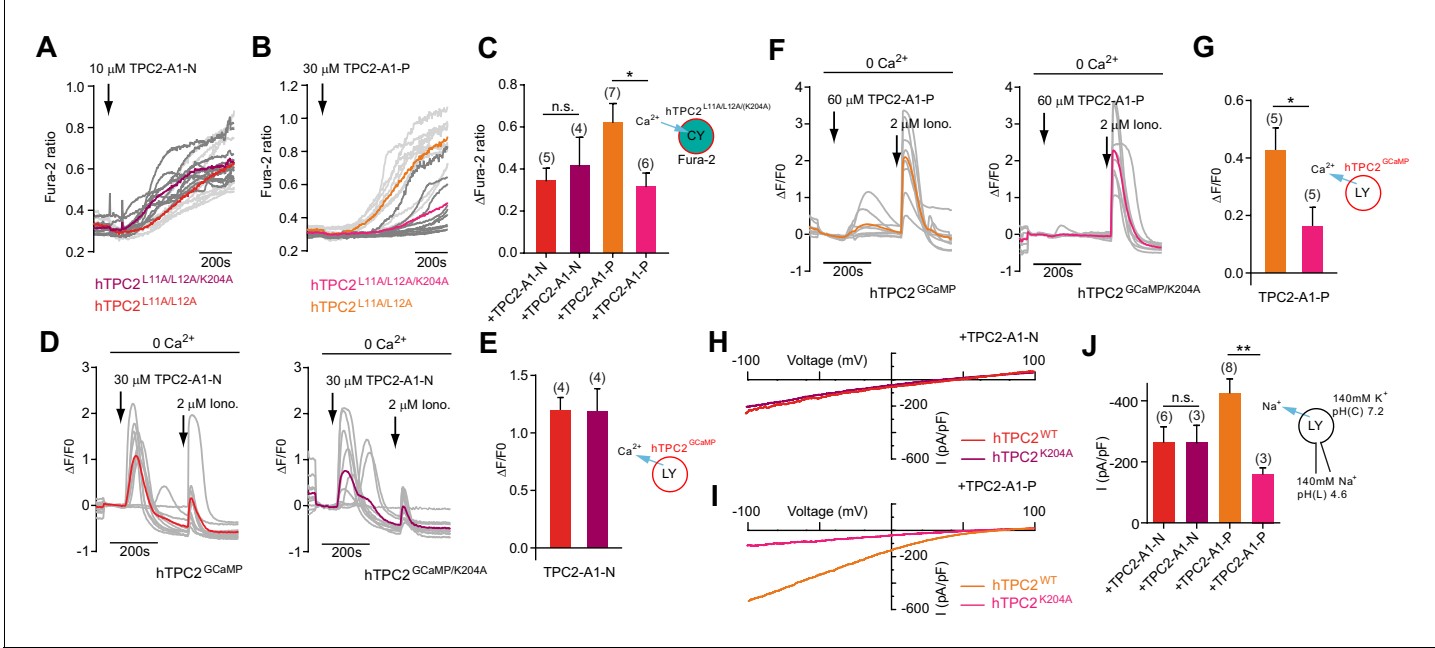

**Figure 3.** TPC2 agonists activate TPC2 through distinct sites. (A–B) Representative $Ca^{2+}$ signals recorded from Fura-2-loaded HeLa cells (n = 10, each) stimulated with the indicated concentration of TPC2-A1-N or -P. Cells were transiently transfected with plasma membrane targeted human TPC2 ($hTPC2^{L11A/L12A}$) or the $PI(3,5)P_2$-insensitive mutant ($TPC2^{L11A/L12A/K204A}$). (C) Statistical analysis of the maximal change in Fura-2 ratio (mean ± SEM) with the number of independent transfections in parentheses. An unpaired t-test was applied. *p<0.05. (D–F) Representative $Ca^{2+}$ signals recorded from HeLa cells (n = 10, each) transiently transfected with GCaMP6(s) fused to wild-type TPC2 ($TPC2^{GCaMP}$) or the $PI(3,5)P_2$-insensitive mutant ($TPC2^{GCaMP/K204A}$). Cells were sequentially stimulated with TPC2-A1-N or -P and the $Ca^{2+}$ ionophore ionomycin in the absence of extracellular $Ca^{2+}$. (E–G) Statistical analysis of the change in fluorescence intensity (mean ± SEM) of experiments as shown in D and F. An unpaired t-test was applied. *p<0.05. (H–I) Representative TPC2-A1-N or -P-evoked currents from enlarged endo-lysosomes isolated from HEK293 cells transiently expressing human TPC2 (hTPC2) or the $PI(3,5)P_2$-insensitive mutant, ($hTPC2^{K204A}$). Currents were obtained before and after addition of 10 µM TPC2-A1-N or -P. Recordings were carried out using standard bath and pipette solutions and applying ramp protocols (−100 mV to +100 mV over 500 ms) every 5 s at a holding potential of −60 mV. (J) Statistical analysis of current densities (mean ± SEM) recorded at −100 mV for endo-lysosomes expressing TPC2 or $TPC2^{K204A}$ using TPC2-A1-N or TPC2-A1-P to activate, respectively. *p<0.05 and **p<0.01, using one-way ANOVA followed by Tukey's post hoc test.

The online version of this article includes the following source data for figure 3:

**Source data 1.** TPC2 agonists activate TPC2 through distinct sites.

regulating lysosomal pH is conflicting (*Ambrosio et al., 2016*; *Cang et al., 2013*; *Grimm et al., 2014*; *Lin et al., 2015*; *Ruas et al., 2015*). We therefore assessed the acute effects of TPC2-A1-N and TPC2-A1-P on lysosomal pH using a recently described novel lysosomal pH sensor, pH-Lemon-GPI (*Burgstaller et al., 2019*; *Figure 4A–K* and *Video 1*). Results from high-resolution array confocal laser scanning microscopy revealed that TPC2-A1-N, but not TPC2-A1-P, increased pH of single vesicles in cells expressing wild-type TPC2 (*Figure 4A–B* and *Video 1*). In contrast, no pH changes were observed in cells expressing $TPC2^{L265P}$. Time-lapse wide-field fluorescence microscopy showed that activation of endogenous TPC2 with TPC2-A1-N modestly increased the lysosomal pH in untransfected cells in a time-dependent manner (*Figure 4C and I*). The pH response was markedly potentiated in cells expressing wild-type TPC2 but not the pore-mutant (*Figure 4D,E and I*). The response in the former was approximately half that evoked by direct alkalization with $NaN_3/NH_4Cl$ (*Figure 4K*). Again, vesicular pH was largely unaffected by TPC2-A1-P (*Figure 4F–H and J*). Time-lapse imaging additionally revealed that vesicle motility was strongly impaired by TPC2-A1-N but not TPC2-A1-P (*Video 2*). The effect of TPC2-A1-N was reversible and appeared to temporally correlate with the increase in vesicular pH (*Video 1*). In endo-lysosomal patch-clamp experiments, we probed the proton permeability of TPC2 and found that in contrast to TPC2-A1-P, TPC2-A1-N rendered the channel proton-permeable, thus providing a direct channel-dependent mechanism leading to alkalinization (*Figure 4L–N*). Proton permeability did not substantially change our estimates of relative $Ca^{2+}$ and $Na^+$ permeability as $P_{Ca}/P_{Na}$ values were similar when proton currents were

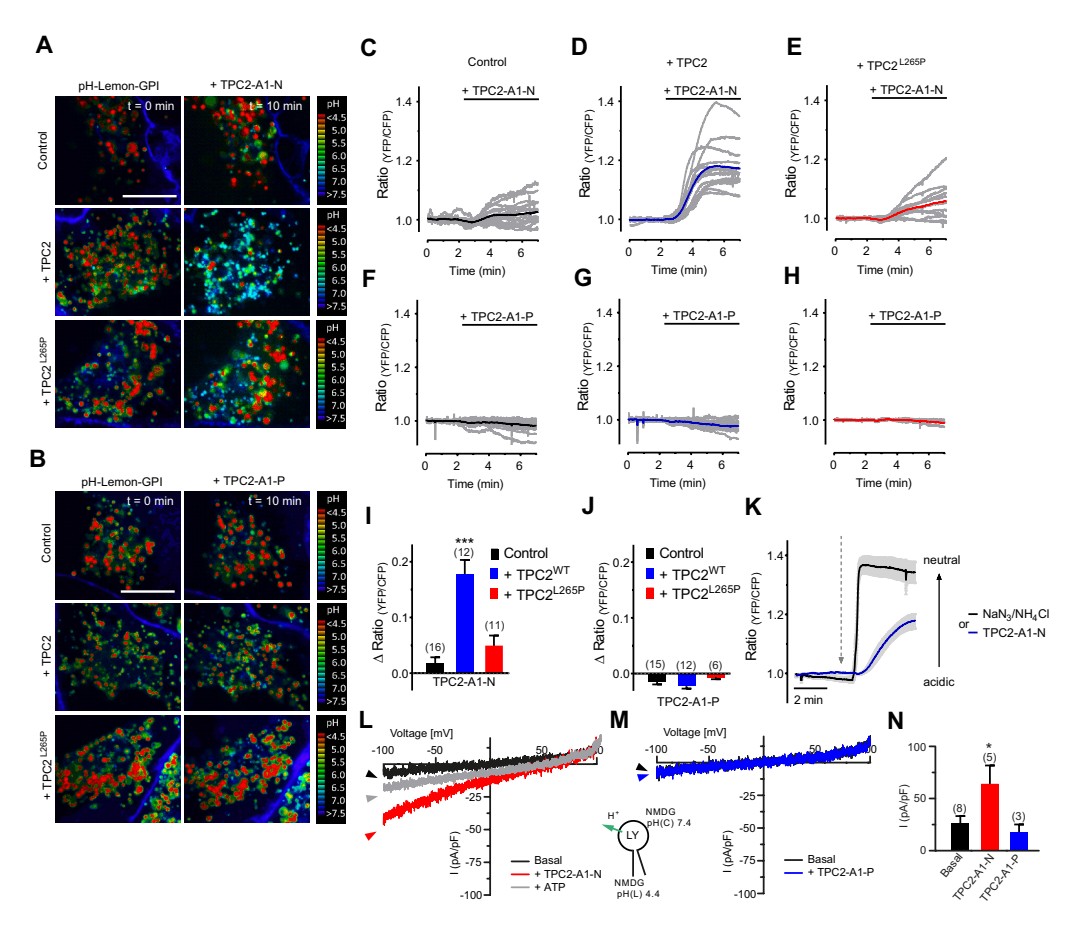

**Figure 4.** TPC2 agonists differentially affect lysosomal pH and proton conductance. (A) Representative pseudo colored ratio images (mTurquoise2/EYFP) of vesicle targeted pH-Lemon-GPI in control HeLa cells (upper panel, n = 4/141 cells), HeLa cells positive for wild-type TPC2-mCherry (middle panel, c = 3/70 cells) and HeLa cells positive for TPC2^L265P-mCherry (lower panel, n = 3/74 cells) in the absence (t = 0 min; left images) and upon treatment for 10 min with 10 µM TPC2-A1-N (t = 10 min; right images). Scale bar = 5 µm. (B) Representative pseudo colored ratio images (mTurquoise2/EYFP) of vesicle targeted pH-Lemon-GPI in control HeLa cells (upper panel, n = 3/121 cells), HeLa cells positive for wild-type TPC2-mCherry (middle panel, n = 3/59 cells) and HeLa cells positive for TPC2^L265P-mCherry (lower panel, n = 3/71 cells) in the absence (t = 0 min; left images) and upon treatment for 10 min with 10 µM TPC2-A1-P (t = 10 min; right images). Scale bar = 5 µm. (C–E) Normalized single cell responses (grey curves) and the respective average response (colored curves) of pH-Lemon-GPI upon treatment with 10 µM TPC2-A1-N in a region of high vesicle density. Shown are ratio curves of pH-Lemon-GPI expressed in control HeLa cells (n = 16 cells; C), in HeLa cells co-expressing wild-type TPC2-mCherry (n = 12 cells; D), and HeLa cells co-expressing TPC2^L265P-mCherry (n = 11 cells; E). (F–H) Experiments as in C-E for TPC2-A1-P. Shown are ratio curves of pH-Lemon-GPI expressed in control HeLa cells (n = 15 cells; F), in HeLa cells co-expressing wild-type TPC2-mCherry (n = 12 cells; G), and HeLa cells co-expressing TPC2L^265P-mCherry (n = 6 cells, (H). (I–J) Columns represent delta ratio values of pH-Lemon-GPI at min six from curves as shown in panels C-E and F-H, respectively. ***p<0.001 using unpaired student's t-test. (K) Average ratio signals over time of pH-Lemon-GPI in HeLa cells expressing wild-type TPC2-mCherry in response to either 0.5% NaN$_3$ and 50 mM NH$_4$Cl (black average curve ± SEM, n = 12 cells) or 10 µM TPC2-A1-N (blue average curve ± SEM, n = 12 cells). Cells were treated with compounds as indicated. (L–M) Representative agonist-evoked inward H$^+$ currents from enlarged endo-lysosomes using hTPC2-expressing HEK293 cells. Both bath and pipette solutions contained 1 mM HCl, 5 mM HEPES, 5 mM MES, 150 mM NMDG, pH as indicated, adjusted with MSA. Ramp protocol (−100 mV to +100 mV in 500 ms, holding voltage = 0 mV) was used. Currents were obtained before and after addition of 10 µM TPC2-A1-N (L) or TPC2-A1-P (M). ATP (1 mM) was added at the end of the experiment to block TPC2 (L). (N) Statistical analysis of current densities (mean ± SEM) recorded at −100 mV (inward H$^+$ conductance; H$^+$ from luminal to cytosolic site) as shown in L and M. *p<0.05, using one-way ANOVA followed by Bonferroni's post hoc test.

The online version of this article includes the following source data for figure 4:

**Source data 1.** Effect of TPC2-A1-N on vesicular pH.

subtracted from currents obtained under bi-ionic conditions (0.44 for TPC2-A1-N and 0.06 for TPC2-A1-P). In sum, these findings demonstrate that TPC2 activation is coupled to lysosomal pH and motility in an agonist-dependent manner.

## TPC2 agonists differentially affect lysosomal exocytosis

To further probe the physiological relevance of TPC2 activation by the agonists, we assessed the effect of TPC2-A1-N and TPC2-A1-P on lysosomal exocytosis (*Figure 5*). Lysosomal exocytosis is involved in a plethora of physiological and pathophysiological processes including release of lysosomal enzymes and inflammatory mediators, clearance of lysosomal storage material and pathogens, plasma membrane repair and cancer progression (*Lopez-Castejon and Brough, 2011*; *Machado et al., 2015*; *Miao et al., 2015*; *Xu and Ren, 2015*). The $Ca^{2+}$-dependence of this process has been mostly ascribed to activation of TRPML1 (*Xu and Ren, 2015*). Here, we used murine macrophages which express high endogenous levels of TPC2 (*Cang et al., 2013*). To assess lysosomal exocytosis, we quantified translocation of LAMP1 to the cell surface. As shown in *Figure 5A–E*, TPC2-A1-N was without effect on lysosomal exocytosis. This was not due to lack of efficacy of TPC2-A1-N on the murine channel because $Ca^{2+}$ measurements with mouse TPC2 fused to GCaMP6 (MmTPC2$^{GCaMP}$) confirmed that TPC2-A1-N evoked larger responses than TPC2-A1-P (*Figure 5—figure supplement 1A*) similar to the human channel (*Figure 1*). In contrast, TPC2-A1-P evoked robust lysosomal exocytosis in a time- and concentration-dependent manner (*Figure 5A–E*), providing a corollary to the effects of the agonists on lysosomal pH. To further corroborate that TPC2-A1-P triggers TPC2-dependent vesicle fusion in primary macrophages, electrophysiological measurements of membrane capacitance via the whole-cell patch-clamp technique were performed (*Figure 5F–H*). These data showed that TPC2-A1-P caused a modest increase in cell size. In the final set of experiments, we examined the effects of TPC2 knockout on agonist responses. As shown in *Figure 5—figure supplement 1*, both TPC2-A1-N and TPC2-A1-P evoked measurable endogenous TPC2-like currents in macrophages from wild-type animals. The amplitudes of the currents were larger for TPC2-A1-P than TPC2-A1-N, again recapitulating agonist effects on human TPC2 (*Figure 2*). Importantly, currents were reduced in macrophages derived from TPC2 KO cells (*Figure 5—figure supplement 1*) and the effects of TPC2-A1-P on lysosomal exocytosis and cell size were abolished (*Figure 5A–E*). TPC2 knockout however failed to affect lysosomal exocytosis in response to ionomycin attesting to specificity (*Figure 5A–E*). Collectively, these data further identify agonist-selective effects of TPC2 on lysosomal physiology in an endogenous setting.

## Discussion

We demonstrate that the ion selectivity of TPC2 is not fixed but rather agonist-dependent. To the best of our knowledge, TPC2 is a unique example of an ion channel that conducts different ions in response to different activating ligands. The novel lipophilic, membrane permeable isoform-selective small molecule agonists of TPC2 (TPC2-A1-N and TPC2-A1-P) characterized herein mimic the physiological actions of NAADP and PI(3,5)P$_2$ most likely through independent binding sites. As such, our data reconcile diametrically opposed opinion regarding activation of TPC2 by endogenous cues and the biophysical nature of the ensuing ion flux. Whether TPC1 also switches its ion selectivity remains to be established. A previous bilayer study found that PI(3,5)P$_2$ did not gate TPC1 but induced modest shifts in relative permeability to different cations when TPC1 was activated by NAADP (*Pitt et al., 2014*).

Physiologically, we demonstrate inverse effects of the agonists on key lysosomal activities. While TPC2-A1-N increases the pH in the lysosomal lumen in a TPC2-dependent manner, TPC2-A1-P has no significant effect on lysosomal pH. An alkalinizing effect of TPC2-A1-N is in accordance with effects described previously for NAADP, which likewise induces alkalinization (*Cosker et al., 2010*; *Morgan and Galione, 2007*) and may be related to agonist-specific effects on proton permeability. The acute nature of these experiments in live cells, made possible by the lipophilicity of TPC2-A1-N, circumvents possible compensatory effects of TPC2 knockout which may have confounded steady-state analyses in previous studies (*Cang et al., 2013*; *Grimm et al., 2014*; *Ruas et al., 2015*). In contrast, TPC2-A1-P but not TPC2-A1-N promoted lysosomal exocytosis. The lack of effect of TPC2-A1-N is surprising given the $Ca^{2+}$-dependence of lysosomal exocytosis but may relate to its inhibitory effect on lysosomal motility. Boosting lysosomal exocytosis is gaining traction as a strategy to

combat disease (*Lopez-Castejon and Brough, 2011*; *Machado et al., 2015*; *Miao et al., 2015*; *Xu and Ren, 2015*). Selective activation of TPC2 in 'PI(3,5)P$_2$-mode' with TPC2-A1-P offers novel scope to achieve this.

In sum, TPC2 can mediate very different physiological and possibly pathophysiological effects depending on how it is activated. TPC2 therefore emerges as an integrator of adenine nucleotide- and phosphoinositide-based messengers with an intrinsic ability to signal through switchable ionic signatures.

# Materials and methods

## Key resources table

| Reagent type (species) or resource | Designation | Source or reference | Identifiers | Additional information |
|---|---|---|---|---|
| Cell line (*Homo-sapiens*) | HEK 293 | DSMZ | ACC 305 | |
| Cell line (*Homo-sapiens*) | HeLa S3 | ATCC | CCL-2.2 | |
| Cell line (*Homo-sapiens*) | HeLa cells stably expressing pH-Lemon-GPI | This paper | | Generated by selective antibiotic G418 disulfate salt from Sigma Aldrich (Cat#A1720). After 4 weeks of cultivation in 800 µg/mL, positive cells were selected by FACS analysis. |
| Cell line (*Homo-sapiens*) | HEK 293 stably expressing TPC2$^{L11A/L12A}$-RFP | This paper | | Generated by selective antibiotic neomycin from Invitrogen (Cat#10486–025) following the guideline V790-20/V795-20 (Invitrogen) |
| Cell line (*Homo-sapiens*) | HEK 293 stably expressing CLN3$^{L253A/I254A}$-RFP | This paper | | Generated by selective antibiotic neomycin from Invitrogen (Cat#10486–025) following the guideline V790-20/V795-20 (Invitrogen) |
| Recombinant DNA reagent | HsTRPML1-YFP | (*Grimm et al., 2010*) PMID:20189104 | | |
| Recombinant DNA reagent | HsTRPML2-YFP | (*Grimm et al., 2010*) PMID:20189104 | | |
| Recombinant DNA reagent | HsTRPML3-YFP | (*Grimm et al., 2010*) PMID:20189104 | | |
| Recombinant DNA reagent | HsTPC2-YFP | (*Chao et al., 2017*) PMID:28923947 | | |
| Recombinant DNA reagent | HsTPC2$^{M484L}$-YFP | (*Chao et al., 2017*) PMID:28923947 | | |
| Recombinant DNA reagent | HsTPC2$^{L11/12A}$-YFP | This paper | | Generated by site-directed mutagenesis from WT plasmid published by *Chao et al. (2017)* |
| Recombinant DNA reagent | HsTPC2$^{L11/12A/L265P}$-YFP | This paper | | Generated by site-directed mutagenesis of WT plasmid published by *Chao et al. (2017)* |
| Recombinant DNA reagent | HsTPC2$^{L11/12A/K204A}$-YFP | This paper | | Generated by site-directed mutagenesis of WT plasmid published by *Chao et al. (2017)* |
| Recombinant DNA reagent | HsTPC2-GCaMP6s | This paper | | Generated by subcloning (*Chao et al., 2017*) WT TPC2 construct into Addgene vector #40753 |
| Recombinant DNA reagent | HsTPC2$^{L265P}$-GCaMP6s | This paper | | Generated by site-directed mutagenesis of WT GCaMP6s plasmid |

*Continued on next page*

*Continued*

| Reagent type (species) or resource | Designation | Source or reference | Identifiers | Additional information |
|---|---|---|---|---|
| Recombinant DNA reagent | HsTPC2$^{K204A}$-GCaMP6s | This paper | | Generated by site-directed mutagenesis of WT GCaMP6s plasmid |
| Recombinant DNA reagent | MmTPC2-GCaMP6s | This paper | | Generated by subcloning GCaMP6s from Addgene vector #40753 into WT MmTPC2 vector |
| Recombinant DNA reagent | HsTPC2$^{L11/12A}$-GFP | (*Brailoiu et al., 2010*) PMID:20880839 | | |
| Recombinant DNA reagent | HsTPC2$^{L11/12A}$-RFP | (*Brailoiu et al., 2010*) PMID:20880839 | | |
| Recombinant DNA reagent | HsTPC2$^{L11/12A/L265P}$-GFP | This paper | | Generated by site-directed mutagenesis of HsTPC2$^{L11/12A}$-GFP (plasmid) |
| Recombinant DNA reagent | TRPML1$^{\Delta NC}$-GFP | (*Yamaguchi et al., 2011*) PMID:21540176 | | |
| Recombinant DNA reagent | HsTPC2$^{L11/12A/K204A}$-GFP | (*She et al., 2019*) PMID:30860481 | | |
| Recombinant DNA reagent | MmTPC1-YFP | (*Zong et al., 2009*) PMID:19557428 | | Generated by subcloning into TOPO pcDNA3.1-YFP, Addgene vector (#13033) |
| Biological sample | Macrophages from TPC2 KO mouse | (*Grimm et al., 2014*) PMID:25144390 | | |
| Antibody | LAMP1 antibody | Santa Cruz Biotechnology | sc19992 | 1:200 |
| Transfection reagent | PolyJet | SignaGen Laboratories | SL100688 | |
| Transfection reagent | TurboFect | Thermo Fisher | R0531 | |
| Transfection reagent | Lipofectamine 2000 | Thermo Fisher | 11668 | |
| Commercial assay or kit | Fluo-4 AM, cell permeant | Thermo Fisher | F14202 | |
| Commercial assay or kit | Fura-2 AM, cell permeant | Thermo Fisher | F1221 | |
| Chemical compound, drug | NAADP | Tocris | 3905 | |
| Chemical compound, drug | PI(3,5)P$_2$ | Echelon Biosciences | P-3508 | |
| Chemical compound, drug | Ionomycin | Sigma Aldrich and Cayman Chemical | I-0634 and 11932 | |
| Chemical compound, drug | Tetrandrine | Sigma Aldrich and Santa Cruz Biotechnology | T2695 and sc201492A | |
| Chemical compound, drug | Raloxifene | Cayman Chemical | 10011620 | |
| Chemical compound, drug | Fluphenazine | Sigma Aldrich | F4765 | |
| Chemical compound, drug | ML-SA1 | Merck | 648493 | |
| Chemical compound, drug | ATP | Sigma Aldrich | A9187 | |
| Chemical compound, drug | Sodiumazide (NaN$_3$) | Sigma Aldrich | 09718 | |

*Continued on next page*

*Continued*

| Reagent type (species) or resource | Designation | Source or reference | Identifiers | Additional information |
|---|---|---|---|---|
| Chemical compound, drug | Ammoniumchloride (NH$_4$Cl) | Sigma Aldrich | S2002 | |
| Software, algorithm | Origin8 | OriginLab | | |
| Software, algorithm | GraphPad Prism | GraphPad Software Inc | | |
| Other | Glass Bottom Dish 35 mm | ibidi | 81218 | |
| Other | Perfusion Chamber PC30 | Next Generation Fluorescence Imaging | PC30 (www.ngfi.eu) | Perfusion chamber used with gravity based perfusion system (NGFI, Graz, Austria) |
| Other | μ-Slide 8 Well | ibidi | 80826 | |

## High-throughput screening

The high-throughput screen was performed using a custom-made fluorescence imaging plate reader built into a robotic liquid handling station (Freedom Evo 150, Tecan, Männedorf, Switzerland) as previously described (*Urban et al., 2016*). HEK293 cells were used that stably expressed RFP fusion proteins of human TPC2 (C-terminally tagged) or human CLN3 (N-terminally tagged) rerouted to the plasma membrane. Targeting was achieved by mutation of the endo-lysosomal targeting motifs (hTPC2$^{L11A/L12A}$, hCLN3$^{L253A/I254A}$). Mutants were generated by site-directed mutagenesis using the QuikChange II XL protocol (Agilent), according to the manufacturer's instructions. Stable HEK293 cell lines were generated using 400 μg/mL geneticin (G418, Sigma). If G418-resistant foci were not identified after 3–4 days, the concentration of G418 was increased to 800 μg/mL. After 2–3 weeks cells were picked from G418-resistant foci and colonies were expanded in six well plates. RFP expression was assessed using confocal microscopy when cells were >50% confluent. Colonies with more than 95% RFP positive cells were selected, grown to >90% confluency, split and further expanded. For HTS experiments, cells were cultured at 37°C with 5% of CO$_2$ in Dulbecco's modified Eagle medium (Thermo Fisher), supplemented with 10% fetal calf serum (Biochrom, Berlin, Germany), 2 mM L-glutamine, 100 U/mL penicillin, 0.1 mg/mL streptomycin, and 400–800 μg/mL G418. Cells were seeded on black-walled, clear-bottom 384-well plates (Greiner, Germany) and incubated with Fluo-4/AM (4 μM; Life Technologies, Eugene, OR) for 30 min at 37°C, washed and resuspended in a HEPES-buffered solution 1 (HBS1) comprising 132 mM NaCl, 6 mM KCl, 1 mM MgCl$_2$, 1 mM CaCl$_2$, 10 mM HEPES, and 5.5 mM D-glucose (pH was adjusted to 7.4 with NaOH). For primary screening, individual compounds from Roche libraries (Xplore libraries X30 and X50, Roche, Basel, CH) were diluted in HBS1 to a working concentration of 100 μM. After recording the baseline for 30 s, compounds were injected to a final concentration of 10 μM. Recording continued for 180 s per quadrant (total 750 s). If high intensities were measured in both cell lines, the compounds were deemed false positives and excluded. Concentration-effect relationships were plotted using GraphPad Prism five and fitted to the Hill equation. The identity of the cell lines used has been authenticated by STR profiling. No mycoplasma contamination has been reported.

## Ca$^{2+}$ imaging

Single cell Ca$^{2+}$ imaging experiments were performed using Fura-2. HeLa cells and HEK293 cells were cultured at 37°C with 5% of CO$_2$ in Dulbecco's modified Eagle medium (Gibco), supplemented with 10% fetal bovine serum, 100 U/mL penicillin, and 0.1 mg/mL streptomycin. Cells were plated onto poly-L-lysine (sigma)-coated glass coverslips, grown overnight and transiently transfected for 18–24 hr with plasmids using lipofectamine 2000 (Invitrogen) or TurboFect (Thermo Fisher) according to the manufacturer's instructions. For Ca$^{2+}$ influx experiments, cells were transfected with human TPC2 (C-terminally tagged with GFP or RFP) (*Brailoiu et al., 2010*) or TRPML1 (N-terminally-tagged with GFP (*Yamaguchi et al., 2011*) or C-terminally tagged with YFP). They were targeted to the plasma membrane by mutation/deletion of the endo-lysosomal targeting motifs. For Ca$^{2+}$ imaging experiments with TRPML2 or TRPML3 (both C-terminally tagged with YFP *Grimm et al., 2010*), the latter two sufficiently locating to the plasma membrane when overexpressed. A pore-dead mutant of plasma membrane, GFP-tagged TPC2 (hTPC2$^{L11A/L12A/L265P}$) was generated by site-

directed mutagenesis. Transfected cells were loaded for 1 hr at room temperature with Fura-2 AM (2.5 μM) and 0.005% (v/v) pluronic acid (both from Invitrogen) in HEPES-buffered solution 2 (HBS2) comprising 1.25 mM $KH_2PO_4$, 2 mM $CaCl_2$, 2 mM $MgSO_4$, 3 mM KCl, 156 mM NaCl, 10 mM D-glucose and 10 mM HEPES (adjusted to pH 7.4 with HCl). After loading, cells were washed three times in HBS2, and mounted in an imaging chamber. All recordings were performed in HBS2. $Ca^{2+}$ imaging experiments were also performed in HEK293 cells stably expressing C-terminally RFP-tagged hTPC2$^{L11A/L12A}$ (see above). These cells were plated, loaded with Fura-2 and recorded as with HeLa cells, except for that in indicated experiments, NaCl in HBS2 was replaced with NMDG. For lysosomal $Ca^{2+}$ release experiments, HeLa cells were transfected with human TPC2, C-terminally tagged with GCaMP6(s) (hTPC2$^{GCaMP}$). The hTPC2 sequence was amplified by PCR from the pcDNA3.1 plasmid encoding YFP-tagged hTPC2 as described previously (*Chao et al., 2017*), using a forward primer carrying a NheI restriction site followed by a Kozak sequence, and a sequence binding the first 17 base-pairs of hTPC2: TATGCTAGCGCCACCATGGCGGAACCCCAGGC. The reverse primer also contained a NheI restriction site, preceded by a GSG-linker coding sequence, and the final 20 base-pairs of hTPC2, excluding the TPC2 stop codon: ATAGCTAGCACCAGAACCCCTGCACAGC-CACAGGTG. The PCR product was cloned into the GCaMP6(s) (Addgene ID 40753) vector by insertion into a NheI site, yielding plasmids encoding 6xHis tag-HsTPC2-GSG-Xpress tag-GCaMP6 fusion proteins. The ligated plasmid was sequenced by Sanger sequencing using standard CMV forward and pEGFP reverse primers, to confirm the insertion took place as desired. A pore dead mutant (hTPC2$^{GCaMP/L265P}$) was generated by site-directed mutagenesis using the QuikChange II XL protocol (Agilent), according to the manufacturer's instructions, using the forward primer sequence GAGTC TCTGACTTCCCCCCTGGTGCTGCTGAC (reverse primer reverse complement of the former). All recordings were performed in nominally $Ca^{2+}$-free HBS2. Images were acquired every 3 s at 20X (Fura-2) or 40X magnification using a cooled coupled device camera (TILL photonics) attached to an Olympus IX71 inverted fluorescence microscope fitted with a monochromator light source. Fura-2 was excited at 340 nm/380 nm, and GCaMP6(s) was excited at 470 nm. Emitted fluorescence was captured using 440 nm or 515 nm long-pass filters, respectively.

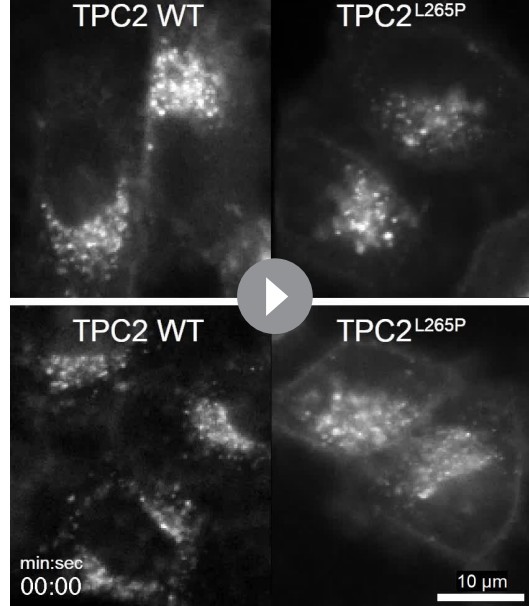

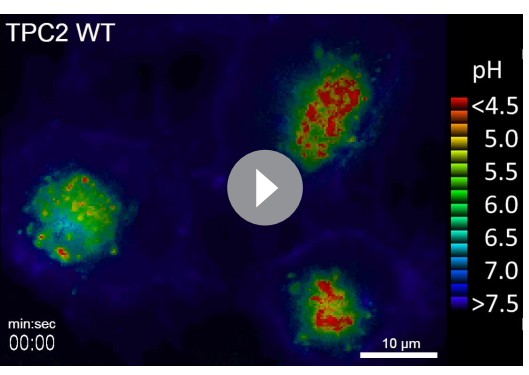

**Video 1.** TPC2-A1-N effectively increases vesicular pH in HeLa cells. Dynamic pseudo-colored ratio (mTurquoise2/EYFP) video of HeLa cells stably expressing pH-Lemon-GPI and co-expressing TPC2 WT upon addition of TPC2-A1-N at the time-point indicated. Pseudo-colored ratio scale with estimated pH values is shown on the right. Scale bar represents 10 μm.
https://elifesciences.org/articles/54712#video1

**Video 2.** Vesicle movement is impaired upon treatment with TPC2-A1-N, but not with TPC-A1-P. HeLa cells stably expressing pH-Lemon-GPI and co-expressing either TPC2 WT (upper left panel and lower left panel) or TPC2$^{L265P}$ (upper right panel and lower right panel) were analyzed. Shown are the mTurquoise2 fluorescence signals of pH-Lemon over time upon treatment with TPC2-A1-N (upper panels) and TPC2-A1-P (lower panels) at the time-points indicated. Time-lapse stacks were acquired using wide-field microscopy. Scale bar represents 10 μm.
https://elifesciences.org/articles/54712#video2

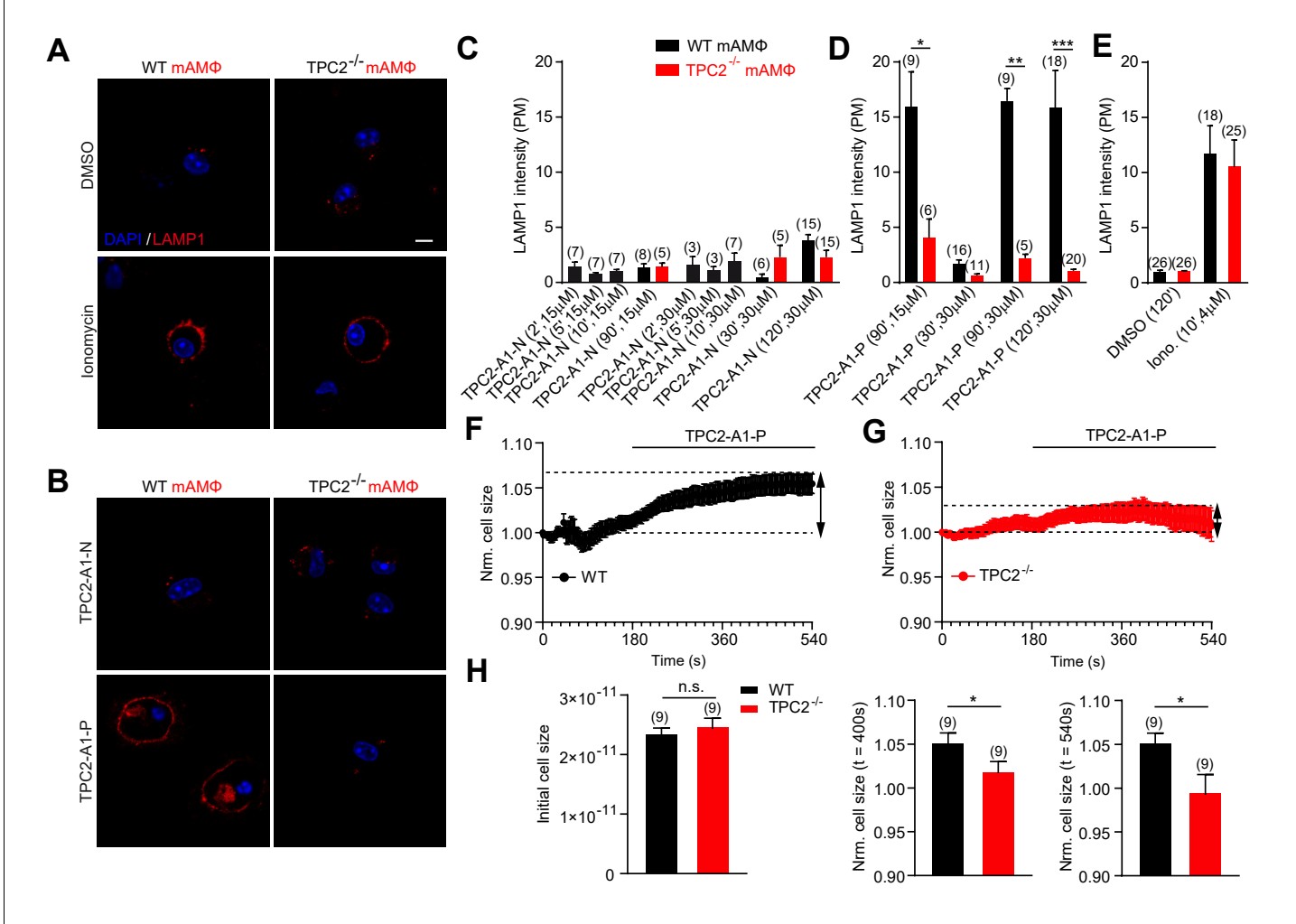

**Figure 5.** TPC2 agonists differentially affect lysosomal exocytosis. (A–B) Representative images of LAMP1 translocation assay using murine wild-type and TPC2 KO alveolar macrophages. Shown are results obtained after 120 min treatment with either DMSO, TPC2-A1-P (30 μM) or TPC2-A1-N (30 μM), or 10 min treatment with ionomycin (4 μM). (C–E) Statistical analysis of experiments as shown in A and B at different compound concentrations and incubation times as indicated. Shown are mean values ± SEM. **p<0.01, and ***p<0.001, using two-way ANOVA followed by Bonferroni's post hoc test. (F–G) Electrophysiological measurements of membrane capacitance via the whole-cell patch-clamp technique as an estimate of cell size (in picofarad [pF]) were used to record fusion of vesicles of primary alveolar macrophages isolated from WT (black, n = 9, F) or TPC2 KO (red, n = 9, G) mice. The normalized, averaged cell size was plotted versus time of the experiment. TPC2-A1-P (30 μM) was applied at 180 s until the end of the experiment indicated by the black bar. Data are shown as mean ± SEM. (H) Bar graphs show mean cell sizes at different time points. The initial cell size was measured immediately after whole cell break in pF and was used for normalization (left panel). Statistical analysis at 400 s (middle panel) and at 540 s (right panel). *p<0.05, unpaired student's t-test.

The online version of this article includes the following source data and figure supplement(s) for figure 5:

**Source data 1.** TPC2 agonists differentially affect lysosomal exocytosis.

**Figure supplement 1.** Effect of TPC2-A1-N and TPC2-A1-P on mouse TPC2 in overexpressing and endogenously expressing cells.

## Endo-lysosomal patch-clamp experiments

Manual whole-endo-lysosomal patch-clamp recordings were performed as described previously (*Chen et al., 2017*). HEK293 cells were plated onto poly-L-lysine (Sigma)-coated glass coverslips, grown over night and transiently transfected for 17–25 hr with plasmids using TurboFect (Thermo Fisher) according to the manufacturer's instructions. Cells expressing wild-type (hTPC2) and a gain-of-function variant (hTPC2^{M484L}) of human TPC2 tagged at their C-termini with YFP were used (*Chao et al., 2017*). Cells were treated with either vacuolin or YM201636 (1 μM and 800 nM over-night, respectively) to enlarge endo-lysosomes. Currents were recorded using an EPC-10 patch-

clamp amplifier (HEKA, Lambrecht, Germany) and PatchMaster acquisition software (HEKA). Data were digitized at 40 kHz and filtered at 2.8 kHz. Fast and slow capacitive transients were cancelled by the compensation circuit of the EPC-10 amplifier. Glass pipettes for recording were polished and had a resistance of 4–8 MΩ. For all experiments, salt-agar bridges were used to connect the reference Ag-AgCl wire to the bath solution to minimize voltage offsets. Liquid junction potential was corrected as described (*Chen et al., 2017*). For the application of agonists, cytoplasmic solution was completely exchanged. Unless otherwise stated, the cytoplasmic solution comprised 140 mM K-MSA, 5 mM KOH, 4 mM NaCl, 0.39 mM $CaCl_2$, 1 mM EGTA and 10 mM HEPES (pH was adjusted with KOH to 7.2). Luminal solution comprised 140 mM Na-MSA, 5 mM K-MSA, 2 mM Ca-MSA, 1 mM $CaCl_2$, 10 mM HEPES and 10 mM MES (pH was adjusted to 4.6 with MSA). 500 ms voltage ramps from −100 to +100 mV were applied every 5 s, holding potential at 0 mV. The current amplitudes at −100 mV were extracted from individual ramp current recordings. For MmTPC1 measurements a one step protocol was applied (+140 mV over 2 s, holding potential of −70 mV) and the cytoplasmic solution contained 140 mM Na-gluconate, 5 mM NaOH, 4 mM KCl, 2 mM MgCl2, 0.39 mM $CaCl_2$, 1 mM EGTA and mM 10 HEPES (pH 7.2). The luminal solution contained 140 mM Na-MSA, 5 mM K-MSA, 2 mM Ca-MSA, 1 mM $CaCl_2$, 10 mM HEPES and 10 mM MES (pH was adjusted to 4.6 with MSA). All statistical analyses were done using Origin8 software.

For analyses under bi-ionic conditions, HEK293 cells stably expressing hTPC2 tagged at its C-terminus with YFP were used. The cytoplasmic solution comprised 160 mM NaCl and 5 mM HEPES (pH was adjusted with NaOH to 7.2) and the luminal solution comprised 105 mM $CaCl_2$, 5 mM HEPES and 5 mM MES (pH was adjusted to 4.6 with MSA). The permeability ratio ($P_{Ca}/P_{Na}$) was calculated according to *Fatt and Ginsborg (1958)*:

$$\frac{P_{Ca}}{P_{Na}} = \frac{\gamma_{Na}}{\gamma_{Ca}} \cdot \frac{[Na]_i}{4[Ca]_o} \cdot exp^{\frac{E_{rev}F}{RT}} \cdot \left( exp^{\frac{E_{rev}F}{RT}} + 1 \right)$$

for bi-ionic test (*Figure 2Q* and *Figure 2—figure supplement 1F*), and equation according to Eq. 13.47 from *Jackson (2006)*,

$$\frac{P_{Ca}}{P_{Na}} = \frac{\gamma_{Na}}{\gamma_{Ca}} \cdot \frac{\left( [Na]_i \cdot exp^{\frac{E_{rev}F}{RT}} - [Na]_o \right)}{4[Ca]_o} \cdot \left( exp^{\frac{E_{rev}F}{RT}} + 1 \right)$$

for internal and external solutions containing the same monovalent and divalent cations (*Figure 2—figure supplement 1D–F*). $P_{Ca}$ = $Ca^{2+}$ permeability; $P_{Na}$ = $Na^+$ permeability; $\gamma_{Ca}$ = $Ca^{2+}$ activity coefficient (0.52); $\gamma_{Na}$ = $Na^+$ activity coefficient (0.75); $[Ca]_o$ = concentration of $Ca^{2+}$ in the lumen; $[Na]_i$ = concentration of $Na^+$ in the cytosol; $[Na]_o$ = concentration of $Na^+$ in the lumen; $E_{rev}$ = reversal potential; F, R = standard thermodynamic constants; T = temperature.

## Isolation of murine alveolar macrophages

For preparation of primary alveolar macrophages, mice were deeply anesthetized and euthanized by exsanguination. The trachea was exposed and cannulated by inserting a 20-gauge catheter (B. Braun). Cells were harvested by eight consecutive lung lavages with 1 mL of DPBS (Dulbeccos's Phosphate-Buffered Saline) and cultured in RPMI 1640 medium supplemented with 10% fetal bovine serum and 1% antibiotics. For experimentation, alveolar macrophages were directly seeded onto 12 mm glass cover slips and used within 5 days after preparation.

## Lysosomal exocytosis experiments

Alveolar macrophages ($3 \times 10^4$), isolated from wild-type and TPC2 KO mice (described in *Grimm et al., 2014*) were seeded on 8-well plates (Ibidi) and cultured overnight. Cells were washed once with Minimum Essential Media (MEM) supplemented with 10 mM HEPES and then treated with TPC2-A1-N or TPC2-A1-P as indicated. Ionomycin (4 µM for 10 min) was used as positive control. Following treatment, cells were incubated with an anti-LAMP1 antibody (1:200, SantaCruz) in MEM supplemented with 10 mM HEPES and 1% BSA for 20 min on ice. Cells were then fixed with 2.6% PFA (Thermo Fisher) for 20 min and incubated with Alexa Fluor 488 conjugated secondary antibody (Thermo Fisher) for 1 hr in PBS containing 1% BSA. Nuclei were stained with DAPI. Confocal images were acquired using an LSM 880 microscope (Zeiss) with 40X magnification.

## Capacitance measurements

Vesicle fusion of wild-type and TPC2-deficient primary alveolar macrophages was analyzed by measuring cell membrane capacitance using the patch-clamp technique as previously described (*Zierler et al., 2016*). In brief, vesicle fusion was recorded using the automated capacitance cancellation function of the EPC-10 (HEKA, Lambrecht, Germany). Measurements were performed in a tight seal whole-cell configuration at room temperature. Membrane capacitance values directly captured after breaking the seal between the membrane and the glass pipette were used as a reference for the initial cell size. Further recorded capacitance values were normalized to this initial determined capacitance. Extracellular solution contained: 140 mM NaCl, 1 mM $CaCl_2$, 2.8 mM KCl, 2 mM $MgCl_2$, 10 mM HEPES-NaOH, 11 mM glucose (pH 7.2, 300 mosmol/L). Internal solution contained: 120 mM potassium glutamate, 8 mM NaCl, 1 mM $MgCl_2$, 10 mM HEPES-NaOH, 0.1 mM GTP (pH 7.2, 280 mosmol/L). At 180 s TPC2-A1-P (30 µM, diluted in external solution) was applied via an application pipette.

## Lysosomal pH measurements

Cell culture and transfection: DMEM (Sigma Aldrich) containing 10% fetal bovine serum, 100 U/mL penicillin, 100 µg/mL streptomycin, and 2.5 µg/mL Fungizone (Thermo Fisher) was used to grow HeLa cells (all obtained from ATCC, Guernsey, UK). Transfection of cells in six-well (Greiner-Bio-One, Kremsmünster, Austria) was performed using PolyJet transfection reagent (SignaGen Laboratories, Rockville). HeLa cells stably expressing pH-Lemon-GPI (NGFI, Graz, Austria) were generated by selection with 800 µg/mL of G418 (Sigma Aldrich, St. Louis, USA) and FACS sorting for CFP (excitation at 405 nm). Wide-field fluorescence microscopy experiments were performed using an OLYMPUS IX73 inverted microscope (OLYMPUS, Vienna, Austria) using a 40X objective (UApo N 340, 40X/1.35 Oil, ∞/0.17/FN22, OLYMPUS). Illumination was performed using an OMICRON LedHUB High-Power LED Light Engine, equipped with a 455 nm and 505 nm LED light source (OMICRON electronics, Vienna, Austria), and 430 nm and 500 nm excitation filters (AHF Analysentechnik, Tübingen, Germany), respectively. Images were captured at a binning of 2 using a Retiga R1 CCD camera (TELEDYNE QIMAGING, Surrey, Canada) and emissions were separated using an optical beam splitter (DV2, Photometrics, Arizona). Array Confocal microscopy was performed using an array confocal laser scanning microscope (ACLSM) built on a fully automatic inverse microscope (Axio Observer.Z1, Zeiss, Göttingen, Germany) using a 100x objective (Plan-Apochromat 100x, 1.4 Oil M27). Excitation was performed using laser light of diode lasers (Visitron Systems): CFP of pH-lemon was excited at 445 nm and 505 nm respectively. Emitted light was acquired with emission filters ET480/40 m for CFP and ET525/50 m for EYFP (Chroma Technologies, VT). A Photometrics CCD camera (CoolSnap HQ2) was used to capture all images at a binning of 1. Device control and image acquisition was performed using VisiView image acquisition and control software (Visitron Systems, Puchheim, Germany) for both devices. Images were processed using MetaMorph analysis software (Molecular Devices, San Jose). Data and statistical analyses were done using GraphPad Prism Software.

## Synthesis of the compounds

All chemicals used were of analytical grade and were obtained from abcr (Karlsruhe, Germany), Fischer Scientific (Schwerte, Germany), Sigma-Aldrich (now Merck, Darmstadt, Germany), TCI (Eschborn, Germany) or Th. Geyer (Renningen, Germany). HPLC grade and dry solvents were purchased from VWR (Darmstadt, Germany) or Sigma-Aldrich, all other solvents were purified by distillation. Hydrophobic phase separation filters (MN 617 WA, 125 mm) were purchased from Macherey Nagel (Düren, Germany). All reactions were monitored by thin-layer chromatography (TLC) using pre-coated plastic sheets POLYGRAM SIL G/UV254 from Macherey-Nagel and detected by irradiation with UV light (254 nm or 366 nm). Flash column chromatography (FCC) was performed on Merck silica gel Si 60 (0.015–0.040 mm).

NMR spectra ($^{1}$H, $^{13}$C, DEPT, COSY, HSQC/HMQC, HMBC) were recorded at 23°C on an Avance III 400 MHz Bruker BioSpin or Avance III 500 MHz Bruker BioSpin instrument, unless otherwise specified. Chemical shifts $\delta$ are stated in parts per million (ppm) and are calibrated using residual protic solvents as an internal reference for proton (CDCl$_3$: $\delta$ = 7.26 ppm, (CD$_3$)$_2$SO: $\delta$ = 2.50 ppm) and for carbon the central carbon resonance of the solvent (CDCl$_3$: $\delta$ = 77.16 ppm, (CD$_3$)$_2$SO: $\delta$ = 39.52 ppm). Multiplicity is defined as s = singlet, d = doublet, t = triplet, q = quartet, sext = sextet,

m = multiplet. NMR spectra were analyzed with NMR software MestReNova, version 12.0.1–20560 (Mestrelab Research S.L.). High-resolution mass spectra were performed by the LMU Mass Spectrometry Service applying a Thermo Finnigan MAT 95 or Joel MStation Sektorfeld instrument at a core temperature of 250°C and 70 eV for EI or a Thermo Finnigan LTQ FT Ultra Fourier Transform Ion Cyclotron Resonance device at 250°C for ESI. IR spectra were recorded on a Perkin Elmer FT-IR Paragon 1000 instrument as neat materials. Absorption bands were reported in wave number (cm$^{-1}$) with ATR PRO450-S. Melting points were determined by the open tube capillary method on a Büchi melting point B-540 apparatus and are uncorrected. Microwave-assisted reactions were carried out in a Discover (S-Class Plus) SP microwave reactor (CEM GmbH, Kamp-Lintfort, Germany). HPLC purities were determined using an HP Agilent 1100 HPLC with a diode array detector and an Agilent Poroshell column (120 EC-C18; 3.0 × 100 mm; 2.7 micron) with acetonitrile/water as eluent (60:40 acetonitrile/water + 0.1% formic acid).

## Preparation of the TPC2-A1-N series
### General procedure A – Amide coupling

According to *Sjogren et al. (1991)* the appropriate aniline (1.0 eq.) and 2-cyanoacetic acid (1.0 eq.) were dissolved in DMF and cooled to 0°C. DCC (1.0 eq.) was added portion wise. The mixture was warmed up to rt over 1 hr and subsequently diluted with hexanes/EtOAc (1:1). Precipitates were removed by filtration and the filtrate was extracted once with 1 M aq. HCl and thrice with EtOAc. The combined organic layers were washed with sat. aq. NaCl solution, dried over $Na_2SO_4$, filtered and concentrated *in vacuo*. Recrystallization from EtOH yielded the desired amides.

### General procedure B – synthesis of *N*-aryl cyanoacetamides

According to *Sjogren et al. (1991)* the appropriate amides received from general procedure **A** (1.0 eq.) were dissolved in dry THF, the solution was cooled to 0°C and NaH (dispersion in paraffin, 60%, 2.3 eq.) was added. After stirring for 15 min, the appropriate benzoyl chloride (1.1 eq.) was added. The mixture was stirred at 0°C for 1 hr and then cautiously treated with 1 M HCl. The precipitate was collected by filtration, washed with ice water and cold EtOH and recrystallized from toluene to give the desired cyanoacetamides.

If the appropriate benzoyl chloride was not commercially available, it was prepared by refluxing the appropriate benzoic acid (1.1 eq.) in $SOCl_2$ (55 eq.) for 1 hr and concentrating *in vacuo*. The resulting acid chloride was immediately transferred to the reaction.

## Preparation of the TPC2-A1-P series
### General procedure C – Paal-Knorr pyrrole synthesis

Following a general procedure published by *Kang et al. (2010)* the appropriate β-ketoester (1.1 eq.) was dissolved in dry THF and cooled to 0°C, before NaH (dispersion in paraffin, 60%, 1.5 eq.) was added portion wise. After the suspension was stirred for 30 min, a solution of appropriate halogenated acetophenone (1.0 eq.) and KI (1.0 eq.) in dry THF was added dropwise. The reaction mixture was allowed to warm up to rt over 2 hr, then poured on water and extracted three times with diethyl ether. The combined organic phases were washed with sat. aq. $NaHCO_3$ solution, dried over $Na_2SO_4$, filtered and concentrated *in vacuo*. The residue was dissolved in acetic acid and the appropriate primary amine (2.0 eq.) was added dropwise. The reaction mixture was stirred at 80°C for 18 hr. The solvent was removed *in vacuo*, the residue disperged in water and extracted three times with diethyl ether. The collected organic phases were washed with sat. aq. $NaHCO_3$ solution, dried over $Na_2SO_4$ and concentrated *in vacuo*. Purification was accomplished by FCC and recrystallization from EtOH if not otherwise specified.

### General procedure D – alkaline deprotection of the pyrrolecarboxylic esters

LiOH (10 eq.) was added to a solution of the appropriate ester (1.0 eq.) in dioxane/$H_2O$ (5:1) and the reaction mixture was stirred in a closed vessel under microwave irradiation ($p_{max}$ = 8 bar, $P_{max}$ = 200 W, $T_{max}$ = 180°C) for 1–18 hr. The suspension was diluted with water to thrice original volume and aq. 2 M HCl was added dropwise under vigorous stirring until the mixture was strongly

acidic. The formed precipitate was collected by filtration, washed with water and dried. If necessary the acids were recrystallized from EtOH to yield the pure products.

## Synthesis of TPC2-A1-N and analogs

**Chemical structure 1.** 2-Cyano-*N*-(4-(trifluoromethyl)phenyl)acetamide – SGA-34.

According to general procedure **A**, 4-(trifluoromethyl)aniline (812 µL, 6.47 mmol, 1.1 eq.), 2-cyano-acetic acid (500 mg, 5.88 mmol, 1.0 eq.) and DCC (1.27 g, 6.17 mmol, 1.1 eq.) in DMF (7.0 mL) were used to yield amide **SGA-34** as colorless crystals (983 mg, 4.31 mmol, 73%). Analytical data are in accordance with literature (*Davies et al., 2009*; *Sjogren et al., 1991*). $R_f$ = 0.14 (4:1 hexanes/ace-tone). **m.p.:** 195°C [(*Sjogren et al., 1991*): 191–193°C]. $^1$**H NMR (400 MHz, (CD$_3$)$_2$SO)** $\delta$/ppm = 10.65 (s, 1H, NH), 7.75 (d, $J$ = 8.8 Hz, 2H, 3'-H, 5'-H), 7.70 (d, $J$ = 8.8 Hz, 2H, 2'-H, 6'-H), 3.95 (s, 2H, 2 hr). $^{13}$**C NMR (101 MHz, (CD$_3$)$_2$SO)** $\delta$/ppm = 161.9 (C-1), 141.9 (C-1'), 126.3 (q, $J_{CF}$ = 3.7 Hz, C-3', C-5'), 124.4 (q, $J_{CF}$ = 271.4 Hz, CF$_3$), 124.0 (q, $J_{CF}$ = 32.0 Hz, C-4'), 119.2 (C-2', C-6'), 115.7 (CN), 27.0 (C-2). **IR (ATR)** $\tilde{v}_{max}$/cm$^{-1}$=3287, 3221, 3147, 1681, 1612, 1557, 1319, 1110, 1065, 849, 835. **HRMS (ESI):** calcd. for C$_{10}$H$_6$F$_3$N$_2$O (M-H)$^-$ 227.04377; found 227.04371. **Purity (HPLC):**>96% ($\lambda$ = 210 nm),>96% ($\lambda$ = 254 nm).

**Chemical structure 2.** 2-Cyano-*N*-(*p*-tolyl)acetamide (1).

According to general procedure **A**, *p*-toluidine (712 µL, 6.47 mmol, 1.1 eq.), 2-cyanoacetic acid (500 mg, 5.88 mmol, 1.0 eq.) and DCC (1.27 g, 6.17 mmol, 1.1 eq.) in DMF (7.0 mL) were used to yield amide one as colorless crystals (728 mg, 4.18 mmol, 71%). Analytical data are in accordance with literature (*Yuan et al., 2019*). $R_f$ = 0.14 (4:1 hexanes/acetone). **m.p.:** 184°C [(*Yuan et al., 2019*): 186°C]. $^1$**H NMR (400 MHz, (CD$_3$)$_2$SO)** $\delta$/ppm = 10.19 (s, 1H, NH), 7.47–7.36 (m, 2H, 2'-H, 6'-H), 7.13 (d, $J$ = 8.2 Hz, 2H, 3'-H, 5'-H), 3.86 (s, 2H, 2 hr), 2.25 (s, 3H, CH$_3$). $^{13}$**C NMR (101 MHz, (CD$_3$)$_2$SO)** $\delta$/ppm = 160.7 (CO), 135.9 (C-1'), 132.9 (C-4'), 129.3 (C-3', C-5'), 119.2 (C-2', C-6'), 116.0 (CN), 26.6 (C-2), 20.4 (CH$_3$). **IR (ATR)** $\tilde{v}_{max}$/cm$^{-1}$=3267, 3207, 3137, 1660, 1613, 1552, 1510, 819. **HRMS (ESI):** calcd. for C$_{10}$H$_9$N$_2$O (M-H)$^-$ 173.07204; found 173.07194. **Purity (HPLC):** >96% ($\lambda$ = 210 nm), >96% ($\lambda$ = 254 nm).

**Chemical structure 3.** 2-Cyano-*N*-phenylacetamide (2).

According to general procedure **A**, aniline (1.96 mL, 21.5 mmol, 1.0 eq.), 2-cyanoacetic acid (2.01 g, 23.6 mmol, 1.1 eq.) and DCC (4.87 g, 23.6 mmol, 1.1 eq.) in DMF (20 mL) were used to yield amide two as colorless crystals (2.60 g, 16.2 mmol, 76%). Analytical data are in accordance with

literature (*Yuan et al., 2019*). **R$_f$** = 0.12 (4:1 hexanes/acetone). **m.p.:** 202°C [(*Yuan et al., 2019*): 172°C]. **$^1$H NMR (500 MHz, (CD$_3$)$_2$SO)** δ/ppm = 10.28 (s, 1H, NH), 7.54 (dt, *J* = 8.7, 1.6 Hz, 2H, 2'-H, 6'-H), 7.39–7.29 (m, 2H, 3'-H, 5'-H), 7.15–7.04 (m, 1H, 4'-H), 3.89 (s, 2H, 2 hr). **$^{13}$C NMR (126 MHz, (CD$_3$)$_2$SO)** δ/ppm = 161.0 (CO), 138.4 (C-1'), 128.9 (C-3', C-5'), 123.9 (C-4'), 119.2 (C-2', C-6'), 115.9 (CN), 26.7 (C-2). **IR (ATR)** $\tilde{V}_{max}$/cm$^{-1}$=3265, 3207, 3143, 3099, 3052, 1653, 1620, 1557, 1299, 943, 761, 696. **HRMS (ESI):** calcd. for C$_9$H$_7$N$_2$O (M-H)⁻ 159.05639; found 159.05628. **Purity (HPLC):**>96% (λ = 210 nm),>96% (λ = 254 nm).

**Chemical structure 4.** 2-Cyano-*N*-(4-methoxyphenyl)acetamide (3).

According to general procedure **A**, *p*-anisidine (2.36 mL, 20.3 mmol, 1.0 eq.), 2-cyanoacetic acid (19.0 g, 22.3 mmol, 1.1 eq.) and DCC (4.61 g, 22.3 mmol, 1.1 eq.) in DMF (20 mL) were used to yield amide three as pale blue crystals (1.58 g, 8.31 mmol, 41%). Analytical data are in accordance with literature (*Yuan et al., 2019*). **R$_f$** = 0.10 (4:1 hexanes/acetone). **m.p.:** 137°C [(*Yuan et al., 2019*): 176°C]. **$^1$H NMR (400 MHz, (CD$_3$)$_2$SO)** δ/ppm = 10.14 (s, 1H, NH), 7.53–7.38 (m, 2H, 2'-H, 6'-H), 7.00–6.83 (m, 2H, 3'-H, 5'-H), 3.84 (s, 2H, 2 hr), 3.72 (s, 3H, OCH$_3$). **$^{13}$C NMR (101 MHz, (CD$_3$)$_2$SO)** δ/ppm = 160.4 (CO), 155.6 (C-4'), 131.4 (C-1'), 120.8 (C-2', C-6'), 116.0 (CN), 114.0 (C-3', C-5'), 55.2 (OCH$_3$), 26.5 (C-2). **IR (ATR)** $\tilde{V}_{max}$/cm$^{-1}$=3299.3150, 1655, 1608, 1557, 1511, 1251, 1032, 828. **HRMS (ESI):** calcd. for C$_{10}$H$_9$N$_2$O$_2$ (M-H)⁻ 189.06695; found 189.06688. **Purity (HPLC):**>96% (λ = 210 nm), >96% (λ = 254 nm).

**Chemical structure 5.** N-(4-Chlorophenyl)−2-cyanoacetamide (4).

According to general procedure **A**, 4-chloroaniline (682 μL, 23.0 mmol, 1.0 eq.), 2-cyanoacetic acid (2.16 g, 25.4 mmol, 1.1 eq.) and DCC (5.23 g, 25.4 mmol, 1.1 eq.) in DMF (20 mL) were used to yield amide four as colorless crystals (3.46 g, 17.8 mmol, 77%). Analytical data are in accordance with literature (*Yuan et al., 2019*). **R$_f$** = 0.14 (3:2 hexanes/acetone). **m.p.:** 207°C [(*Yuan et al., 2019*): 179°C]. **$^1$H NMR (500 MHz, (CD$_3$)$_2$SO)** δ/ppm = 10.42 (s, 1H, NH), 7.62–7.51 (m, 2H, 2'-H, 6'-H), 7.50–7.27 (m, 2H, 3'-H, 5'-H), 3.91 (s, 2H, 2 hr). **$^{13}$C NMR (126 MHz, (CD$_3$)$_2$SO)** δ/ppm = 161.2 (CO), 137.3 (C-1'), 128.8 (C-3', C-5'), 127.5 (C-4'), 120.8 (C-2', C-6'), 115.8 (CN), 26.8 (C-2). **IR (ATR)** $\tilde{V}_{max}$/cm$^{-1}$=3264, 3200, 3132, 3083, 1664, 1610, 1548, 1491, 832. **HRMS (ESI):** calcd. for C$_9$H$_6$$^{35}$ClN$_2$O (M-H)⁻ 193.01741; found 193.01750. **Purity (HPLC):**>96% (λ = 210 nm),>96% (λ = 254 nm).

**Chemical structure 6.** N-(4-Bromophenyl)−2-cyanoacetamide (5).

According to general procedure **A**, 4-bromoaniline (2.00 mL, 17.1 mmol, 1.0 eq.), 2-cyanoacetic acid (1.60 g, 18.8 mmol, 1.1 eq.) and DCC (3.88 g, 18.8 mmol, 1.1 eq.) in DMF (20 mL) were used to

yield amide five as colorless crystals (1.62 g, 6.76 mmol, 40%). Analytical data are in accordance with literature (*Yuan et al., 2019*). $R_f$ = 0.14 (4:1 hexanes/acetone). **m.p.:** 186°C [(*Yuan et al., 2019*): 185°C]. **1H NMR (400 MHz, (CD₃)₂SO)** δ/ppm = 10.42 (s, 1H, NH), 7.52 (s, 4H, 2'-H, 3'-H, 5'-H, 6'-H), 3.90 (s, 2H, 2 hr). **13C NMR (101 MHz, (CD₃)₂SO)** δ/ppm = 161.2 (CO), 137.7 (C-1'), 131.7 (C-2', C-6' or C-3', C-5'), 121.2 (C-2', C-6' or C-3', C-5'), 115.8 (C-4'), 115.5 (CN), 26.8 (C-2). **IR (ATR)** $\tilde{V}_{max}$/cm⁻¹=3322, 2927, 2849, 1608, 1547, 1245, 828. **HRMS (ESI):** calcd. for $C_9H_6{}^{79}BrN_2O$ (M-H)⁻ 236.96690; found 236.96692. **Purity (HPLC):**>96% (λ = 210 nm),>96% (λ = 254 nm).

**Chemical structure 7.** 2-Cyano-N-(4-fluorophenyl)acetamide (6).

According to general procedure **A**, 4-fluoroaniline (2.16 mL, 22.5 mmol, 1.0 eq.), 2-cyanoacetic acid (1.91 g, 22.5 mmol, 1.0 eq.) and DCC (4.64 g, 22.5 mmol, 1.0 eq.) in DMF (20 mL) were used to yield amide six as colorless crystals (3.00 g, 16.9 mmol, 75%). Analytical data are in accordance with literature (*Ammar et al., 2006*). $R_f$ = 0.11 (4:1 hexanes/acetone). **m.p.:** 179°C [(*Ammar et al., 2006*): 158–160°C]. **1H NMR (400 MHz, (CD₃)₂SO)** δ/ppm = 10.34 (s, 1H, NH), 7.66–7.46 (m, 2H, 2'-H, 6'-H), 7.27–7.09 (m, 2H, 3'-H, 5'-H), 3.89 (s, 2H, 2 hr). **13C NMR (101 MHz, (CD₃)₂SO)** δ/ppm = 161.0 (CO), 158.3 (d, $J_{CF}$ = 240.5 Hz, C-4'), 134.7 (d, $J_{CF}$ = 2.7 Hz, C-1'), 121.1 (d, $J_{CF}$ = 7.9 Hz, C-2', C-6'), 115.9 (CN), 115.5 (d, $J_{CF}$ = 22.3 Hz, C-3', C-5'), 26.6 (C-2). **IR (ATR)** $\tilde{V}_{max}$/cm⁻¹=3274, 3166, 3107, 1662, 1623, 1566, 1505, 834. **HRMS (ESI):** calcd. for $C_9H_6FN_2O$ (M-H)⁻ 177.04696; found 177.04687. **Purity (HPLC):**>96% (λ = 210 nm),>96% (λ = 254 nm).

**Chemical structure 8.** 2-Cyano-N-(4-iodophenyl)acetamide (7).

According to general procedure **A**, 4-iodoaniline (4.50 g, 20.5 mmol, 1.0 eq.), 2-cyanoacetic acid (17.5 g, 20.5 mmol, 1.0 eq.) and DCC (4.24 g, 20.5 mmol, 1.0 eq.) in DMF (20 mL) were used to yield amide seven as pale blue crystals (4.50 g, 15.7 mmol, 77%). The compound is literature known, but no analytical data are available (*Sjogren et al., 1991*). $R_f$ = 0.11 (4:1 hexanes/acetone). **m.p.:** 218°C. **1H NMR (400 MHz, (CD₃)₂SO)** δ/ppm = 10.38 (s, 1H, NH), 7.78–7.60 (m, 2H, 3'-H, 5'-H), 7.49–7.29 (m, 2H, 2'-H, 6'-H), 3.90 (s, 2H, 2 hr). **13C NMR (101 MHz, (CD₃)₂SO)** δ/ppm = 161.2 (CO), 138.2 (C-1'), 137.6 (C-3', C-5'), 121.4 (C-2', C-6'), 115.8 (CN), 87.6 (C-4'), 26.8 (C-2). **IR (ATR)** $\tilde{V}_{max}$/cm⁻¹=3265, 3188, 3113, 3078, 1666, 1543, 1391, 1299, 823. **HRMS (ESI):** calcd. for $C_9H_6IN_2O$ (M-H)⁻ 284.95303; found 284.95302. **Purity (HPLC):**>96% (λ = 210 nm),>96% (λ = 254 nm).

**Chemical structure 9.** 2-Cyano-N-(4-nitrophenyl)acetamide (8).

According to general procedure **A**, 4-nitroaniline (696 µL, 7.24 mmol, 1.0 eq.), 2-cyanoacetic acid (616 mg, 7.24 mmol, 1.0 eq.) and DCC (1.49 g, 7.24 mmol, 1.0 eq.) in DMF (20 mL) were used to

yield amide eight as yellow solid (918 mg, 4.47 mmol, 62%). Analytical data are in accordance with literature (*Sohn et al., 2017*). $R_f$ = 0.11 (4:1 hexanes/acetone). **m.p.:** 218°C [(*Sohn et al., 2017*): 220°C]. **$^1$H NMR (400 MHz, (CD$_3$)$_2$SO)** $\delta$/ppm = 10.88 (s, 1H, NH), 8.35–8.15 (m, 2H, 3'-H, 5'-H), 7.93–7.71 (m, 2H, 2'-H, 6'-H), 4.00 (s, 2H, 2 hr). **$^{13}$C NMR (101 MHz, (CD$_3$)$_2$SO)** $\delta$/ppm = 162.2 (CO), 144.4 (C-1'), 142.7 (C-4'), 125.1 (C-3', C-5'), 119.0 (C-2', C-6'), 115.5 (CN), 27.2 (C-2). **IR (ATR)** $\tilde{V}_{max}$/cm$^{-1}$=3287, 1673, 1562, 1503, 1336, 1259, 860, 748. **HRMS (ESI):** calcd. for $C_9H_6N_3O_3$ (M-H)$^-$ 204.04146; found 204.04146. **Purity (HPLC):**>96% ($\lambda$ = 210 nm),>96% ($\lambda$ = 254 nm).

**Chemical structure 10.** 2-Cyano-*N*-(4-cyanophenyl)acetamide (9).

According to general procedure **A**, 4-aminobenzonitrile (1.00 g, 8.46 mmol, 1.0 eq.), 2-cyanoacetic acid (7.20 g, 8.46 mmol, 1.0 eq.) and DCC (1.75 g, 8.46 mmol, 1.0 eq.) in DMF (10 mL) were used to yield amide nine as yellow solid (1.22 g, 6.57 mmol, 78%). The compound is literature known, but no analytical data are available (*Sjogren et al., 1991*). $R_f$ = 0.08 (4:1 hexanes/acetone). **m.p.:** 201°C. **$^1$H NMR (500 MHz, (CD$_3$)$_2$SO)** $\delta$/ppm = 10.72 (s, 1H, NH), 7.82–7.78 (m, 2H, 3'-H, 5'-H), 7.74–7.70 (m, 2H, 2'-H, 6'-H), 3.97 (s, 2H, 2 hr). **$^{13}$C NMR (126 MHz, (CD$_3$)$_2$SO)** $\delta$/ppm = 162.0 (CO), 142.5 (C-1'), 133.5 (C-3', C-5'), 119.3 (C-2', C-6'), 118.9 (C-4'), 115.6 (CH$_2$CN), 105.7 (CN), 27.1 (C-2). **IR (ATR)** $\tilde{V}_{max}$/cm$^{-1}$=3268, 3194, 3118, 2229, 1599, 1538, 1504, 845. **HRMS (ESI):** calcd. for $C_{10}H_6N_3O$ (M-H)$^-$ 184.05164; found 184.05161. **Purity (HPLC):**>96% ($\lambda$ = 210 nm),>96% ($\lambda$ = 254 nm).

**Chemical structure 11.** N-(4-Acetylphenyl)−2-cyanoacetamide (10).

According to general procedure **A**, 4-aminoacetophenone (1.30 mL, 7.40 mmol, 1.0 eq.), 2-cyanoacetic acid (629 mg, 7.40 mmol, 1.0 eq.) and DCC (1.53 g, 7.40 mmol, 1.0 eq.) in DMF (10 mL) were used to yield amide **10** as yellow crystals (802 mg, 3.97 mmol, 54%). Analytical data are in accordance with literature (*Metwally et al., 2017*). $R_f$ = 0.37 (3:2 hexanes/acetone). **m.p.:** 194°C [(*Metwally et al., 2017*): 225°C]. **$^1$H NMR (500 MHz, (CD$_3$)$_2$SO)** $\delta$/ppm = 10.63 (s, 1H, NH), 7.99–7.91 (m, 2H, 3'-H, 5'-H), 7.74–7.62 (m, 2H, 2'-H, 6'-H), 3.96 (s, 2H, 2 hr), 2.53 (s, 3H, CH$_3$). **$^{13}$C NMR (101 MHz, (CD$_3$)$_2$SO)** $\delta$/ppm = 196.5 ($\underline{C}$OCH$_3$), 161.7 (HNCO), 142.6 (C-1'), 132.2 (C-4'), 129.6 (C-3', C-5'), 118.5 (C-2', C-6'), 115.7 (CN), 27.0 (C-2), 26.5 (CH$_3$). **IR (ATR)** $\tilde{V}_{max}$/cm$^{-1}$=3286, 2250, 1695, 1651, 1599, 1536, 1279, 1249, 833, 720. **HRMS (ESI):** calcd. for $C_{11}H_9N_2O_2$ (M-H)$^-$ 201.06695; found 201.06694. **Purity (HPLC):**>96% ($\lambda$ = 210 nm),>96% ($\lambda$ = 254 nm).

**Chemical structure 12.** 2-Cyano-*N*-(4-propoxyphenyl)acetamide (11).

According to general procedure **A**, 4-propoxyaniline (679 µL, 4.49 mmol, 1.0 eq.), 2-cyanoacetic acid (382 mg, 4.49 mmol, 1.0 eq.) and DCC (927 mg, 4.49 mmol, 1.0 eq.) in DMF (2.0 mL) were used to yield amide **11** as colorless crystals (604 mg, 2.77 mmol, 62%). $R_f$ = 0.24 (95:5 $CH_2Cl_2$/EtOH). **m. p.:** 183°C. **$^1$H NMR (400 MHz, $(CD_3)_2SO$)** δ/ppm = 10.13 (s, 1H, NH), 7.51–7.38 (m, 2H, 2'-H, 6'-H), 6.98–6.83 (m, 2H, 3'-H, 5'-H), 3.88 (t, $J$ = 7.1 Hz, 2H, C$\underline{H}_2$CH$_2$CH$_3$), 3.84 (s, 2H, 2 hr), 1.70 (sext, $J$ = 7.1 Hz, 2H, CH$_2$C$\underline{H}_2$CH$_3$), 0.96 (t, $J$ = 7.1 Hz, 3H, CH$_2$CH$_2$C$\underline{H}_3$). **$^{13}$C NMR (101 MHz, $(CD_3)_2SO$)** δ/ppm = 160.4 (CO), 155.1 (C-4'), 131.4 (C-1'), 120.8 (C-2', C-6'), 116.0 (CN), 114.5 (C-3', C-5'), 69.0 ($\underline{C}H_2CH_2CH_3$), 26.5 (C-2), 22.0 (CH$_2$$\underline{C}$H$_2$CH$_3$), 10.4 (CH$_2$CH$_2$$\underline{C}$H$_3$). IR (ATR) $\tilde{V}_{max}$/cm$^{-1}$=3283, 3096, 1607, 1559, 1508, 1239, 828, 570. **HRMS (ESI):** calcd. for $C_{12}H_{13}N_2O_2$ (M-H)$^-$ 217.09825; found 217.09832. **Purity (HPLC):**>96% (λ = 210 nm),>96% (λ = 254 nm).

**Chemical structure 13.** 2-Cyano-*N*-(4-(trifluoromethoxy)phenyl)acetamide (12).

According to general procedure **A**, 4-(trifluoromethoxy)aniline (939 µL, 7.00 mmol, 1.0 eq.), 2-cyanoacetic acid (595 mg, 7.00 mmol, 1.0 eq.) and DCC (1.44 g, 7.00 mmol, 1.0 eq.) in DMF (10 mL) were used to yield amide **12** as colorless solid (1.23 g, 5.03 mmol, 72%). The compound is literature known, but no analytical data are available (*Sjogren et al., 1991*). $R_f$ = 0.39 (95:5 $CH_2Cl_2$/EtOH). **m. p.:** 154°C. **$^1$H NMR (400 MHz, $(CD_3)_2SO$)** δ/ppm = 10.49 (s, 1H, NH), 7.74–7.55 (m, 2H, 2'-H, 6'-H), 7.47–7.26 (m, 2H (3'-H, 5'-H), 3.92 (s, 2H, 2 hr). **$^{13}$C NMR (101 MHz, $(CD_3)_2SO$)** δ/ppm = 161.3 (CO), 144.0 (C-4'), 137.5 (C-1'), 121.8 (C-3', C-5'), 120.7 (C-2', C-6'), 120.1 (q, $J_{CF}$ = 255.7 Hz, OCF$_3$), 115.8 (CN), 26.8 (C-2). IR (ATR) $\tilde{V}_{max}$/cm$^{-1}$=3278, 2975, 1667, 1616, 1557, 1508, 1277, 1205, 1171. **HRMS (ESI):** calcd. for $C_{10}H_6F_3N_2O_2$ (M-H)$^-$ 243.03869; found 243.03869. **Purity (HPLC):**>96% (λ = 210 nm),>96% (λ = 254 nm).

**Chemical structure 14.** Methyl 4-(2-cyanoacetamido)benzoate (13).

According to general procedure **A**, methyl 4-aminobenzoate (3.00 g, 19.5 mmol, 1.0 eq.), 2-cyanoacetic acid (1.66 g, 19.5 mmol, 1.0 eq.) and DCC (4.01 g, 19.5 mmol, 1.0 eq.) in DMF (15 mL) were used to yield amide **13** as colorless solid (2.76 g, 12.6 mmol, 65%). The compound is literature known, but no analytical data are available (*Sjogren et al., 1991*). $R_f$ = 0.44 (3:2 hexanes/acetone). **m.p.:** 162°C. **$^1$H NMR (500 MHz, $(CD_3)_2SO$)** δ/ppm = 10.63 (s, 1H, NH), 8.01–7.87 (m, 2H, 2 hr, 6 hr), 7.76–7.61 (m, 2H, 3 hr, 5 hr), 3.96 (s, 2H, 2 hr), 3.82 (s, 3H, CH$_3$). **$^{13}$C NMR (126 MHz, $(CD_3)_2SO$)** δ/ppm = 165.7 ($\underline{C}$OOCH$_3$), 161.7 (HNCO), 142.7 (C-4), 130.4 (C-2, C-6), 124.6 (C-1), 118.6 (C-3, C-5), 115.7 (CN), 52.0 (CH$_3$), 27.0 (C-2). IR (ATR) $\tilde{V}_{max}$/cm$^{-1}$=2809, 1722, 1608, 1558, 1507, 1431, 1274, 1110, 757. **HRMS (ESI):** calcd. for $C_{11}H_9N_2O_3$ (M-H)$^-$ 217.06187; found 217.06187. **Purity (HPLC):** >96% (λ = 210 nm),>96% (λ = 254 nm).

**Chemical structure 15.** N-(2-Bromo-4-chlorophenyl)—2-cyanoacetamide (14).

According to general procedure **A**, 2-bromo-4-chloroaniline (1.00 g, 4.84 mmol, 1.0 eq.), 2-cyano-acetic acid (412 mg, 4.84 mmol, 1.0 eq.) and DCC (999 mg, 4.84 mmol, 1.0 eq.) in DMF (10 mL) were used to yield amide **14** as colorless crystals (919 mg, 3.36 mmol, 69%). $R_f$ = 0.22 (3:2 hexanes/acetone). **m.p.:** 157˚C. **$^1$H NMR (400 MHz, (CD$_3$)$_2$SO)** δ/ppm = 9.97 (s, 1H, NH), 7.83 (d, J = 2.4 Hz, 1H, 3'-H), 7.63 (d, J = 8.7 Hz, 1H, 6'-H), 7.49 (dd, J = 8.7, 2.4 Hz, 1H, 5'-H), 3.98 (s, 2H, 2 hr). **$^{13}$C NMR (101 MHz, (CD$_3$)$_2$SO)** δ/ppm = 161.8 (CO), 134.7 (C-1'), 132.0 (C-3'), 130.7 (C-4'), 128.2 (C-5'), 128.0 (C-6'), 118.5 (C-2'), 115.7 (CN), 26.1 (C-2). **IR (ATR)** $\tilde{V}_{max}$/cm$^{-1}$=3281, 2258, 1666, 1577, 1530, 1470, 1284, 822. **HRMS (ESI):** calcd. for C$_9$H$_5$$^{79}$Br$^{35}$ClN$_2$O (M-H)$^-$ 270.92793; found 270.92809. **Purity (HPLC):** >93% (λ = 210 nm), >93% (λ = 254 nm).

**Chemical structure 16.** 2-Cyano-N-(2-iodophenyl)acetamide (15).

According to general procedure **A**, 2-iodoaniline (1.00 g, 4.57 mmol, 1.0 eq.), 2-cyanoacetic acid (388 mg, 4.57 mmol, 1.0 eq.) and DCC (942 mg, 4.57 mmol, 1.0 eq.) in DMF (10 mL) were used to yield amide **15** as brown crystals (913 g, 3.19 mmol, 70%). $R_f$ = 0.16 (4:1 hexanes/acetone). **m.p.:** 161˚C. **$^1$H NMR (400 MHz, (CD$_3$)$_2$SO)** δ/ppm = 9.88 (s, 1H, NH), 7.90 (d, J = 7.6 Hz, 1H, 3'-H), 7.46–7.38 (m, 2H, 5'-H, 6'-H), 7.03 (ddd, J = 8.6, 5.4, 3.6 Hz, 1H, 4'-H), 3.93 (s, 2H, 2 hr). **$^{13}$C NMR (101 MHz, (CD$_3$)$_2$SO)** δ/ppm = 161.4 (CO), 139.1 (C-3'), 138.7 (C-1'), 128.8 (C-5'), 128.3 (C-4'), 127.4 (C-6'), 115.8 (CN), 96.4 (C-2'), 26.0 (C-2). **IR (ATR)** $\tilde{V}_{max}$/cm$^{-1}$=3252, 2263, 1658, 1577, 1542, 1433, 1015, 758, 768. **HRMS (ESI):** calcd. for C$_9$H$_6$IN$_2$O (M-H)$^-$ 284.95303; found 284.95295. **Purity (HPLC):** >96% (λ = 210 nm), >96% (λ = 254 nm).

**Chemical structure 17.** N-(3-Chloro-2,4-difluorophenyl)—2-cyanoacetamide (16).

According to general procedure **A**, 3-chloro-2,4-difluoroaniline (1.06 g, 6.48 mmol, 1.0 eq.), 2-cyanoacetic acid (551 mg, 6.48 mmol, 1.0 eq.) and DCC (1.34 g, 6.48 mmol, 1.0 eq.) in DMF (10 mL) were used to yield amide **16** as colorless crystals (1.01 g, 4.39 mmol, 68%). $R_f$ = 0.16 (4:1 hexanes/acetone). **m.p.:** 144˚C. **$^1$H NMR (400 MHz, (CD$_3$)$_2$SO)** δ/ppm = 10.32 (s, 1H, NH), 7.79 (td, J = 9.0, 5.8 Hz, 1H, 6'-H), 7.33 (td, J = 9.0, 2.1 Hz, 1H, 5'-H), 3.99 (s, 2H, 2 hr). **$^{13}$C NMR (101 MHz, (CD$_3$)$_2$SO)** δ/ppm = 162.0 (CO), 154.8 (dd, $J_{CF}$ = 246.5, 2.0 Hz, C-2' or C-4'), 150.4 (dd, $J_{CF}$ = 250.4, 3.3 Hz, C-2' or C-4'), 123.4 (dd, $J_{CF}$ = 8.8, 2.4 Hz, C-6'), 123.0 (dd, $J_{CF}$ = 11.8, 3.5 Hz, C-1'), 115.7 (CN), 111.9 (dd, $J_{CF}$ = 21.4, 3.8 Hz, C-5'), 108.7 (dd, $J_{CF}$ = 21.9, 19.7 Hz, C-3'), 26.3 (C-2). **IR (ATR)** $\tilde{V}_{max}$/cm$^{-1}$=3274.2935, 2264, 1681, 1551, 1488, 1443, 1012, 831, 628. **HRMS (ESI):** calcd. for

$C_9H_4{}^{35}ClF_2N_2O$ (M-H)$^-$ 228.99857; found 228.99850. **Purity (HPLC):**>96% ($\lambda$ = 210 nm),>96% ($\lambda$ = 254 nm).

**Chemical structure 18.** 2-Cyano-*N*-(3,4-dimethoxyphenyl)acetamide (17).

According to general procedure **A**, 3,4-dimethoxyaniline (1.00 g, 6.53 mmol, 1.0 eq.), 2-cyano-acetic acid (555 mg, 6.53 mmol, 1.0 eq.) and DCC (1.35 g, 6.53 mmol, 1.0 eq.) in DMF (10 mL) were used to yield amide **17** as violet crystals (1.10 g, 4.99 mmol, 76%). The compound is literature known, but no analytical data are available (*Edraki et al., 2016*). $R_f$ = 0.05 (4:1 hexanes/acetone). **m. p.:** 174°C. **¹H NMR (500 MHz, (CD₃)₂SO)** $\delta$/ppm = 10.15 (s, 1H, NH), 7.21 (d, *J* = 2.4 Hz, 1H, 2'-H), 7.04 (dd, *J* = 8.7, 2.4 Hz, 1H, 6'-H), 6.90 (d, *J* = 8.7 Hz, 1H, 5'-H), 3.84 (s, 2H, 2 hr), 3.79–3.66 (m, 6H, 2x OCH₃). **¹³C NMR (126 MHz, (CD₃)₂SO)** $\delta$/ppm = 160.5 (CO), 148.6 (C-3'), 145.3 (C-4'), 131.9 (C-1'), 116.0 (CN), 112.0 (C-5'), 111.2 (C-6'), 104.3 (C-2'), 55.7 (OCH₃), 55.4 (OCH₃), 26.6 (C-2). IR (ATR) $\tilde{V}_{max}$/cm$^{-1}$=3273, 2914, 2256, 1660, 1513, 1239, 1132, 1020, 837. **HRMS (ESI):** calcd. for $C_{11}H_{11}N_2O_3$ (M-H)$^-$ 219.07752; found 219.07751. **Purity (HPLC):** >96% ($\lambda$ = 210 nm), >96% ($\lambda$ = 254 nm).

**Chemical structure 19.** 2-Cyano-*N*-(2,3-dichlorophenyl)acetamide (18).

According to general procedure **A**, 2,3-dichloroaniline (745 μL, 6.30 mmol, 1.0 eq.), 2-cyanoacetic acid (536 mg, 6.30 mmol, 1.0 eq.) and DCC (1.30 g, 6.30 mmol, 1.0 eq.) in DMF (10 mL) were used to yield amide **18** as colorless crystals (414 mg, 1.81 mmol, 29%). $R_f$ = 0.21 (4:1 hexanes/acetone). **m.p.:** 176°C. **¹H NMR (500 MHz, (CD₃)₂SO)** $\delta$/ppm = 10.11 (s, 1H, NH), 7.69 (dd, *J* = 8.1, 1.3 Hz, 1H, 4'-H or 6'-H), 7.51 (dd, *J* = 8.1, 1.3 Hz, 1H, 4'-H or 6'-H), 7.38 (t, *J* = 8.1 Hz, 1H, 5'-H), 4.02 (s, 2H, 2 hr). **¹³C NMR (126 MHz, (CD₃)₂SO)** $\delta$/ppm = 161.9 (CO), 136.1 (C-1' or C-3'), 132.0 (C-1' or C-3'), 128.2 (C-5'), 127.3 (C-4' or C-6'), 125.3 (C-2'), 124.8 (C-4' or C-6'), 115.7 (CN), 26.3 (C-2). IR (ATR) $\tilde{V}_{max}$/cm$^{-1}$=3287, 2253, 1666, 1580, 1527, 1415, 1338, 1182, 953, 788. **HRMS (ESI):** calcd. for $C_9H_5{}^{35}Cl_2N_2O$ (M-H)$^-$ 226.97844; found 226.97859. **Purity (HPLC):** >96% ($\lambda$ = 210 nm), >96% ($\lambda$ = 254 nm).

**Chemical structure 20.** 2-Cyano-*N*-(2,6-dibromophenyl)acetamide (19).

According to general procedure **A**, 2,6-dibromoaniline (1.00 g, 4.00 mmol, 1.0 eq.), 2-cyanoacetic acid (340 mg, 4.00 mmol, 1.0 eq.) and DCC (825 mg, 4.00 mmol, 1.0 eq.) in DMF (10 mL) were used to yield amide **19** as colorless crystals (514 mg, 1.62 mmol, 40%). $R_f$ = 0.48 (3:2 hexanes/acetone). **m.p.:** 187°C. **¹H NMR (500 MHz, (CD₃)₂SO)** $\delta$/ppm = 10.37 (s, 1H, NH), 7.74 (d, *J* = 8.1 Hz, 2H, 3'-H, 5'-H), 7.22 (t, *J* = 8.1 Hz, 1H, 4'-H), 3.93 (s, 2H, 2 hr). **¹³C NMR (126 MHz, (CD₃)₂SO)** $\delta$/ppm = 161.1

(CO), 134.7 (C-1'), 132.4 (C-3', C-5'), 130.7 (C-4'), 123.9 (C-2', C-6'), 115.6 (CN), 25.4 (C-2). IR (ATR) $\tilde{V}_{max}/cm^{-1}$=3326, 2926, 2851, 1626, 1568, 1539, 1242, 642. **HRMS (ESI):** calcd. for $C_9H_5{}^{79}Br_2N_2O$ (M-H)$^-$ 314.87741; found 314.87761. **Purity (HPLC):**>96% ($\lambda$ = 210 nm),>96% ($\lambda$ = 254 nm).

**Chemical structure 21.** N-(3,5-Bis(trifluoromethyl)phenyl)−2-cyanoacetamide (20).

According to general procedure **A**, 3,5-bis(trifluoromethyl)aniline (1.09 mL, 7.00 mmol, 1.0 eq.), 2-cyanoacetic acid (595 mg, 7.00 mmol, 1.0 eq.) and DCC (1.44 g, 7.00 mmol, 1.0 eq.) in DMF (10 mL) were used to yield amide **20** as colorless crystals (1.68 g, 5.67 mmol, 81%). The compound is literature known, but no analytical data are available (*Shah et al., 2018*). **R$_f$** = 0.43 (95:5 CH$_2$Cl$_2$/EtOH). **m.p.:** 141℃. **$^1$H NMR (500 MHz, (CD$_3$)$_2$SO)** $\delta$/ppm = 10.96 (s, 1H, NH), 8.28–8.12 (m, 2H, 2'-H, 6'-H), 7.90–7.77 (m, 1H, 4'-H), 4.00 (s, 2H, 2 hr). **$^{13}$C NMR (126 MHz, (CD$_3$)$_2$SO)** $\delta$/ppm = 162.4 (CO), 140.2 (C-1'), 130.9 (q, $J_{CF}$ = 32.9 Hz, C-3', C-5'), 123.1 (q, $J_{CF}$ = 272.7 Hz, CF$_3$), 119.2–118.7 (m, C-2', C-6'), 116.9–116.5 (m, C-4'), 115.4 (CN), 27.1 (C-2). IR (ATR) $\tilde{V}_{max}/cm^{-1}$=3313, 1695, 1572, 1471, 1381, 1272, 1132, 889, 703, 681. **HRMS (ESI):** calcd. for $C_{11}H_5F_6N_2O$ (M-H)$^-$ 295.03116; found 295.03127. **Purity (HPLC):** >96% ($\lambda$ = 210 nm), >96% ($\lambda$ = 254 nm).

**Chemical structure 22.** 2-Cyano-N-methyl-N-(4-(trifluoromethyl)phenyl)acetamide (21).

According to general procedure **A**, N-methyl-4-(trifluoromethyl)aniline (403 µL, 2.85 mmol, 1.0 eq.), 2-cyanoacetic acid (243 mg, 2.85 mmol, 1.0 eq.) and DCC (589 mg, 2.85 mmol, 1.0 eq.) in DMF (10 mL) were used to yield amide **21** as colorless solid (513 mg, 2.12 mmol, 74%). Analytical data are in accordance with literature (*Kobayashi and Harayama, 2009*). **R$_f$** = 0.58 (95:5 CH$_2$Cl$_2$/EtOH). **m.p.:** 70℃ [(*Kobayashi and Harayama, 2009*): 66–68℃]. **$^1$H NMR (400 MHz, (CD$_3$)$_2$SO)** $\delta$/ppm = 7.77 (d, $J$ = 8.6 Hz, 2H, 3'-H, 5'-H), 7.40 (d, $J$ = 8.2 Hz, 2H, 2'-H, 6'-H), 3.34 (s, 3H, CH$_3$), 3.23 (s, 2H, 2 hr). **$^{13}$C NMR (101 MHz, (CD$_3$)$_2$SO)** $\delta$/ppm = 161.4 (CO), 145.6 (C-1'), 131.7 (C-4'), 127.9 (C-2', C-6' or C-3', C-5'), 127.8 (C-2', C-6' or C-3', C-5'), 123.5 (q, $J$ = 271.4 Hz, CF$_3$), 113.7 (CN), 38.1 (CH$_3$), 25.6 (C-2). IR (ATR) $\tilde{V}_{max}/cm^{-1}$=3152, 2355, 1657, 1611, 1322, 1122, 1103, 1065, 848. **HRMS (ESI):** calcd. for $C_{11}H_8F_3N_2O$ (M-H)$^-$ 241.05942; found 241.05939. **Purity (HPLC):**>96% ($\lambda$ = 210 nm),>96% ($\lambda$ = 254 nm).

**Chemical structure 23.** 2-Cyano-N-(4-(trifluoromethyl)benzyl)acetamide (22).

According to general procedure **A**, 4-(trifluoromethyl)benzylamine (2.50 mL, 17.5 mmol, 1.0 eq.), 2-cyanoacetic acid (1.49 g, 17.5 mmol, 1.0 eq.) and DCC (3.62 g, 17.5 mmol, 1.0 eq.) in DMF (15 mL) were used to yield amide **22** as colorless solid (299 mg, 1.23 mmol, 7%). Analytical data are in accordance with literature (*Guo et al., 2011*). **R$_f$** = 0.46 (3:2 hexanes/acetone). **m.p.:** 113℃ [(*Guo et al.,*

*2011*): 128°C]. **¹H NMR (500 MHz, (CD₃)₂SO)** δ/ppm = 8.83 (t, *J* = 5.7 Hz, 1H, NH), 7.74–7.66 (m, 2H, 3'-H, 5'-H), 7.54–7.44 (m, 2H, 2'-H, 6'-H), 4.38 (d, *J* = 5.7 Hz, 2H, CH₂), 3.73 (s, 2H, 2 hr). **¹³C NMR (126 MHz, (CD₃)₂SO)** δ/ppm = 162.5 (CO), 143.6 (C-1'), 128.0 (C-2', C-6'), 127.7 (q, *J*$_{CF}$ = 31.7 Hz, C-4'), 125.2 (q, *J*$_{CF}$ = 3.7 Hz, C-3', C-5'). 124.3 (q, *J*$_{CF}$ = 272.1 Hz, CF₃), 116.1 (CN), 42.2 (CH₂), 25.3 (C-2). **IR (ATR)** $\tilde{V}_{max}$/cm⁻¹=3316, 2937, 2364, 1734, 1664, 1547, 1325, 1152, 1107, 1066. **HRMS (ESI):** calcd. for C₁₁H₈F₃N₂O (M-H)⁻ 241.05942; found 241.05949. **Purity (HPLC):** >96% (λ = 210 nm), >96% (λ = 254 nm).

**TPC2-A1-N**

**Chemical structure 24.** 2-Cyano-3-(3,5-dichlorophenyl)−3-hydroxy-*N*-(4-(trifluoromethyl)phenyl)acrylamide – TPC2-A1-N.

According to general procedure **B**, amide **SGA-34** (228 mg, 1.00 mmol, 1.0 eq.) in dry THF (10 mL), NaH (92.0 mg, 2.30 mmol, 2.3 eq.) and 3,5-dichlorobenzoyl chloride (230 mg, 1.10 mmol, 1.1 eq.) were used to yield **TPC2-A1-N** as colorless crystals (276 mg, 0.688 mmol, 69%). **R$_f$** = 0.62 (4:1 hexanes/acetone). **m.p.:** 202°C [(*Sjogren et al., 1991*): 208–210°C]. **¹H NMR (500 MHz, (CD₃)₂SO)** δ/ppm = 12.36 (s, 1H, NH), 7.79–7.76 (m, 2H, 2'-H, 6'-H), 7.65 (t, *J* = 1.9 Hz, 1H, 4''-H), 7.62–7.60 (m, 2H, 3'-H, 5'-H), 7.59 (d, *J* = 1.9 Hz, 2H, 2''-H, 6''-H). **¹³C NMR (126 MHz, (CD₃)₂SO)** δ/ppm = 182.7 (C-3), 166.5 (C-1), 144.9 (C-1''), 143.6 (C-1'), 133.5 (C-3'', C-5''), 128.6 (C-4''), 126.2–125.9 (m, C-3', C-5' and C-2'', C-6''), 124.6 (q, *J*$_{CF}$ = 286.1 Hz, CF₃), 123.3 (C-2), 121.8 (q, *J*$_{CF}$ = 31.8 Hz, C-4'), 118.7 (C-2', C-6'), 77.7 (CN). **IR (ATR)** $\tilde{V}_{max}$/cm⁻¹=3293, 2213, 1538, 1409, 1320, 1268, 1244, 1167, 1106, 1070, 837, 810, 660, 591. **HRMS (ESI):** calcd. for C₁₇H₈³⁵Cl₂F₃N₂O₂ (M-H)⁻ 398.99204; found 398.99202. **Purity (HPLC):**>96% (λ = 210 nm),>96% (λ = 254 nm).

**SGA-10**

**Chemical structure 25.** 2-Cyano-3-hydroxy-3-phenyl-*N*-(4-(trifluoromethyl)phenyl)acrylamide – SGA-10.

According to general procedure **B**, amide **SGA-34** (228 mg, 1.00 mmol, 1.0 eq.) in dry THF (10 mL), NaH (92.0 mg, 2.30 mmol, 2.3 eq.) and benzoyl chloride (128 µL, 1.10 mmol, 1.1 eq.) were used to give **SGA-10** as colorless crystals (185 mg, 0.557 mmol, 56%). Analytical data are in accordance with literature (*Davies et al., 2009*). **R$_f$** = 0.28 (3:2 hexanes/acetone). **m.p.:** 242°C [(*Davies et al., 2009*): 245–247°C]. **¹H NMR (400 MHz, (CD₃)₂SO)** δ/ppm = 12.12 (s, 1H, NH), 7.84–7.74 (m, 2H, 2'-H, 6'-H), 7.70–7.58 (m, 4H, 3'-H, 5'-H, Ph), 7.48–7.39 (m, 3H, Ph). **¹³C NMR (101 MHz, (CD₃)₂SO)** δ/ppm = 185.7 (C-3), 167.3 (C-1), 143.1 (C-1'), 139.7 (qPh), 135.0 (C-2), 130.0 (Ph), 127.8 (Ph), 127.6 (Ph), 126.0 (q, *J*$_{CF}$ = 3.8 Hz, C-3', C-5'), 125.0 (d, *J*$_{CF}$ = 270.6 Hz, CF₃), 122.3 (d, *J*$_{CF}$ = 35.2 Hz, C-4'), 119.4 (C-2', C-6'), 77.8 (CN). **IR (ATR)** $\tilde{V}_{max}$/cm⁻¹=3283, 2216, 1592, 1550, 1309, 1109, 1067, 840, 694. **HRMS (ESI):** calcd. for C₁₇H₁₀F₃N₂O₂ (M-H)⁻ 331.06999; found 331.06985. **Purity (HPLC):**>96% (λ = 210 nm),>96% (λ = 254 nm).

**Chemical structure 26.** 2-Cyano-3-(3,5-dinitrophenyl)−3-hydroxy-*N*-(4-(trifluoromethyl)phenyl)acrylamide – SGA-11.

According to general procedure **B**, amide **SGA-34** (228 mg, 1.00 mmol, 1.0 eq.) in dry THF (10 mL), NaH (92.0 mg, 2.30 mmol, 2.3 eq.) and 3,5-dinitrobenzoyl chloride (254 mg, 1.10 mmol, 1.1 eq.) were used to give **SGA-11** as red crystals (124 mg, 0.294 mmol, 29%). $R_f$ = 0.21 (3:2 hexanes/ acetone). **m.p.:** 240°C. **$^1$H NMR (500 MHz, (CD$_3$)$_2$SO)** $\delta$/ppm = 12.32 (s, 1H, NH), 10.00 (s, 1H, OH), 8.89–8.80 (m, 3H, 2''-H, 4''-H, 6''-H), 7.84–7.75 (m, 2H, 2'-H, 6'-H), 7.67–7.58 (m, 2H, 3'-H, 5'-H). **$^{13}$C NMR (126 MHz, (CD$_3$)$_2$SO)** $\delta$/ppm = 180.5 (C-3), 166.1 (C-1), 147.6 (C-3'', C-5''), 144.4 (C-1''), 143.5 (C-1'), 127.6 (C-2'', C-6''),126.1 (q, $J_{CF}$ = 3.5 Hz, C-3', C-5'),124.2 (q, $J_{CF}$ = 271.0 Hz, CF$_3$), 123.3 (C-2), 121.8 (q, $J_{CF}$ = 31.7 Hz, C-4'), 119.0 (C-4''), 118.6 (C-2', C-6'), 77.7 (CN). IR (ATR) $\tilde{v}_{max}$/cm$^{-1}$=3262, 3093, 2223, 1539, 1342, 1317, 1115, 1067, 841, 730, 703, 687. **HRMS (ESI):** calcd. for C$_{17}$H$_8$F$_3$N$_4$O$_6$ (M-H)$^-$ 421.04014; found 421.04021. **Purity (HPLC):**>96% ($\lambda$ = 210 nm), >96% ($\lambda$ = 254 nm).

**Chemical structure 27.** 2-Cyano-3-(4-nitrophenyl)−3-hydroxy-*N*-(4-(trifluoromethyl)phenyl)acrylamide – SGA-15.

According to general procedure **B**, amide **SGA-34** (228 mg, 1.00 mmol, 1.0 eq.) in dry THF (10 mL), NaH (92.0 mg, 2.30 mmol, 2.3 eq.) and 4-nitrobenzoyl chloride (204 mg, 1.10 mmol, 1.1 eq.) were used to give **SGA-15** as yellow crystals (276 mg, 0.688 mmol, 69%). $R_f$ = 0.19 (3:2 hexanes/acetone). **m.p.:** 217°C [(*Sjogren et al., 1991*): 211–214°C]. **$^1$H NMR (500 MHz, (CD$_3$)$_2$SO)** $\delta$/ppm = 12.39 (s, 1H, NH), 8.28–8.19 (m, 2H, 3''-H, 5''-H), 7.84–7.80 (m, 2H, 2''-H, 6''-H), 7.78 (d, $J$ = 8.6 Hz, 2H, 2'-H, 6'-H), 7.60 (d, $J$ = 8.6 Hz, 2H, 3''-H, 5''-H). **$^{13}$C NMR (126 MHz, (CD$_3$)$_2$SO)** $\delta$/ppm = 184.3 (C-3), 166.4 (C-1), 148.1 (C-1''), 147.5 (C-4''), 143.7 (C-1'), 128.6 (C-2'', C-6''), 126.1 (q, $J_{CF}$ = 3.6 Hz, C-3', C-5'), 124.6 (q, $J_{CF}$ = 270.9 Hz, CF$_3$), 123.2 (C-2), 123.1 (C-3'', C-5''), 121.6 (q, $J_{CF}$ = 32.0 Hz, C-4'), 118.6 (C-2', C-6'), 77.9 (CN). IR (ATR) $\tilde{v}_{max}$/cm$^{-1}$=3307, 2219, 1551, 1320, 1111, 1069, 844, 750, 700. **HRMS (ESI):** calcd. for C$_{17}$H$_9$F$_3$N$_3$O$_4$ (M-H)$^-$ 376.05506; found 376.05509. **Purity (HPLC):**>96% ($\lambda$ = 210 nm),>96% ($\lambda$ = 254 nm).

**Chemical structure 28.** 3-(4-Chlorophenyl)−2-cyano-3-hydroxy-*N*-(4-(trifluoromethyl)phenyl)acrylamide – SGA-16.

According to general procedure **B**, amide **SGA-34** (228 mg, 1.00 mmol, 1.0 eq.) in dry THF (10 mL), NaH (92.0 mg, 2.30 mmol, 2.3 eq.) and 4-chlorobenzoyl chloride (141 µL, 1.10 mmol, 1.1 eq.) were used to give **SGA-16** as colorless crystals (222 mg, 0.605 mmol, 61%). $R_f$ = 0.32 (3:2 hexanes/ acetone). **m.p.:** 220°C [(*Sjogren et al., 1991*): 218–220°C]. **$^1$H NMR (500 MHz, (CD$_3$)$_2$SO)** $\delta$/ ppm = 12.31 (s, 1H, NH), 7.81–7.75 (m, 2H, 2'-H, 6'-H), 7.68–7.63 (m, 2H, 2''-H, 6''-H), 7.63–7.58 (m, 2H, 3'-H, 5'-H), 7.49–7.44 (m, 2H, 3''-H, 5''-H). **$^{13}$C NMR (126 MHz, (CD$_3$)$_2$SO)** $\delta$/ppm = 184.8 (C-3), 167.0 (C-1), 143.5 (C-1'), 139.6 (C-1''), 134.1 (C-4''), 129.4 (C-2'', C-6''), 127.8 (C-3'', C-5''), 126.0 (q, $J_{CF}$ = 3.6 Hz, C-3', C-5'), 124.6 (q, $J_{CF}$ = 271.0 Hz, CF$_3$), 123.0 (C-2), 121.8 (q, $J_{CF}$ = 31.0 Hz, C-4'), 118.9 (C-2', C-6'), 77.5 (CN). IR (ATR) $\tilde{V}_{max}$/cm$^{-1}$=3282.2215, 1587, 1240, 1302, 1129, 1113, 1097, 839. **HRMS (ESI):** calcd. for C$_{17}$H$_9$$^{35}$ClF$_3$N$_2$O$_2$ (M-H)$^-$ 365.03101; found 365.03108. **Purity (HPLC):** >96% ($\lambda$ = 210 nm), >96% ($\lambda$ = 254 nm).

**Chemical structure 29.** 2-Cyano-3-hydroxy-3-(2,3,4,5,6-pentafluorophenyl)-*N*-(4-(trifluoromethyl)phenyl)acrylamide – SGA-40.

According to general procedure **B**, amide **SGA-34** (228 mg, 1.00 mmol, 1.0 eq.) in dry THF (10 mL), NaH (92.0 mg, 2.30 mmol, 2.3 eq.) and 2,3,4,5,6-pentafluorobenzoyl chloride (158 µL, 1.10 mmol, 1.1 eq.) were used to give **SGA-40** as colorless crystals (236 mg, 0.560 mmol, 56%). $R_f$ = 0.28 (3:2 hexanes/acetone). **m.p.:** 161°C. **$^1$H NMR (400 MHz, (CD$_3$)$_2$SO)** $\delta$/ppm = 11.70 (s, 1H, NH), 7.81– 7.72 (m, 2H, 2'-H, 6'-H), 7.67–7.56 (m, 2H, 3'-H, 5'-H). **$^{13}$C NMR (101 MHz, (CD$_3$)$_2$SO)** $\delta$/ ppm = 172.6 (C-3), 164.9 (C-1), 143.9–140.9 (m, C-2'', C-6'' or C-3'', C-5''), 143.3 (C-1'), 141.9–138.9 (m, C-4''), 138.5–135.1 (m, C-2'', C-6'' or C-3'', C-5'), 126.1 (q, $J_{CF}$ = 3.6 Hz, C-3', C-5'), 124.6 (q, $J_{CF}$ = 271.0 Hz, CF$_3$), 122.0 (q, $J_{CF}$ = 31.9 Hz, C-4'), 117.5–117.0 (m, C-1''), 121.7 (C-2', C-6'), 118.7 (C-2), 81.8 (CN). IR (ATR) $\tilde{V}_{max}$/cm$^{-1}$=2230, 1590, 1543, 1524, 1497, 1323, 1116, 1000, 839. **HRMS (ESI):** calcd. for C$_{17}$H$_5$F$_8$N$_2$O$_2$ (M-H)$^-$ 421.02288; found 421.02337. **Purity (HPLC):**>96% ($\lambda$ = 210 nm),>96% ($\lambda$ = 254 nm).

**Chemical structure 30.** 2-Cyano-3-(3,5-dibromophenyl)-3-hydroxy-*N*-(4-(trifluoromethyl)phenyl)acrylamide – SGA-70.

According to general procedure **B**, amide **SGA-34** (228 mg, 1.00 mmol, 1.0 eq.) in dry THF (10 mL), NaH (92.0 mg, 2.30 mmol, 2.3 eq.) and 3,5-dibromobenzoic acid (308 mg, 1.10 mmol, 1.1 eq.; converted into the corresponding aryl chloride) were used to give **SGA-70** as light yellow crystals (206 mg, 0.420 mmol, 42%). $R_f$ = 0.32 (3:2 hexanes/acetone). **m.p.:** 211°C. **$^1$H NMR (400 MHz, (CD$_3$)$_2$SO)** $\delta$/ppm = 12.32 (s, 1H, NH), 8.96 (s, 1H, OH), 7.91–7.81 (m, 1H, 4''-H), 7.78–7.72 (m, 4H, 2'-H, 6'-H, 2''-H, 6''-H), 7.65–7.55 (m, 2H, 3'-H, 5'-H). **$^{13}$C NMR (101 MHz, (CD$_3$)$_2$SO)** $\delta$/ppm = 182.5 (C-3), 166.4 (C-1), 145.5 (C-1''), 143.6 (C-1'), 133.7 (C-4''), 129.2 (C-2'', C-6''), 126.0 (q, $J_{CF}$ = 3.5 Hz, C-3', C-5'), 123.9 (q, $J_{CF}$ = 271.2 Hz, CF$_3$), 123.3 (C-2), 121.8 (C-3'', C-5''), 121.6 (q, $J_{CF}$ = 31.9 Hz, C-4'), 118.5 (C-2', C-6'), 77.4 (CN). IR (ATR) $\tilde{V}_{max}$/cm$^{-1}$=2218, 1594, 1538, 1315, 1166, 1109, 1068, 838, 750. **HRMS (ESI):** calcd. for C$_{17}$H$_8$$^{79}$Br$_2$F$_3$N$_2$O$_2$ (M-H)$^-$ 486.89101; found 486.89128. **Purity (HPLC):**>96% ($\lambda$ = 210 nm),>96% ($\lambda$ = 254 nm).

**Chemical structure 31.** 2-Cyano-3-hydroxy-3-(2,4,6-trichlorophenyl)-*N*-(4-(trifluoromethyl)phenyl)acrylamide – SGA-71.

According to general procedure **B**, amide **SGA-34** (228 mg, 1.00 mmol, 1.0 eq.) in dry THF (10 mL), NaH (92.0 mg, 2.30 mmol, 2.3 eq.) and 2,4,6-trichlorobenzoyl chloride (172 µL, 1.10 mmol, 1.1 eq.) were used to give **SGA-71** as colorless crystals (158 mg, 0.363 mmol, 36%). $R_f$ = 0.14 (3:2 hexanes/acetone). **m.p.:** 220°C. **$^1$H NMR (500 MHz, (CD$_3$)$_2$SO)** $\delta$/ppm = 11.92 (s, 1H, NH), 7.79–7.74 (m, 2H, 2'-H, 6'-H), 7.66 (s, 2H, 3''-H, 5''-H), 7.63–7.58 (m, 2H, 3'-H, 5'-H). **$^{13}$C NMR (126 MHz, (CD$_3$)$_2$SO)** $\delta$/ppm = 180.7 (C-3), 165.7 (C-1), 143.6 (C-1'), 139.5 (C-1''), 133.1 (C-4''), 132.0 (C-2'', C-6''), 127.8 (C-3'', C-5''), 126.1 (q, $J_{CF}$ = 3.6 Hz, C-3', C-5'), 124.6 (q, $J_{CF}$ = 271.1 Hz, CF$_3$), 121.9 (C-2), 121.6 (q, $J_{CF}$ = 31.9 Hz, C-4'), 118.4 (C-2', C-6'), 79.5 (CN). IR (ATR) $\tilde{V}_{max}$/cm$^{-1}$=2230, 1598, 1541, 1318, 1116, 841. **HRMS (ESI):** calcd. for C$_{17}$H$_7$$^{35}$Cl$_3$F$_3$N$_2$O$_2$ (M-H)$^-$ 432.95307; found 432.95394. **Purity (HPLC):**>96% ($\lambda$ = 210 nm),>96% ($\lambda$ = 254 nm).

**SGA-94**

**Chemical structure 32.** 2-Cyano-3-hydroxy-*N*-(4-(trifluoromethyl)phenyl)but-2-enamide (Teriflunomide) – SGA-94.

According to general procedure **B**, amide **SGA-34** (228 mg, 1.00 mmol, 1.0 eq.) in dry THF (10 mL), NaH (92.0 mg, 2.30 mmol, 2.3 eq.) and acetyl chloride (78.5 µL, 1.10 mmol, 1.1 eq.) were used to give **SGA-94** as colorless crystals (206 mg, 0.761 mmol, 76%). Analytical data are in accordance with literature (*Métro et al., 2012*). $R_f$ = 0.65 (3:2 hexanes/acetone). **m.p.:** 224°C [(*Métro et al., 2012*): 230–232°C]. **$^1$H NMR (400 MHz, (CD$_3$)$_2$SO)** $\delta$/ppm = 12.28 (s, 1H, OH), 10.91 (s, 1H, NH), 7.81–7.72 (m, 2H, 2'-H, 6'-H), 7.70–7.61 (m, 2H, 3'-H, 5'-H), 2.25 (s, 3H, 4 hr). **$^{13}$C NMR (101 MHz, (CD$_3$)$_2$SO)** $\delta$/ppm = 187.1 (C-3), 166.4 (C-1), 141.9 (C-1'), 125.9 (q, $J_{CF}$ = 3.7 Hz, C-3', C-5'), 124.4 (q, $J_{CF}$ = 271.3 Hz, CF$_3$), 123.5 (q, $J_{CF}$ = 32.1 Hz, C-4'), 120.7 (C-2', C-6'), 118.9 (C-2), 80.5 (CN), 23.5 (C-4). IR (ATR) $\tilde{V}_{max}$/cm$^{-1}$=2335, 2214, 1551, 1319, 1154, 1113, 840, 679. **HRMS (ESI):** calcd. for C$_{12}$H$_8$F$_3$N$_2$O$_2$ (M-H)$^-$ 269.05434; found 269.05423. **Purity (HPLC):**>96% ($\lambda$ = 210 nm),>96% ($\lambda$ = 254 nm).

**SGA-111**

**Chemical structure 33.** 3-(3,5-Bis(trifluoromethyl)phenyl)−2-cyano-3-hydroxy-*N*-(4-(trifluoromethyl)phenyl)-acrylamide – SGA-111.

According to general procedure **B**, amide **SGA-34** (228 mg, 1.00 mmol, 1.0 eq.) in dry THF (10 mL), NaH (92.0 mg, 2.30 mmol, 2.3 eq.) and 3,5-bis(trifluoromethyl)benzoyl chloride (199 µL, 1.10 mmol, 1.1 eq.) were used to give **SGA-111** as colorless crystals (357 mg, 0.762 mmol, 76%). $R_f$ = 0.43 (3:2 hexanes/acetone). **m.p.:** 230°C. **$^1$H NMR (400 MHz, (CD$_3$)$_2$SO)** $\delta$/ppm = 12.36 (s, 1H, NH), (m, 3H, 2''-H, 6''-H, 4''-H), 7.87–7.71 (m, 2H, 2'-H, 6'-H), 7.69–7.51 (m, 2H, 3'-H, 5'-H). **$^{13}$C NMR (101 MHz, (CD$_3$)$_2$SO)** $\delta$/ppm = 182.4 (C-3), 166.3 (C-1), 144.0 (C-1''), 143.6 (C-1'), 129.8 (q, $J_{CF}$ = 32.8 Hz, C-3'', C-5''), 128.1 (q, $J_{CF}$ = 4.7 Hz, C-2'', C-6''), 126.1 (q, $J_{CF}$ = 3.6 Hz, C-3', C-5'), 126.0 (q, $J_{CF}$ = 257.2 Hz, *m*-CF$_3$), 123.4 (C-2), 123.3 (q, $J_{CF}$ = 271.6 Hz, *p*-CF$_3$), 122.7 (q, $J_{CF}$ = 4.0 Hz, C-4''), 121.7 (q, $J_{CF}$ = 31.9 Hz, C-4'), 118.5 (C-2', C-6'), 77.7 (CN). IR (ATR) $\tilde{V}_{max}$/cm$^{-1}$=3289, 2218, 1349, 1323, 1284, 1186, 1139, 1114, 836. **HRMS (ESI):** calcd. for C$_{19}$H$_8$F$_9$N$_2$O$_2$ (M-H)$^-$ 467.04475; found 467.04496. **Purity (HPLC):**>96% ($\lambda$ = 210 nm),>96% ($\lambda$ = 254 nm).

**SGA-112**

**Chemical structure 34.** 2-Cyano-3-(3,5-dimethylphenyl)−3-hydroxy-*N*-(4-(trifluoromethyl)phenyl)acrylamide – SGA-112.

According to general procedure **B**, amide **SGA-34** (228 mg, 1.00 mmol, 1.0 eq.) in dry THF (10 mL), NaH (92.0 mg, 2.30 mmol, 2.3 eq.) and 3,5-dimethylbenzoyl chloride (163 µL, 1.10 mmol, 1.1 eq.) were used to give **SGA-112** as colorless crystals (175 mg, 0.486 mmol, 49%). $R_f$ = 0.48 (3:2 hexanes/acetone). **m.p.:** 187˚C. **$^1$H NMR (500 MHz, (CD$_3$)$_2$SO)** δ/ppm = 12.08 (s, 1H, NH), 7.85–7.74 (m, 2H, 2'-H, 6'-H), 7.66–7.58 (m, 2H, 3'-H, 5'-H), 7.25 (s, 2H, 2''-H, 6''-H), 7.09 (s, 1H,4''-H), 2.30 (s, 6H, CH$_3$). **$^{13}$C NMR (126 MHz, (CD$_3$)$_2$SO)** δ/ppm = 186.5 (C-3), 167.8 (C-1), 143.6 (C-1'), 140.1 (C-1''), 137.2 (C-3'', C-5''), 131.8 (C-4''), 126.4 (q, $J_{CF}$ = 3.8 Hz, C-3', C-5'), 125.8 (C-2'', C-6''), 125.3 (q, $J_{CF}$ = 270.8 Hz, CF$_3$), 122.6 (q, $J_{CF}$ = 31.9 Hz, C-4'), 121.8 (C-2), 119.9 (C-2', C-6'), 78.2 (CN), 21.4 (CH$_3$). IR (ATR) $\tilde{V}_{max}$/cm$^{-1}$=3270, 2218, 1526.1319, 1247, 1157, 1110, 1066, 839. **HRMS (ESI):** calcd. for C$_{19}$H$_{14}$F$_3$N$_2$O$_2$ (M-H)$^-$ 359.10129; found 359.10142. **Purity (HPLC):**>96% (λ = 210 nm),>96% (λ = 254 nm).

**SGA-113**

**Chemical structure 35.** 2-Cyano-3-(3,5-dimethoxyphenyl)−3-hydroxy-*N*-(4-(trifluoromethyl)phenyl)acrylamide − SGA-113.

According to general procedure **B**, amide **SGA-34** (228 mg, 1.00 mmol, 1.0 eq.) in dry THF (10 mL), NaH (92.0 mg, 2.30 mmol, 2.3 eq.) and 3,5-dimethoxybenzoyl chloride (221 mg, 1.10 mmol, 1.1 eq.) were used to give **SGA-113** as colorless crystals (309 mg, 0.789 mmol, 79%). $R_f$ = 0.54 (3:2 hexanes/acetone). **m.p.:** 203˚C. **$^1$H NMR (400 MHz, CDCl$_3$)** δ/ppm = 7.99 (s, 1H, NH), 7.73–7.63 (m, 4H, 2'-H, 6'-H, 3'-H, 5'-H), 7.12 (d, *J* = 2.3 Hz, 2H, 2''-H, 6''-H), 6.69 (t, *J* = 2.3 Hz, 1H, 4''-H), 3.85 (s, 6H, OCH$_3$). **$^{13}$C NMR (101 MHz, CDCl$_3$)** δ/ppm = 184.1 (C-3), 168.7 (C-1), 161.0 (C-3'', C-5''), 139.1 (C-1'), 133.9 (C-1''), 127.8 (q, $J_{CF}$ = 33.2 Hz, C-4'), 126.7 (q, $J_{CF}$ = 3.8 Hz, C-3', C-5'), 123.9 (q, $J_{CF}$ = 270.1 Hz, CF$_3$), 121.0 (C-2', C-6'), 117.5 (C-2), 106.4 (C-2'', C-6''), 106.0 (C-4''), 78.5 (CN), 55.8 (OCH$_3$). IR (ATR) $\tilde{V}_{max}$/cm$^{-1}$=3297, 2215, 1550, 1324, 1208, 1156, 1095, 1067, 834. **HRMS (ESI):** calcd. for C$_{19}$H$_{14}$F$_3$N$_2$O$_4$ (M-H)$^-$ 391.09112; found 391.09140. **Purity (HPLC):**>96% (λ = 210 nm), >96% (λ = 254 nm).

**SGA-114**

**Chemical structure 36.** 2-Cyano-3-hydroxy-3-(4-(trifluoromethoxy)phenyl)-*N*-(4-(trifluoromethyl)phenyl)-acrylamide − SGA-114.

According to general procedure **B**, amide **SGA-34** (228 mg, 1.00 mmol, 1.0 eq.) in dry THF (10 mL), NaH (92.0 mg, 2.30 mmol, 2.3 eq.) and 4-(trifluoromethoxy)benzoyl chloride (173 µL, 1.10 mmol, 1.1 eq.) were used to give **SGA-114** as colorless crystals (200 mg, 0.481 mmol, 48%). $R_f$ = 0.58 (3:2 hexanes/acetone). **m.p.:** 198˚C [(*Sjogren et al., 1991*): 188–190˚C]. **$^1$H NMR (500 MHz, CDCl$_3$)** δ/ppm = 8.12–8.05 (m, 2H, 2''-H, 6''-H), 7.96 (s, 1H, NH), 7.78–7.62 (m, 4H, 2'-H, 6'-H, 3'-H, 5'-H), 7.42–7.32 (m, 2H, 3''-H, 5''-H). **$^{13}$C NMR (126 MHz, CDCl$_3$)** δ/ppm = 182.6 (C-3), 168.5 (C-1), 152.8 (C-4''), 139.0 (C-1'), 130.7 (C-2'', C-6''), 130.5 (C-1''), 128.0 (q, $J_{CF}$ = 33.0 Hz, C-4'),

126.7 (q, $J_{CF}$ = 3.7 Hz, C-3', C-5'), 123.9 (q, $J_{CF}$ = 271.7 Hz, CF$_3$), 121.0 (C-2', C-6'), 120.8 (C-3'', C-5''), 120.6 (q, $J_{CF}$ = 259.4 Hz, OCF$_3$), 117.3 (C-2), 78.6 (CN). **IR (ATR)** $\tilde{V}_{max}$/cm$^{-1}$=3285, 2215, 1597, 1551, 1505, 1268, 1168, 1128, 839. **HRMS (ESI):** calcd. for C$_{18}$H$_9$F$_6$N$_2$O$_3$ (M-H)$^-$ 415.05228; found 415.05225. **Purity (HPLC):**>96% ($\lambda$ = 210 nm),>96% ($\lambda$ = 254 nm).

**Chemical structure 37.** 2-Cyano-3-hydroxy-3-(pyridin-3-yl)-*N*-(4-(trifluoromethyl)phenyl)acrylamide – SGA-127.

According to general procedure **B**, amide **SGA-34** (228 mg, 1.00 mmol, 1.0 eq.) in dry THF (10 mL), NaH (92.0 mg, 2.30 mmol, 2.3 eq.) and nicotinoyl chloride hydrochloride (196 mg, 1.10 mmol, 1.1 eq.) were used to give **SGA-127** as orange solid (169 mg, 0.506 mmol, 51%). **R$_f$** = 0.00 (3:2 hexanes/EtOAc). **m.p.:** 235˚C. **$^1$H NMR (400 MHz, (CD$_3$)$_2$SO)** $\delta$/ppm = 12.22 (s, 1H, NH), 9.12 (s, 1H, 2''-H), 8.88 (d, *J* = 5.6 Hz, 1H, 6''-H), 8.70 (d, *J* = 8.0 Hz, 1H, 4''-H), 8.04 (dd, *J* = 8.0, 5.6 Hz, 1H, 5''-H), 7.83–7.72 (m, 2H, 2'-H, 6'-H), 7.67–7.56 (m, 2H, 3'-H, 5'-H). **$^{13}$C NMR (101 MHz, (CD$_3$)$_2$SO)** $\delta$/ppm = 180.0 (C-3), 165.9 (C-1), 143.5 (C-1'), 143.3 (C-6''), 143.0 (C-4''), 141.9 (C-2''), 139.9 (C-3''), 126.2 (C-5''), 126.1 (q, $J_{CF}$ = 3.6 Hz, C-3', C-5'), 124.2 (q, $J_{CF}$ = 270.7 Hz, CF$_3$), 123.3 (C-2), 121.8 (q, $J_{CF}$ = 32.0 Hz, C-4'), 118.5 (C-2', C-6'), 78.5 (CN). **IR (ATR)** $\tilde{V}_{max}$/cm$^{-1}$=2356, 2191, 1533, 1317, 1105, 1060, 849, 698. **HRMS (ESI):** calcd. for C$_{16}$H$_9$F$_3$N$_3$O$_2$ (M-H)$^-$ 332.06523; found 332.06517. **Purity (HPLC):**>96% ($\lambda$ = 210 nm),>96% ($\lambda$ = 254 nm).

**Chemical structure 38.** 3-(3-Bromo-5-iodophenyl)−2-cyano-3-hydroxy-*N*-(4-(trifluoromethyl)phenyl)acrylamide – SGA-132.

According to general procedure **B**, amide **SGA-34** (228 mg, 1.00 mmol, 1.0 eq.) in dry THF (10 mL), NaH (92.0 mg, 2.30 mmol, 2.3 eq.) and 3-bromo-5-iodobenzoic acid (360 mg, 1.10 mmol, 1.1 eq.; converted into the corresponding aryl chloride) were used to give **SGA-132** as colorless crystals (332 mg, 0.618 mmol, 62%). **R$_f$** = 0.38 (3:2 hexanes/EtOAc). **m.p.:** 207˚C. **$^1$H NMR (400 MHz, (CD$_3$)$_2$SO)** $\delta$/ppm = 12.31 (s, 1H, NH), 7.97 (s, 1H, 4''-H), 7.89 (s, 1H, 2''-H or 6''-H), 7.78–7.72 (m, 3H, 2'-H, 6'-H and 2''-H or 6''-H), 7.63–7.55 (m, 2H, 3'-H, 5'-H). **$^{13}$C NMR (101 MHz, (CD$_3$)$_2$SO)** $\delta$/ppm = 182.6 (C-3), 166.4 (C-1), 145.4 (C-1''), 143.6 (C-1'), 139.1 (C-4''), 135.0 (C-2'' or C-6''), 129.5 (C-2'' or C-6''), 126.0 (q, $J_{CF}$ = 3.8 Hz, C-3', C-5'), 123.9 (q, $J_{CF}$ = 263.8 Hz, CF$_3$), 123.3 (C-2), 121.7 (q, $J_{CF}$ = 31.6 Hz, C-4'), 121.6 (C-3'' or C-5''), 118.6 (C-2', C-6'), 95.0 (C-3'' or C-5''), 77.4 (CN). **IR (ATR)** $\tilde{V}_{max}$/cm$^{-1}$=3276, 2213, 1532, 1318, 1163, 1112, 731. **HRMS (ESI):** calcd. for C$_{17}$H$_8{}^{79}$BrIF$_3$N$_2$O$_2$ (M-H)$^-$ 534.87714; found 534.87813. **Purity (HPLC):**>96% ($\lambda$ = 210 nm),>96% ($\lambda$ = 254 nm).

**Chemical structure 39.** 3-(5-Chloropyridin-3-yl)−2-cyano-3-hydroxy-N-(4-(trifluoromethyl)phenyl)acrylamide – SGA-136.

According to general procedure **B**, amide **SGA-34** (228 mg, 1.00 mmol, 1.0 eq.) in dry THF (10 mL), NaH (92.0 mg, 2.30 mmol, 2.3 eq.) and 5-chloronicotinic acid (173 mg, 1.10 mmol, 1.1 eq.; converted into the corresponding aryl chloride) were used to give **SGA-136** as light pink crystals (225 mg, 0.611 mmol, 61%). $R_f$ = 0.15 (3:2 hexanes/acetone). **m.p.:** 221°C. **$^1$H NMR (500 MHz, (CD$_3$)$_2$SO)** $\delta$/ppm = 12.33 (s, 1H, NH), 8.74 (s, 1H, 2''-H or 4''-H), 8.69–8.64 (m, 1H, 6''-H), 8.11–8.05 (m, 1H, 2''-H or 4''-H), 7.81–7.73 (m, 2H, 2'-H, 6'-H), 7.64–7.55 (m, 2H, 3'-H, 5'-H). **$^{13}$C NMR (126 MHz, (CD$_3$)$_2$SO)** $\delta$/ppm = 181.7 (C-3), 166.2 (C-1), 147.9 (C-6''), 146.1 (C-2'' or C-4''), 143.6 (C-1'), 138.8 (C-3'' or C-5''), 134.9 (C-2'' or C-4''), 130.5 (C-3'' or C-5''), 126.1 (q, $J_{CF}$ = 3.7 Hz, C-3', C-5'), 124.6 (q, $J_{CF}$ = 271.0 Hz, CF$_3$), 123.5 (C-2), 121.6 (q, $J_{CF}$ = 31.5 Hz, C-4'), 118.5 (C-2', C-6'), 78.2 (CN). IR (ATR) $\tilde{V}_{max}$/cm$^{-1}$=3285, 2225, 1539, 1308, 1113, 951, 838. **HRMS (ESI):** calcd. for C$_{16}$H$_8$$^{35}$ClF$_3$N$_3$O$_2$ (M-H)$^-$ 366.02626; found 366.02652. **Purity (HPLC):**>96% ($\lambda$ = 210 nm),>96% ($\lambda$ = 254 nm).

**Chemical structure 40.** 2-Cyano-3-(1-methyl-1*H*-pyrrol-2-yl)−3-hydroxy-N-(4-(trifluoromethyl)phenyl)acrylamide – SGA-32.

According to general procedure **B**, amide **SGA-34** (228 mg, 1.00 mmol, 1.0 eq.) in dry THF (10 mL), NaH (92.0 mg, 2.30 mmol, 2.3 eq.) and 1-methylpyrrole-2-carbonyl chloride (158 mg, 1.10 mmol, 1.1 eq.) were used to give **SGA-32** as colorless crystals (221 mg, 0.660 mmol, 66%). $R_f$ = 0.57 (3:2 hexanes/acetone). **m.p.:** 203°C. **$^1$H NMR (400 MHz, CDCl$_3$)** $\delta$/ppm = 7.86 (s, 1H, NH), 7.69–7.61 (m, 4H, 2'-H, 6'-H, 3'-H, 5'-H), 7.57 (dd, $J$ = 4.3, 1.6 Hz, 1H, 3''-H), 6.93–6.90 (m, 1H, 4''-H), 6.27 (dd, $J$ = 4.3, 2.5 Hz, 1H, 5''-H), 3.93 (s, 3H, CH$_3$). **$^{13}$C NMR (101 MHz, CDCl$_3$)** $\delta$/ppm = 175.6 (C-3), 170.2 (C-1), 139.5 (C-1'), 132.6 (C-5''), 127.4 (q, $J_{CF}$ = 31.8 Hz, C-4'), 126.6 (q, $J$ = 3.8 Hz, C-3', C-5'), 124.1 (C-2''), 124.0 (q, $J_{CF}$ = 271.4 Hz, CF$_3$), 121.3 (C-3''), 120.7 (C-2', C-6'), 119.0 (C-2), 110.0 (C-4''), 72.6 (CN), 38.7 (CH$_3$). IR (ATR) $\tilde{V}_{max}$/cm$^{-1}$=3273, 2210, 1518, 1379, 1228, 1111, 996, 837, 745. **HRMS (ESI):** calcd. for C$_{16}$H$_{11}$F$_3$N$_3$O$_2$ (M-H)$^-$ 334.08088; found 334.08090. **Purity (HPLC):**>96% ($\lambda$ = 210 nm),>96% ($\lambda$ = 254 nm).

**SGA-31**

**Chemical structure 41.** 2-Cyano-3-(1-methyl-1$H$-pyrrol-2-yl)−3-hydroxy-$N$-phenylacrylamide (Prinomide) – SGA-31.

According to general procedure **B**, amide **2** (228 mg, 1.00 mmol, 1.0 eq.) in dry THF (10 mL), NaH (92.0 mg, 2.30 mmol, 2.3 eq.) and 1-methylpyrrole-2-carbonyl chloride (158 mg, 1.10 mmol, 1.1 eq.) were used to give **SGA-31** as colorless crystals (200 mg, 0.747 mmol, 75%). $R_f$ = 0.65 (3:2 hexanes/acetone). **m.p.:** 171°C [(*Walker, 1981*): 174–175°C]. $^1$**H NMR (400 MHz, CDCl$_3$)** $\delta$/ppm = 7.74 (s, 1H, NH), 7.54 (dd, $J$ = 4.3, 1.6 Hz, 1H, 3''-H), 7.50 (d, $J$ = 8.0 Hz, 2H, 2'-H, 6'-H), 7.44–7.34 (m, 2H, 3'-H, 5'-H), 7.20 (t, $J$ = 7.4 Hz, 1H, 4'-H), 6.89 (t, $J$ = 1.9 Hz, 1H, 5''-H), 6.25 (dd, $J$ = 4.3, 2.5 Hz, 1H, 4''-H), 3.92 (s, 3H, CH$_3$). $^{13}$**C NMR (101 MHz, CDCl$_3$)** $\delta$/ppm = 175.6 (C-3), 170.0 (C-1), 136.2 (C-1'), 132.0 (C-5''), 129.3 (C-3', C-5'), 125.7 (C-4'), 124.4 (C-2''), 121.3 (C-2', C-6'), 120.7 (C-3''), 119.1 (C-2), 109.7 (C-4''), 72.6 (CN), 38.5 (CH$_3$). IR (ATR) $\tilde{V}_{max}$/cm$^{-1}$=3293, 2210, 1526, 1382, 1231, 751, 736, 687. **HRMS (ESI):** calcd. for C$_{15}$H$_{12}$N$_3$O$_2$ (M-H)$^-$ 266.09350; found 266.09370. **Purity (HPLC):** >96% ($\lambda$ = 210 nm),>96% ($\lambda$ = 254 nm).

**SGA-1**

**Chemical structure 42.** 2-Cyano-3-(3,5-dichlorophenyl)−3-hydroxy-$N$-phenylacrylamide – SGA-1.

According to general procedure **B**, amide **2** (160 mg, 1.00 mmol, 1.0 eq.) in dry THF (10 mL), NaH (92.0 mg, 2.30 mmol, 2.3 eq.) and 3,5-dichlorobenzoyl chloride (230 mg, 1.10 mmol, 1.1 eq.) were used to give **SGA-1** as yellow solid (227 mg, 0.681 mmol, 68%). $R_f$ = 0.67 (3:2 hexanes/acetone). **m.p.:** 200°C. $^1$**H NMR (500 MHz, (CD$_3$)$_2$SO)** $\delta$/ppm = 11.95 (s, 1H, NH), 7.61 (t, $J$ = 1.9 Hz, 1H, 4''-H), 7.56 (d, $J$ = 1.9 Hz, 2H, 2''-H, 6''-H), 7.53 (dd, $J$ = 8.5, 1.0 Hz, 2H, 2'-H, 6'-H), 7.27–7.22 (m, 2H, 3'-H, 5'-H), 6.96–6.90 (m, 1H, 4'-H). $^{13}$**C NMR (126 MHz, (CD$_3$)$_2$SO)** $\delta$/ppm = 182.2 (C-3), 166.1 (C-1), 145.3 (C-1''), 140.1 (C-1'), 133.4 (C-3'', C-5''), 128.7 (C-3', C-5'), 128.3 (C-4''), 126.1 (C-2'', C-6''), 123.6 (C-2), 121.7 (C-4'), 118.8 (C-2', C-6'), 77.4 (CN). IR (ATR) $\tilde{V}_{max}$/cm$^{-1}$=3294, 2212, 1579, 1445, 810, 748, 683. **HRMS (ESI):** calcd. for C$_{16}$H$_9{}^{35}$Cl$_2$N$_2$O$_2$ (M-H)$^-$ 331.00466; found 331.00465. **Purity (HPLC):**>96% ($\lambda$ = 210 nm),>96% ($\lambda$ = 254 nm).

**SGA-4**

**Chemical structure 43.** 2-Cyano-3-(3,5-dichlorophenyl)−3-hydroxy-$N$-($p$-tolyl)acrylamide – SGA-4.

According to general procedure **B**, amide **1** (174 mg, 1.00 mmol, 1.0 eq.) in dry THF (10 mL), NaH (92.0 mg, 2.30 mmol, 2.3 eq.) and 3,5-dichlorobenzoyl chloride (230 mg, 1.10 mmol, 1.1 eq.) were used to give **SGA-4** as yellow crystals (185 mg, 0.534 mmol, 53%). $R_f$ = 0.28 (3:2 hexanes/acetone). **m.p.:** 214°C. **$^1$H NMR (400 MHz, (CD$_3$)$_2$SO)** $\delta$/ppm = 11.82 (s, 1H, NH), 7.63 (t, $J$ = 1.9 Hz, 1H, 4''-H), 7.56 (d, $J$ = 1.9 Hz, 2H, 2''-H, 6''-H), 7.46–7.39 (m, 2H, 2'-H, 6'-H), 7.09–7.02 (m, 2H, 3'-H, 5'-H), 2.24 (s, 3H, CH$_3$). **$^{13}$C NMR (101 MHz, (CD$_3$)$_2$SO)** $\delta$/ppm = 182.0 (C-3), 166.0 (C-1), 145.1 (C-1''), 137.4 (C-1'), 133.4 (C-3'', C-5''), 130.6 (C-4'), 129.1 (C-3', C-5'), 128.3 (C-4''), 126.1 (C-2'', C-6''), 123.5 (C-2), 119.0 (C-2', C-6'), 77.5 (CN), 20.4 (CH$_3$). IR (ATR) $\tilde{V}_{max}$/cm$^{-1}$=3299, 2212, 1543, 1518.866, 806, 654. **HRMS (ESI):** calcd. for C$_{17}$H$_{11}^{35}$Cl$_2$N$_2$O$_2$ (M-H)$^-$ 345.02031; found 345.02052. **Purity (HPLC):**>96% ($\lambda$ = 210 nm),>96% ($\lambda$ = 254 nm).

**Chemical structure 44.** 3-(3,5-Bis(trifluoromethyl)phenyl)−2-cyano-3-hydroxy-*N*-(*p*-tolyl)acrylamide – SGA-108.

According to general procedure **B**, amide **1** (174 mg, 1.00 mmol, 1.0 eq.) in dry THF (10 mL), NaH (92.0 mg, 2.30 mmol, 2.3 eq.) and 3,5-bis(trifluoromethyl)benzoyl chloride (199 µL, 1.10 mmol, 1.1 eq.) were used to give **SGA-108** as light yellow crystals (208 mg, 0.502 mmol, 50%). $R_f$ = 0.28 (3:2 hexanes/acetone). **m.p.:** 183°C. **$^1$H NMR (500 MHz, CDCl$_3$)** $\delta$/ppm = 8.45 (s, 2H, 2''-H, 6''-H), 8.10 (s, 1H, 4''-H), 7.86 (s, 1H, NH), 7.43–7.37 (m, 2H, 2'-H, 6'-H), 7.25–7.20 (m, 2H, 3'-H, 5'-H), 2.37 (s, 3H, CH$_3$). **$^{13}$C NMR (126 MHz, CDCl$_3$)** $\delta$/ppm = 180.4 (C-3), 167.9 (C-1), 136.7 (C-4'), 134.7 (C-1''), 132.7 (C-1'), 132.6 (q, $J_{CF}$ = 34.2 Hz, C-3'', C-5''), 130.1 (C-3', C-5'), 128.6 (q, $J_{CF}$ = 3.1 Hz, C-2'', C-6''), 126.3 (q, $J_{CF}$ = 3.7 Hz, C-4''), 122.8 (q, $J_{CF}$ = 272.9 Hz, CF$_3$), 121.8 (C-2', C-6'), 116.5 (C-2), 79.7 (CN), 21.2 (CH$_3$). IR (ATR) $\tilde{V}_{max}$/cm$^{-1}$=3276, 2213, 1538, 1278, 1129, 811, 681. **HRMS (ESI):** calcd. for C$_{19}$H$_{11}$F$_6$N$_2$O$_2$ (M-H)$^-$ 413.07302; found 413.07301. **Purity (HPLC):**>96% ($\lambda$ = 210 nm),>96% ($\lambda$ = 254 nm).

**Chemical structure 45.** N-(4-Chlorophenyl)−2-cyano-3-(3,5-dichlorophenyl)−3-hydroxyacrylamide – SGA-2.

According to general procedure **B**, amide **4** (195 mg, 1.00 mmol, 1.0 eq.) in dry THF (10 mL), NaH (92.0 mg, 2.30 mmol, 2.3 eq.) and 3,5-dichlorobenzoyl chloride (230 mg, 1.10 mmol, 1.1 eq.) were used to give **SGA-2** as yellow solid (256 mg, 0.695 mmol, 70%). $R_f$ = 0.71 (3:2 hexanes/acetone). **m.p.:** 230°C. **$^1$H NMR (500 MHz, (CD$_3$)$_2$SO)** $\delta$/ppm = 12.12 (s, 2H, NH), 7.62 (t, $J$ = 1.9 Hz, 1H, 4''-H), 7.60–7.57 (m, 2H, 2'-H, 6'-H), 7.55 (d, $J$ = 1.9 Hz, 2H, 2''-H, 6''-H), 7.30–7.26 (m, 2H, 3'-H, 5'-H). **$^{13}$C NMR (126 MHz, (CD$_3$)$_2$SO)** $\delta$/ppm = 182.5 (C-3), 166.1 (C-1), 145.4 (C-1''), 139.1 (C-1'), 133.4 (C-3'', C-5''), 128.6 (C-3', C-5'), 128.3 (C-4''), 126.0 (C-2'', C-6''), 125.0 (C-4'), 123.6 (C-2), 120.2 (C-2', C-6'), 77.2 (CN). IR (ATR) $\tilde{V}_{max}$/cm$^{-1}$=3305, 2212, 1545, 1495, 1506, 1316, 1098, 809. **HRMS (ESI):** calcd. for C$_{16}$H$_8^{35}$Cl$_3$N$_2$O$_2$ (M-H)$^-$ 364.96568; found 364.96593. **Purity (HPLC):**>96% ($\lambda$ = 210 nm),>96% ($\lambda$ = 254 nm).

**Chemical structure 46.** N-(4-Bromophenyl)−2-cyano-3-(3,5-dichlorophenyl)−3-hydroxyacrylamide – SGA-3.

According to general procedure **B**, amide **5** (239 mg, 1.00 mmol, 1.0 eq.) in dry THF (10 mL), NaH (92.0 mg, 2.30 mmol, 2.3 eq.) and 3,5-dichlorobenzoyl chloride (230 mg, 1.10 mmol, 1.1 eq.) were used to give **SGA-3** as yellow crystals (289 mg, 0.702 mmol, 70%). $R_f$ = 0.73 (3:2 hexanes/acetone). **m.p.:** 237°C. **¹H NMR (400 MHz, (CD₃)₂SO)** $\delta$/ppm = 12.09 (s, 1H, NH), 7.62 (t, *J* = 1.9 Hz, 1H, 4''-H), 7.58–7.51 (m, 4H, 2'-H, 6'-H, 2''-H, 6''-H), 7.44–7.38 (m, 2H, 3'-H, 5'-H).**¹³C NMR (101 MHz, (CD₃)₂SO)** $\delta$/ppm = 182.4 (C-3), 166.1 (C-1), 145.2 (C-1''), 139.4 (C-1'), 133.4 (C-3'', C-5''), 131.4 (C-3', C-5'), 128.3 (C-4''), 126.0 (C-2'', C-6''), 122.6 (C-2), 120.7 (C-2', C-6'), 112.9 (C-4'), 77.4 (CN). IR (ATR) $\tilde{V}_{max}$/cm$^{-1}$=3309, 2211, 1540, 1493, 1403, 1316, 1290, 1012, 809. **HRMS (ESI):** calcd. for $C_{16}H_8{}^{79}Br^{35}Cl_2N_2O_2$ (M-H)$^-$ 408.91517; found 408.91581. **Purity (HPLC):**>96% ($\lambda$ = 210 nm), >96% ($\lambda$ = 254 nm).

**Chemical structure 47.** N-(4-Bromophenyl)−2-cyano-3-(3,5-dibromophenyl)−3-hydroxyacrylamide – SGA-75.

According to general procedure **B**, amide **5** (239 mg, 1.00 mmol, 1.0 eq.) in dry THF (10 mL), NaH (92.0 mg, 2.30 mmol, 2.3 eq.) and 3,5-dibromobenzoic acid (308 mg, 1.10 mmol, 1.1 eq.; converted into the corresponding aryl chloride) were used to give **SGA-75** as colorless crystals (222 mg, 0.443 mmol, 44%). $R_f$ = 0.20 (3:2 hexanes/acetone). **m.p.:** 233°C. **¹H NMR (400 MHz, (CD₃)₂SO)** $\delta$/ppm = 12.05 (s, 1H, NH), 7.86 (t, *J* = 1.8 Hz, 1H, 4'-H), 7.73 (d, *J* = 1.8 Hz, 2H, 2''-H, 6''-H), 7.57–7.50 (m, 2H, 2'-H, 6'-H), 7.46–7.37 (m, 2H, 3'-H, 5'-H). **¹³C NMR (101 MHz, (CD₃)₂SO)** $\delta$/ppm = 182.2 (C-3), 166.1 (C-1), 145.5 (C-1''), 139.4 (C-1'), 133.7 (C-4''), 131.5 (C-3', C-5'), 129.2 (C-2'', C-6''), 122.7 (C-2), 121.8 (C-3'', C-5''), 120.8 (C-2', C-6'), 113.0 (C-4'), 77.4 (CN). IR (ATR) $\tilde{V}_{max}$/cm$^{-1}$=3314, 2214, 1596, 1543, 865, 817, 748, 658. **HRMS (ESI):** calcd. for $C_{16}H_8{}^{79}Br_3N_2O_2$ (M-H)$^-$ 496.81414; found 496.81449. **Purity (HPLC):**>96% ($\lambda$ = 210 nm),>96% ($\lambda$ = 254 nm).

**Chemical structure 48.** N-(4-Bromophenyl)−2-cyano-3-hydroxy-3-(2,4,6-trichlorophenyl)acrylamide – SGA-72.

According to general procedure **B**, amide **5** (239 mg, 1.00 mmol, 1.0 eq.) in dry THF (10 mL), NaH (92.0 mg, 2.30 mmol, 2.3 eq.) and 2,4,6-trichlorobenzoyl chloride (172 μL, 1.10 mmol, 1.1 eq.) were used to give **SGA-72** as colorless solid (227 mg, 0.507 mmol, 51%). $R_f$ = 0.14 (3:2 hexanes/acetone). **m.p.:** 191°C. **$^1$H NMR (400 MHz, (CD$_3$)$_2$SO)** δ/ppm = 11.69 (s, 1H, NH), 7.65 (s, 2H, 3''-H, 5''-H), 7.58–7.52 (m, 2H, 2'-H, 6'-H), 7.44–7.38 (m, 2H, 3'-H, 5'-H). **$^{13}$C NMR (101 MHz, (CD$_3$)$_2$SO)** δ/ppm = 180.3 (C-3), 165.4 (C-1), 139.6 (C-1' or C-1''), 139.4 (C-1' or C-1''), 133.0 (C-4''), 132.0 (C-2'', C-6''), 131.5 (C-3', C-5'), 127.8 (C-3'', C-5''), 122.1 (C-2), 120.5 (C-2', C-6'), 112.9 (C-4'), 79.5 (CN). IR (ATR) $\tilde{V}_{max}$/cm$^{-1}$=2238, 1588, 1539, 1488, 1307, 856, 818. **HRMS (ESI):** calcd. for C$_{16}$H$_7$$^{79}$Br$^{35}$Cl$_3$N$_2$O$_2$ (M-H)$^-$ 442.87620; found 442.87747. **Purity (HPLC):**>96% (λ = 210 nm),>96% (λ = 254 nm).

**Chemical structure 49.** Cyano-3-(3,5-dichlorophenyl)-N-(4-fluorophenyl)−3-hydroxyacrylamide – SGA-8.

According to general procedure **B**, amide **6** (178 mg, 1.00 mmol, 1.0 eq.) in dry THF (10 mL), NaH (92.0 mg, 2.30 mmol, 2.3 eq.) and 3,5-dichlorobenzoyl chloride (230 mg, 1.10 mmol, 1.1 eq.) were used to give **SGA-8** as white crystals (220 mg, 0.626 mmol, 63%). $R_f$ = 0.69 (3:2 hexanes/acetone). **m.p.:** 225°C. **$^1$H NMR (400 MHz, (CD$_3$)$_2$SO)** δ/ppm = 11.92 (s, 1H, NH), 7.63 (t, J = 1.9 Hz, 1H, 4''-H), 7.60–7.53 (m, 4H, 2'-H, 6'-H, 2''-H, 6''-H), 7.12–7.03 (m, 2H, 3'-H, 5'-H). **$^{13}$C NMR (101 MHz, (CD$_3$)$_2$SO)** δ/ppm = 182.2 (C-3), 166.1 (C-1), 157.3 (d, $J_{CF}$ = 236.9 Hz, C-4'), 144.9 (C-1''), 136.3 (d, $J_{CF}$ = 2.4 Hz, C-1'), 133.4 (C-3'', C-5''), 128.4 (C-4''), 126.1 (C-2'', C-6''), 123.3 (C-2), 120.5 (d, $J_{CF}$ = 7.6 Hz, C-2', C-6'), 115.2 (d, $J_{CF}$ = 22.0 Hz, C-3', C-5'), 77.4 (CN). IR (ATR) $\tilde{V}_{max}$/cm$^{-1}$=3296, 2212, 1739, 1549, 1506, 1210, 823, 809, 645. **HRMS (ESI):** calcd. for C$_{16}$H$_8$$^{35}$Cl$_2$FN$_2$O$_2$ (M-H)$^-$ 348.99523; found 348.99526. **Purity (HPLC):**>96% (λ = 210 nm),>96% (λ = 254 nm).

**Chemical structure 50.** 2-Cyano-3-(3,5-dichlorophenyl)-*N*-(4-iodophenyl)−3-hydroxyacrylamide – SGA-9.

According to general procedure **B**, amide **7** (286 mg, 1.00 mmol, 1.0 eq.) in dry THF (10 mL), NaH (92.0 mg, 2.30 mmol, 2.3 eq.) and 3,5-dichlorobenzoyl chloride (230 mg, 1.10 mmol, 1.1 eq.) were used to give **SGA-9** as yellow crystals (279 mg, 0.608 mmol, 61%). $R_f$ = 0.71 (3:2 hexanes/acetone). **m.p.:** 238°C. $^1$**H NMR (400 MHz, (CD$_3$)$_2$SO)** $\delta$/ppm = 12.04 (s, 1H, NH), 7.63 (t, *J* = 1.9 Hz, 1H, 4''-H), 7.59–7.53 (m, 4H, 3'-H, 5'-H, 2''-H, 6''-H), 7.44–7.38 (m, 2H, 2'-H, 6'-H). $^{13}$**C NMR (101 MHz, (CD$_3$)$_2$SO)** $\delta$/ppm = 182.4 (C-3), 166.2 (C-1), 145.1 (C-1''), 139.8 (C-1'), 137.3 (C-3', C-5' or C-2'', C-6''), 133.4 (C-3'', C-5''), 128.4 (C-4''), 126.1 (C-3', C-5' or C-2'', C-6''), 123.3 (C-2), 121.1 (C-2', C-6'), 84.5 (C-4'), 77.5 (CN). **IR (ATR)** $\tilde{V}_{max}$/cm$^{-1}$=3303, 2217, 1592, 1523, 1485, 1314, 817, 806, 658. **HRMS (ESI):** calcd. for C$_{16}$H$_8$$^{35}$Cl$_2$IN$_2$O$_2$ (M-H)$^-$ 456.90130; found 456.90014. **Purity (HPLC):** >96% ($\lambda$ = 210 nm),>96% ($\lambda$ = 254 nm).

**Chemical structure 51.** 2-Cyano-3-(3,5-dichlorophenyl)-*N*-(4-nitrophenyl)−3-hydroxyacrylamide – SGA-13.

According to general procedure **B**, amide **8** (205 mg, 1.00 mmol, 1.0 eq.) in dry THF (16 mL), NaH (92.0 mg, 2.30 mmol, 2.3 eq.) and 3,5-dichlorobenzoyl chloride (230 mg, 1.10 mmol, 1.1 eq.) were used to give **SGA-13** as yellow solid (208 mg, 0.550 mmol, 55%). $R_f$ = 0.82 (3:2 hexanes/acetone). **m.p.:** 246°C. $^1$**H NMR (400 MHz, (CD$_3$)$_2$SO)** $\delta$/ppm = 12.64 (s, 1H, NH), 8.19–8.13 (m, 2H, 3'-H, 5'-H), 7.82–7.77 (m, 2H, 2'-H, 6'-H), 7.64 (t, *J* = 1.9 Hz, 1H, 4''-H), 7.57 (d, *J* = 1.9 Hz, 2H, 2''-H, 6''-H). $^{13}$**C NMR (101 MHz, (CD$_3$)$_2$SO)** $\delta$/ppm = 181.6 (C-3), 166.9 (C-1), 149.4 (C-1''), 147.0 (C-1'), 141.3 (C-4'), 133.9 (C-3'', C-5''), 129.0 (C-4''), 126.5 (C-2'', C-6''), 125.7 (C-3', C-5'), 121.8 (C-2), 118.6 (C-2', C-6'), 77.0 (CN). **IR (ATR)** $\tilde{V}_{max}$/cm$^{-1}$=3314, 2209, 1568, 1546, 1514, 1498, 1340, 1309, 847, 813, 656. **HRMS (ESI):** calcd. for C$_{16}$H$_8$$^{35}$Cl$_2$N$_3$O$_4$ (M-H)$^-$ 375.98973; found 375.98970. **Purity (HPLC):**>96% ($\lambda$ = 210 nm),>96% ($\lambda$ = 254 nm).

**Chemical structure 52.** 2-Cyano-3-(3,5-dichlorophenyl)-*N*-(4-methoxyphenyl)−3-hydroxyacrylamide – SGA-12.

According to general procedure **B**, amide **3** (190 mg, 1.00 mmol, 1.0 eq.) in dry THF (10 mL), NaH (92.0 mg, 2.30 mmol, 2.3 eq.) and 3,5-dichlorobenzoyl chloride (230 mg, 1.10 mmol, 1.1 eq.) were used to give **SGA-12** as yellow solid (220 mg, 0.606 mmol, 61%). $R_f$ = 0.76 (3:2 hexanes/acetone). **m.p.:** 207°C. **$^1$H NMR (400 MHz, (CD$_3$)$_2$SO)** $\delta$/ppm = 11.36 (s, 1H, NH), 9.87 (s, 1H, OH), 7.71 (t, $J$ = 1.8 Hz, 1H, 4'-H), 7.65 (d, $J$ = 1.8 Hz, 2H, 2''-H, 6''-H), 7.49–7.41 (m, 2H, 2'-H, 6'-H), 6.90–6.85 (m, 2H, 3'-H, 5'-H), 3.72 (s, 3H, OCH$_3$). $^{13}$C NMR (101 MHz, (CD$_3$)$_2$SO) $\delta$/ppm = 181.6 (C-3), 166.4 (C-1), 155.2 (C-4'), 142.6 (C-1''), 133.7 (C-3'', C-5''), 131.9 (C-1'), 129.3 (C-4''), 126.3 (C-2'', C-6''), 121.7 (C-2', C-6'), 121.4 (C-2), 113.9 (C-3', C-5'), 78.1 (CN), 55.2 (OCH$_3$). IR (ATR) $\tilde{V}_{max}$/cm$^{-1}$=3296, 2211, 1601, 1467, 1441, 1297, 1251, 1032, 764. HRMS (ESI): calcd. for C$_{17}$H$_{11}$$^{35}$Cl$_2$N$_2$O$_3$ (M-H)$^-$ 361.01522; found 361.01516. Purity (HPLC):>96% ($\lambda$ = 210 nm),>96% ($\lambda$ = 254 nm).

**Chemical structure 53.** 2-Cyano-*N*-(4-cyanophenyl)−3-(3,5-dichlorophenyl)−3-hydroxyacrylamide – SGA-38.

According to general procedure **B**, amide **9** (185 mg, 1.00 mmol, 1.0 eq.) in dry THF (10 mL), NaH (92.0 mg, 2.30 mmol, 2.3 eq.) and 3,5-dichlorobenzoyl chloride (230 mg, 1.10 mmol, 1.1 eq.) were used to give **SGA-38** as yellow solid (137 mg, 0.382 mmol, 38%). $R_f$ = 0.12 (3:2 hexanes/acetone). **m.p.:** 236°C. **$^1$H NMR (500 MHz, (CD$_3$)$_2$SO)** $\delta$/ppm = 12.42 (s, 1H, NH), 7.75–7.72 (m, 2H, 3'-H, 5'-H), 7.71–7.67 (m, 2H, 2'-H, 6'-H), 7.64 (t, $J$ = 1.9 Hz, 1H, 4''-H), 7.57 (d, $J$ = 1.9 Hz, 2H, 2''-H, 6''-H). **$^{13}$C NMR (126 MHz, (CD$_3$)$_2$SO)** $\delta$/ppm = 183.0 (C-3), 166.4 (C-1), 145.1 (C-1''), 144.3 (C-1'), 133.5 (C-3'', C-5''), 133.3 (C-3', C-5'), 128.5 (C-4''), 126.0 (C-2'', C-6''), 123.2 (C-4'), 119.4 (C-2), 118.8 (C-2', C-6'), 103.1 (4'-CN), 77.4 (2-CN). IR (ATR) $\tilde{V}_{max}$/cm$^{-1}$=3321, 2360, 2340, 1533, 839, 655. **HRMS (ESI):** calcd. for C$_{17}$H$_8$$^{35}$Cl$_2$N$_3$O$_2$ (M-H)$^-$ 355.99991; found 356.00057. **Purity (HPLC):**>96% ($\lambda$ = 210 nm),>96% ($\lambda$ = 254 nm).

**Chemical structure 54.** N-(4-Acetylphenyl)−2-cyano-3-(3,5-dichlorophenyl)−3-hydroxyacrylamide – SGA-76.

According to general procedure **B**, amide **10** (202 mg, 1.00 mmol, 1.0 eq.) in dry THF (10 mL), NaH (92.0 mg, 2.30 mmol, 2.3 eq.) and 3,5-dichlorobenzoyl chloride (230 mg, 1.10 mmol, 1.1 eq.) were used to give **SGA-76** as off white solid (296 mg, 0.789 mmol, 79%). $R_f$ = 0.08 (3:2 hexanes/acetone). **m.p.:** 209°C. **$^1$H NMR (500 MHz, (CD$_3$)$_2$SO)** $\delta$/ppm = 12.32 (s, 1H, NH), 7.91–7.86 (m, 2H, 3'-H, 5'-H), 7.71–7.66 (m, 2H, 2'-H, 6'-H), 7.64 (t, $J$ = 1.9 Hz, 1H, 4''-H), 7.58 (d, $J$ = 1.9 Hz, 2H, 2''-H, 6''-H), 2.51 (s, 3H, CH$_3$, collapses with DMSO). $^{13}$C NMR (101 MHz, (CD$_3$)$_2$SO) $\delta$/ppm = 196.3 (CO), 182.8 (C-3), 166.3 (C-1), 145.1 (C-1''), 144.5 (C-1'), 133.5 (C-3'', C-5''), 130.4 (C-4'), 129.7 (C-3', C-5'), 128.5 (C-4''), 126.1 (C-2'', C-6''), 123.3 (C-2), 117.9 (C-2', C-6'), 77.6 (CN), 26.3 (CH$_3$). IR (ATR) $\tilde{V}_{max}$/cm$^{-1}$=3304, 2207, 1682, 1596, 1544, 1355, 1272, 809. **HRMS (ESI):** calcd. for C$_{18}$H$_{11}$$^{35}$Cl$_2$N$_2$O$_3$ (M-H)$^-$ 373.01522; found 373.01580. **Purity (HPLC):**>96% ($\lambda$ = 210 nm),>96% ($\lambda$ = 254 nm).

**Chemical structure 55.** 2-Cyano-3-(3,5-dichlorophenyl)−3-hydroxy-*N*-(4-propoxyphenyl)acrylamide – SGA-84.

According to general procedure **B**, amide **11** (218 mg, 1.00 mmol, 1.0 eq.) in dry THF (10 mL), NaH (92.0 mg, 2.30 mmol, 2.3 eq.) and 3,5-dichlorobenzoyl chloride (230 mg, 1.10 mmol, 1.1 eq.) were used to give **SGA-84** as yellow solid (245 mg, 0.626 mmol, 63%). $R_f$ = 0.15 (3:2 hexanes/acetone + 2% TEA). **m.p.:** 184°C. **$^1$H NMR (500 MHz, (CD$_3$)$_2$SO)** $\delta$/ppm = 11.36 (s, 1H, NH), 9.30 (s, 1H, OH), 7.71 (t, *J* = 1.7 Hz, 1H, 4''-H), 7.67–7.62 (m, 2H, 2''-H, 6''-H), 7.49–7.40 (m, 2H, 3'-H, 5'-H), 6.90–6.82 (m, 2H, 2'-H, 6'-H), 3.88 (t, *J* = 6.5 Hz, 2H, C$\underline{H}_2$CH$_2$CH$_3$), 1.71 (sext, *J* = 7.2 Hz, 2H, CH$_2$C$\underline{H}_2$CH$_3$), 0.97 (t, *J* = 7.4 Hz, 3H, CH$_2$CH$_2$C$\underline{H}_3$). **$^{13}$C NMR (126 MHz, (CD$_3$)$_2$SO)** $\delta$/ppm = 181.6 (C-3), 166.4 (C-1), 154.6 (C-4'), 142.7 (C-1''), 133.7 (C-3'', C-5''), 131.9 (C-1'), 129.3 (C-4''), 126.3 (C-2'', C-6''), 121.6 (C-3', C-5'), 121.5 (C-2), 114.5 (C-2', C-6'), 78.0 (CN), 69.1 ($\underline{C}$H$_2$CH$_2$CH$_3$), 22.1 (CH$_2$$\underline{C}$H$_2$CH$_3$), 10.4 (CH$_2$CH$_2$$\underline{C}$H$_3$). IR (ATR) $\tilde{V}_{max}$/cm$^{-1}$=3304, 2208, 1601, 1550, 1511, 1249, 1235, 822, 811. **HRMS (ESI):** calcd. for C$_{19}$H$_{15}$$^{35}$Cl$_2$N$_2$O$_3$ (M-H)$^-$ 389.04652; found 389.04631. **Purity (HPLC):**>96% ($\lambda$ = 210 nm),>96% ($\lambda$ = 254 nm).

**Chemical structure 56.** 2-Cyano-3-(3,5-dichlorophenyl)−3-hydroxy-*N*-(4-(trifluoromethoxy)phenyl)acrylamide – SGA-86.

According to general procedure **B**, amide **12** (244 mg, 1.00 mmol, 1.0 eq.) in dry THF (10 mL), NaH (92.0 mg, 2.30 mmol, 2.3 eq.) and 3,5-dichlorobenzoyl chloride (230 mg, 1.10 mmol, 1.1 eq.) were used to give **SGA-86** as colorless solid (344 mg, 0.825 mmol, 83%). $R_f$ = 0.53 (3:2 hexanes/acetone). **m.p.:** 195°C. **$^1$H NMR (500 MHz, (CD$_3$)$_2$SO)** $\delta$/ppm = 12.02 (s, 1H, NH), 9.41 (s, 1H, OH), 7.69–7.64 (m, 3H, 2'-H, 6'-H, 4''-H), 7.59 (d, *J* = 1.9 Hz, 2H, 2''-H, 6''-H), 7.28–7.23 (m, 2H, 3'-H, 5'-H). **$^{13}$C NMR (126 MHz, (CD$_3$)$_2$SO)** $\delta$/ppm = 182.3 (C-3), 166.3 (C-1), 144.6 (C-1''), 142.7 (C-4'), 139.1 (C-1'), 133.5 (C-3'', C-5''), 128.6 (C-4''), 126.1 (C-2'', C-6''), 123.0 (C-2), 121.6 (C-3', C-5'), 120.3 (q, $J_{CF}$ = 255.2 Hz, OCF$_3$), 120.2 (C-2', C-6'), 77.6 (CN). IR (ATR) $\tilde{V}_{max}$/cm$^{-1}$=3304, 2218, 1614, 1536, 1506, 1262, 1208, 1164, 661. **HRMS (ESI):** calcd. for C$_{17}$H$_8$$^{35}$Cl$_2$F$_3$N$_2$O$_3$ (M-H)$^-$ 414.98696; found 414.98676. **Purity (HPLC):**>96% ($\lambda$ = 210 nm),>96% ($\lambda$ = 254 nm).

**SGA-133**

**Chemical structure 57.** Methyl 4-(2-cyano-3-(3,5-dichlorophenyl)−3-hydroxyacrylamido)benzoate – SGA-133.

According to general procedure **B**, amide **13** (218 mg, 1.00 mmol, 1.0 eq.) in dry THF (10 mL), NaH (92.0 mg, 2.30 mmol, 2.3 eq.) and 3,5-dichlorobenzoyl chloride (230 mg, 1.10 mmol, 1.1 eq.) were used. The resulting solid was washed with hexanes, EtOH and water to give **SGA-133** as colorless solid (331 mg, 0.846 mmol, 85%). $R_f$ = 0.15 (3:2 hexanes/acetone). **m.p.:** 234°C. **$^1$H NMR (400 MHz, (CD$_3$)$_2$SO)** $\delta$/ppm = 12.33 (s, 1H, NH), 11.41 (s, 1H, OH), 7.90–7.83 (m, 2H, 2 hr, 6 hr), 7.71–7.66 (m, 2H, 3 hr, 5 hr), 7.64 (t, $J$ = 1.9 Hz, 1H, 4''-H), 7.57 (d, $J$ = 1.9 Hz, 2H, 2''-H, 6''-H), 3.81 (s, 3H, CH$_3$). **$^{13}$C NMR (101 MHz, (CD$_3$)$_2$SO)** $\delta$/ppm = 182.8 (C-3'), 166.3 (C-1'), 166.0 (CO), 145.1 (C-1''), 144.6 (C-4), 133.4 (C-3'', C-5''), 130.4 (C-2, C-6), 128.5 (C-4''), 126.1 (C-2'', C-6''), 123.3 (C-2'), 122.3 (C-1), 118.1 (C-3, C-5), 77.6 (CN), 51.7 (CH$_3$). IR (ATR) $\tilde{V}_{max}$/cm$^{-1}$=3304, 3093, 2215, 1727, 1591, 1534, 1415, 1283, 1262, 1112, 810, 766. **HRMS (ESI):** calcd. for C$_{18}$H$_{11}$$^{35}$Cl$_2$N$_2$O$_4$ (M-H)$^-$ 389.01014; found 389.01077. **Purity (NMR):**>96% ($\lambda$ = 210 nm),>96% ($\lambda$ = 254 nm).

**SGA-137**

**Chemical structure 58.** 4-(2-Cyano-3-(3,5-dichlorophenyl)−3-hydroxyacrylamido)benzoic acid – SGA-137.

Ester **SGA-133** (95.0 mg, 0.243 mmol, 1.0 eq.) was dissolved in dioxane/H$_2$O (3:1; 4.0 mL) and LiOH (61.2 mg, 2.43 mmol, 10 eq.) was added. The mixture was stirred at rt for 3 hr, before 1 M aq. HCl (5.0 mL) was added. The precipitate was filtered off, washed with cold water and dried to give **SGA-137** as colorless solid (59.6 mg, 0.158 mmol, 65%). $R_f$ = 0.00 (3:2 hexanes/acetone). **m.p.:** 244°C. **$^1$H NMR (500 MHz, (CD$_3$)$_2$SO)** $\delta$/ppm = 12.30 (s, 1H, NH), 7.87–7.81 (m, 2H, 2 hr, 6 hr), 7.68–7.64 (m, 2H, 3 hr, 5 hr), 7.63 (t, $J$ = 1.9 Hz, 1H, 4''-H), 7.57 (d, $J$ = 1.9 Hz, 2H, 2''-H, 6''-H). **$^{13}$C NMR (126 MHz, (CD$_3$)$_2$SO)** $\delta$/ppm = 182.7 (C-3'), 167.1 (CO), 166.3 (C-1'), 145.2 (C-1''), 144.2 (C-4), 133.4 (C-3'', C-5''), 130.5 (C-2, C-6), 128.4 (C-4''), 126.1 (C-2'', C-6''), 123.5 (C-1), 123.4 (C-2'), 118.0 (C-3, C-5), 77.5 (CN). IR (ATR) $\tilde{V}_{max}$/cm$^{-1}$=3311, 2215, 1696, 1595, 1550, 1415, 1294, 855, 770. **HRMS (ESI):** calcd. for C$_{17}$H$_9$$^{35}$Cl$_2$N$_2$O$_4$ (M-H)$^-$ 374.99449; found 374.99480. **Purity (HPLC):**>96% ($\lambda$ = 210 nm),>96% ($\lambda$ = 254 nm).

**SGA-27**

**Chemical structure 59.** N-(2-Bromo-4-chlorophenyl)−2-cyano-3-(3,5-dichlorophenyl)−3-hydroxyacrylamide – SGA-27.

According to general procedure **B**, amide **14** (274 mg, 1.00 mmol, 1.0 eq.) in dry THF (10 mL), NaH (92.0 mg, 2.30 mmol, 2.3 eq.) and 3,5-dichlorobenzoyl chloride (230 mg, 1.10 mmol, 1.1 eq.) were used to give **SGA-27** as colorless solid (330 mg, 0.738 mmol, 74%). $R_f$ = 0.25 (3:2 hexanes/acetone). **m.p.:** 203°C. **$^1$H NMR (500 MHz, (CD$_3$)$_2$SO)** $\delta$/ppm = 12.29 (s, 1H, NH), 8.56 (d, $J$ = 9.0 Hz, 1H, 6'-H), 7.68 (d, $J$ = 2.5 Hz, 1H, 3'-H), 7.63 (t, $J$ = 1.9 Hz, 1H, 4''-H), 7.58 (d, $J$ = 1.9 Hz, 2H, 2''-H, 6''-H), 7.36 (dd, $J$ = 9.0, 2.5 Hz, 1H, 5'-H). **$^{13}$C NMR (126 MHz, (CD$_3$)$_2$SO)** $\delta$/ppm = 182.7 (C-3), 166.4 (C-1), 145.2 (C-1''), 137.8 (C-1'), 133.4 (C-3'', C-5''), 131.4 (C-6'), 128.4 (C-4''), 127.8 (C-5'), 126.1 (C-2'', C-6''), 125.4 (C-4'), 123.5 (C-2), 122.1 (C-3'), 112.1 (C-2'), 77.3 (CN). IR (ATR) $\tilde{v}_{max}$/cm$^{-1}$=3366, 3087, 2208, 1556, 1521, 1469, 1360, 1292, 863, 815. **HRMS (ESI):** calcd. for C$_{16}$H$_7$$^{79}$Br$^{35}$Cl$_3$N$_2$O$_2$ (M-H)$^-$ 442.87620; found 442.87736. **Purity (HPLC):**>96% ($\lambda$ = 210 nm),>96% ($\lambda$ = 254 nm).

**SGA-28**

**Chemical structure 60.** 2-Cyano-3-(3,5-dichlorophenyl)-N-(3,4-dimethoxyphenyl)−3-hydroxyacrylamide – SGA-28.

According to general procedure **B**, amide **17** (220 mg, 1.00 mmol, 1.0 eq.) in dry THF (10 mL), NaH (92.0 mg, 2.30 mmol, 2.3 eq.) and 3,5-dichlorobenzoyl chloride (230 mg, 1.10 mmol, 1.1 eq.) were used to give **SGA-28** as yellow solid (200 mg, 0.508 mmol, 51%). $R_f$ = 0.17 (3:2 hexanes/acetone). **m.p.:** 195°C. **$^1$H NMR (500 MHz, (CD$_3$)$_2$SO)** $\delta$/ppm = 11.52 (s, 1H, NH), 7.68 (t, $J$ = 1.8 Hz, 1H, 4''-H), 7.62 (d, $J$ = 1.8 Hz, 2H, 2''-H, 6''-H), 7.30 (d, $J$ = 2.2 Hz, 1H, 2'-H), 7.01 (dd, $J$ = 8.7, 2.2 Hz, 1H, 6'-H), 6.86 (d, $J$ = 8.7 Hz, 1H, 5'-H), 3.73 (s, 3H, OCH$_3$), 3.71 (s, 3H, OCH$_3$). **$^{13}$C NMR (126 MHz, (CD$_3$)$_2$SO)** $\delta$/ppm = 181.7 (C-3), 166.2 (C-1), 148.6 (C-3'), 144.4 (C-4'), 143.6 (C-1''), 133.6 (C-3'', C-5''), 133.0 (C-1'), 128.9 (C-4''), 126.2 (C-2'', C-6''), 122.2 (C-2), 112.3 (C-5'), 111.5 (C-6'), 104.9 (C-2'), 77.9 (CN), 55.8 (OCH$_3$), 55.4 (OCH$_3$). IR (ATR) $\tilde{v}_{max}$/cm$^{-1}$=3284, 2217, 1608, 1549, 1516, 1238, 1028, 810. **HRMS (ESI):** calcd. for C$_{18}$H$_{13}$$^{35}$Cl$_2$N$_2$O$_4$ (M-H)$^-$ 391.02579; found 391.02608. **Purity (HPLC):**>96% ($\lambda$ = 210 nm),>96% ($\lambda$ = 254 nm).

**SGA-39**

**Chemical structure 61.** 2-Cyano-3-(3,5-dichlorophenyl)−3-hydroxy-*N*-(2-iodophenyl)acrylamide – SGA-39.

According to general procedure **B**, amide **15** (286 mg, 1.00 mmol, 1.0 eq.) in dry THF (10 mL), NaH (92.0 mg, 2.30 mmol, 2.3 eq.) and 3,5-dichlorobenzoyl chloride (230 mg, 1.10 mmol, 1.1 eq.) were used to give **SGA-39** as yellow solid (237 mg, 0.515 mmol, 52%). $R_f$ = 0.20 (3:2 hexanes/acetone). **m.p.:** 171°C. $^1$**H NMR (400 MHz, (CD$_3$)$_2$SO)** $\delta$/ppm = 11.87 (s, 1H, NH), 8.27 (dd, *J* = 8.3, 1.5 Hz, 1H, 6′-H), 7.80 (dd, *J* = 7.9, 1.5 Hz, 1H, 3′-H), 7.63 (t, *J* = 1.9 Hz, 1H, 4′′-H), 7.59 (d, *J* = 1.9 Hz, 2H, 2′′-H, 6′′-H), 7.29 (ddd, *J* = 8.4, 7.3, 1.5 Hz, 1H, 5′-H), 6.80–6.67 (m, 1H, 4′-H). $^{13}$**C NMR (126 MHz, (CD$_3$)$_2$SO)** $\delta$/ppm = 182.2 (C-3), 166.4 (C-1), 145.1 (C-1′′), 141.5 (C-1′), 139.0 (C-3′), 133.4 (C-3′′, C-5′′), 128.4 (C-4′′), 128.3 (C-5′), 126.1 (C-2′′, C-6′′), 124.0 (C-4′), 123.6 (C-2), 122.1 (C-6′), 89.2 (C-2′), 77.2 (CN). IR (ATR) $\tilde{V}_{max}$/cm$^{-1}$=3337, 2213, 1579, 1537, 1294, 742. **HRMS (ESI):** calcd. for C$_{16}$H$_8{}^{35}$Cl$_2$IN$_2$O$_2$ (M-H)$^-$ 456.90130; found 456.90095. **Purity (HPLC):**>96% ($\lambda$ = 210 nm),>96% ($\lambda$ = 254 nm).

**SGA-33**

**Chemical structure 62.** N-(3-Chloro-2,4-difluorophenyl)−2-cyano-3-(3,5-dichlorophenyl)−3-hydroxyacrylamide – SGA-33.

According to general procedure **B**, amide **16** (231 mg, 1.00 mmol, 1.0 eq.) in dry THF (10 mL), NaH (92.0 mg, 2.30 mmol, 2.3 eq.) and 3,5-dichlorobenzoyl chloride (230 mg, 1.10 mmol, 1.1 eq.) were used to give **SGA-33** as colorless crystals (290 mg, 0.719 mmol, 72%). $R_f$ = 0.15 (3:2 hexanes/acetone). **m.p.:** 193°C. $^1$**H NMR (400 MHz, (CD$_3$)$_2$SO)** $\delta$/ppm = 12.34 (s, 1H, NH), 8.43 (td, *J* = 9.0, 6.0 Hz, 1H, 6′-H), 7.63 (t, *J* = 1.9 Hz, 1H, 4′′-H), 7.58 (d, *J* = 1.9 Hz, 2H, 2′′-H, 6′′-H), 7.23 (td, *J* = 9.2, 2.1 Hz, 1H, 5′-H). $^{13}$**C NMR (101 MHz, (CD$_3$)$_2$SO)** $\delta$/ppm = 182.9 (C-3), 166.3 (C-1), 152.1 (d, $J_{CF}$ = 242.4 Hz, C-4′), 147.7 (d, $J_{CF}$ = 245.4 Hz, C-2′), 145.0 (C-1′′), 133.4 (C-3′′, C-5′′), 128.5 (C-4′′), 126.3 (dd, $J_{CF}$ = 10.3, 2.6 Hz, C-1′), 126.1 (C-2′′, C-6′′), 123.2 (C-2), 119.1 (dd, $J_{CF}$ = 7.8, 3.0 Hz, C-6′), 111.5 (dd, $J_{CF}$ = 20.7, 3.6 Hz, C-5′), 107.8 (dd, $J_{CF}$ = 22.2, 2.9 Hz, C-3′), 77.3 (CN). IR (ATR) $\tilde{V}_{max}$/cm$^{-1}$=3293, 2206, 1539, 1497, 1370, 1289, 1023, 803. **HRMS (ESI):** calcd. for C$_{16}$H$_6{}^{35}$Cl$_3$F$_2$N$_2$O$_2$ (M-H)$^-$ 400.94684; found 400.94724. **Purity (HPLC):**>96% ($\lambda$ = 210 nm),>96% ($\lambda$ = 254 nm).

**SGA-73**

**Chemical structure 63.** N-(3-Chloro-2,4-difluorophenyl)−2-cyano-3-hydroxy-3-(2,4,6-trichlorophenyl)acrylamide – SGA-73.

According to general procedure **B**, amide **16** (231 mg, 1.00 mmol, 1.0 eq.) in dry THF (10 mL), NaH (92.0 mg, 2.30 mmol, 2.3 eq.) and 2,4,6-trichlorobenzoyl chloride (172 µL, 1.10 mmol, 1.1 eq.) were used to give **SGA-73** as colorless solid (217 mg, 0.496 mmol, 50%). $R_f$ = 0.16 (1:1 hexanes/acetone). **m.p.:** 199°C. $^1$**H NMR (500 MHz, (CD$_3$)$_2$SO)** $\delta$/ppm = 11.93 (s, 1H, NH), 8.45 (td, $J$ = 9.1, 5.9 Hz, 1H, 6'-H), 7.66 (s, 2H, 3''-H, 5''-H), 7.23 (td, $J$ = 9.2, 2.0 Hz, 1H, 5'-H). $^{13}$**C NMR (126 MHz, (CD$_3$)$_2$SO)** $\delta$/ppm = 180.8 (C-3), 165.6 (C-1), 152.1 (d, $J_{CF}$ = 242.6 Hz, C-4'), 147.5 (d, $J_{CF}$ = 245.6 Hz, C-2'), 139.4 (C-1'' or C-4''), 133.1 (C-1'' or C-4''), 132.0 (C-2'', C-6''), 127.8 (C-3'', C-5''), 126.3 (dd, $J_{CF}$ = 10.2, 3.2 Hz, C-1'), 121.8 (C-2), 118.8 (dd, $J_{CF}$ = 7.9, 2.9 Hz, C-6'), 111.5 (dd, $J_{CF}$ = 20.7, 3.5 Hz, C-5'), 107.9 (dd, $J_{CF}$ = 22.0, 19.2 Hz, C-3'), 79.4 (CN). IR (ATR) $\tilde{V}_{max}$/cm$^{-1}$=3279, 2222, 1626, 1590, 1519, 1484, 1445, 1359, 1275, 1017, 816, 631. **HRMS (ESI):** calcd. for C$_{16}$H$_5$$^{35}$Cl$_4$F$_2$N$_2$O$_2$ (M-H)$^-$ 434.90787; found 434.90791. **Purity (HPLC):**>96% ($\lambda$ = 210 nm),>96% ($\lambda$ = 254 nm).

**SGA-90**

**Chemical structure 64.** N-(3-Chloro-2,4-difluorophenyl)−2-cyano-3-(3,5-dibromophenyl)−3-hydroxyacrylamide – SGA-90.

According to general procedure **B**, amide **16** (231 mg, 1.00 mmol, 1.0 eq.) in dry THF (10 mL), NaH (92.0 mg, 2.30 mmol, 2.3 eq.) and 3,5-dibromobenzoic acid (308 mg, 1.10 mmol, 1.1 eq.; converted into the corresponding aryl chloride) were used to give **SGA-90** as light yellow solid (374 mg, 0.759 mmol, 76%). $R_f$ = 0.16 (1:1 hexanes/acetone). **m.p.:** 192°C. $^1$**H NMR (400 MHz, (CD$_3$)$_2$SO)** $\delta$/ppm = 12.33 (s, 1H, NH), 8.43 (td, $J$ = 8.9, 6.0 Hz, 1H, 6'-H), 7.86 (t, $J$ = 1.7 Hz, 1H, 4''-H), 7.74 (d, $J$ = 1.7 Hz, 2H, 2''-H, 6''-H), 7.23 (td, $J$ = 9.3, 2.0 Hz, 1H, 5'-H). $^{13}$**C NMR (101 MHz, (CD$_3$)$_2$SO)** $\delta$/ppm = 182.7 (C-3), 166.3 (C-1), 152.1 (d, $J_{CF}$ = 242.4 Hz, C-2'), 147.7 (d, $J_{CF}$ = 242.4 Hz, C-4'), 145.4 (C-1''), 133.7 (C-4''), 129.2 (C-2'', C-6''), 126.3 (dd, $J_{CF}$ = 10.3, 3.2 Hz, C-1'), 123.2 (C-2), 121.8 (C-3'', C-5''), 119.1 (dd, $J_{CF}$ = 7.5, 2.9 Hz, C-6'), 111.5 (dd, $J_{CF}$ = 20.7, 3.6 Hz, C-5'), 108.3–107.7 (m, C-3'), 77.3 (CN). IR (ATR) $\tilde{V}_{max}$/cm$^{-1}$=3297, 2205, 1586, 1531, 1496, 1022, 803, 750. **HRMS (ESI):** calcd. for C$_{16}$H$_6$$^{79}$Br$_2$ClF$_2$N$_2$O$_2$ (M-H)$^-$ 488.84581; found 488.84607. **Purity (HPLC):**>96% ($\lambda$ = 210 nm),>96% ($\lambda$ = 254 nm).

**SGA-78**

**Chemical structure 65.** 2-Cyano-N-(2,3-dichlorophenyl)─3-(3,5-dichlorophenyl)─3-hydroxyacrylamide – SGA-78.

According to general procedure **B**, amide **18** (229 mg, 1.00 mmol, 1.0 eq.) in dry THF (10 mL), NaH (92.0 mg, 2.30 mmol, 2.3 eq.) and 3,5-dichlorobenzoyl chloride (230 mg, 1.10 mmol, 1.1 eq.) were used to give **SGA-78** as colorless crystals (344 mg, 0.856 mmol, 86%). $R_f$ = 0.17 (3:2 hexanes/acetone). **m.p.:** 211℃. **$^1$H NMR (400 MHz, (CD$_3$)$_2$SO)** $\delta$/ppm = 12.49 (s, 1H, NH), 8.58 (dd, $J$ = 8.3, 1.5 Hz, 1H, 6'-H), 7.63 (t, $J$ = 1.9 Hz, 1H, 4''-H), 7.59 (d, $J$ = 1.9 Hz, 2H, 2''-H, 6''-H), 7.27 (t, $J$ = 8.2 Hz, 1H, 5'-H), 7.20 (dd, $J$ = 8.0, 1.5 Hz, 1H, 4'-H). **$^{13}$C NMR (126 MHz, (CD$_3$)$_2$SO)** $\delta$/ppm = 182.8 (C-3), 166.4 (C-1), 145.1 (C-1''), 139.3 (C-1'), 133.4 (C-3'', C-5''), 131.4 (C-3'), 128.5 (C-4''), 128.0 (C-5'), 126.1 (C-2'', C-6''), 123.4 (C-2), 122.6 (C-4'), 119.3 (C-2'), 119.1 (C-6'), 77.5 (CN). IR (ATR) $\tilde{V}_{max}$/cm$^{-1}$=3355, 2203, 1649, 1584, 1539, 1453, 1415, 872, 812, 777. **HRMS (ESI):** calcd. for C$_{16}$H$_7$$^{35}$Cl$_4$N$_2$O$_2$ (M-H)$^-$ 398.92671; found 398.92789. **Purity (HPLC):**>96% ($\lambda$ = 210 nm),>96% ($\lambda$ = 254 nm).

**SGA-77**

**Chemical structure 66.** 2-Cyano-N-(2,6-dibromophenyl)─3-(3,5-dichlorophenyl)─3-hydroxyacrylamide – SGA-77.

According to general procedure **B**, amide **19** (318 mg, 1.00 mmol, 1.0 eq.) in dry THF (10 mL), NaH (92.0 mg, 2.30 mmol, 2.3 eq.) and 3,5-dichlorobenzoyl chloride (230 mg, 1.10 mmol, 1.1 eq.) were used to give **SGA-77** as colorless crystals (66.0 mg, 0.134 mmol, 13%). $R_f$ = 0.78 (3:2 hexanes/acetone). **m.p.:** 226℃. **$^1$H NMR (400 MHz, (CD$_3$)$_2$SO)** $\delta$/ppm = 10.64 (s, 1H, NH), 8.09–7.90 (m, 3H, 2''-H, 6''-H, 4''-H), 7.79 (d, $J$ = 8.0 Hz, 2H, 3'-H, 5'-H), 7.26 (t, $J$ = 8.0 Hz, 1H, 4'-H). **$^{13}$C NMR (101 MHz, (CD$_3$)$_2$SO)** $\delta$/ppm = 162.4 (C-1, C-3), 136.6 (C-1''), 135.2 (C-1'), 134.6 (C-3'', C-5''), 132.3 (C-3', C-5'), 131.5 (C-4''), 130.7 (C-4'), 126.4 (C-2'', C-6''), 124.4 (C-2', C-6'), 76.4 (CN), C-2 is missing. IR (ATR) $\tilde{V}_{max}$/cm$^{-1}$=3206, 2215, 1650.1567, 1516, 1283, 779, 750, 723. **HRMS (ESI):** calcd. for C$_{16}$H$_7$$^{79}$Br$_2$$^{35}$Cl$_2$N$_2$O$_2$ (M-H)$^-$ 486.82568; found 486.82870. **Purity (HPLC):**>96% ($\lambda$ = 210 nm),>96% ($\lambda$ = 254 nm).

**SGA-85**

**Chemical structure 67.** N-(3,5-Bis(trifluoromethyl)phenyl)─2-cyano-3-(3,5-dichlorophenyl)─3-hydroxyacrylamide –

SGA-85.

According to general procedure **B**, amide **20** (296 mg, 1.00 mmol, 1.0 eq.) in dry THF (10 mL), NaH (92.0 mg, 2.30 mmol, 2.3 eq.) and 3,5-dichlorobenzoyl chloride (230 mg, 1.10 mmol, 1.1 eq.) were used to give **SGA-85** as colorless solid (320 mg, 0.682 mmol, 68%). **R$_f$** = 0.15 (3:2 hexanes/acetone). **m.p.:** 210°C. **$^1$H NMR (400 MHz, (CD$_3$)$_2$SO)** δ/ppm = 12.58 (s, 1H, NH), 8.93 (s, 1H, OH), 8.25 (s, 2H, 2'-H, 6'-H), 7.65 (t, $J$ = 1.9 Hz, 1H, 4''-H), 7.62–7.56 (m, 3H, 4'-H, 2''-H, 6''-H). **$^{13}$C NMR (101 MHz, (CD$_3$)$_2$SO)** δ/ppm = 183.5 (C-3), 167.3 (C-1), 145.4 (C-1''), 142.3 (C-1'), 134.0 (C-3'', C-5''), 131.2 (q, $J_{CF}$ = 32.6 Hz, C-3', C-5'), 129.1 (C-4''), 126.5 (C-2'', C-6''), 123.7 (q, $J_{CF}$ = 270.8 Hz, CF$_3$), 123.4 (C-2), 119.1–118.8 (m, C-2', C-6'), 114.8–114.5 (m, C-4'), 77.6 (CN). **IR (ATR)** $\tilde{V}_{max}$/cm$^{-1}$=2230, 1637, 1571, 1547, 1375, 1275, 1175, 1128, 810. **HRMS (ESI):** calcd. for C$_{18}$H$_7$$^{35}$Cl$_2$F$_6$N$_2$O$_2$ (M-H)$^-$ 466.97943; found 466.97933. **Purity (HPLC):**>96% (λ = 210 nm),>96% (λ = 254 nm).

**Chemical structure 68.** 2-Cyano-3-(3,5-dichlorophenyl)—3-hydroxy-*N*-methyl-*N*-(4-(trifluoromethyl)phenyl)-acrylamide – SGA-115.

According to general procedure **B**, amide **21** (242 mg, 1.00 mmol, 1.0 eq.) in dry THF (10 mL), NaH (92.0 mg, 2.30 mmol, 2.3 eq.) and 3,5-dichlorobenzoyl chloride (230 mg, 1.10 mmol, 1.1 eq.) were used to give **SGA-115** as colorless solid (198 mg, 0.476 mmol, 48%). **R$_f$** = 0.67 (3:2 hexanes/acetone). **m.p.:** 147°C. **$^1$H NMR (400 MHz, CDCl$_3$)** δ/ppm = 7.80–7.73 (m, 2H, 3'-H, 5'-H), 7.64 (d, $J$ = 1.9 Hz, 2H, 2''-H, 6''-H), 7.49 (t, $J$ = 1.9 Hz, 1H, 4''-H), 7.47–7.41 (m, 2H, 2'-H, 6'-H), 3.48 (s, 3H, CH$_3$). **$^{13}$C NMR (101 MHz, CDCl$_3$)** δ/ppm = 184.2 (C-3), 170.4 (C-1), 145.2 (C-1'), 135.8 (C-1''), 135.5 (C-3'', C-5''), 132.5 (C-4''), 131.4 (q, $J_{CF}$ = 32.9 Hz, C-4'), 127.8 (C-2', C-6'), 127.5 (q, $J$ = 3.6 Hz, C-3', C-5'), 127.2 (C-2'', C-6''), 123.7 (q, $J_{CF}$ = 272.6 Hz, CF$_3$), 115.3 (C-2), 78.6 (CN), 39.9 (CH$_3$). **IR (ATR)** $\tilde{V}_{max}$/cm$^{-1}$=3074, 2214, 1578, 1540, 1396, 1331, 1165, 1118, 807. **HRMS (ESI):** calcd. for C$_{18}$H$_{10}$$^{35}$Cl$_2$F$_3$N$_2$O$_2$ (M-H)$^-$ 413.00769; found 413.00806. **Purity (HPLC):**>96% (λ = 210 nm),>96% (λ = 254 nm).

**Chemical structure 69.** 2-Cyano-3-(3,5-dichlorophenyl)—3-hydroxy-*N*-(4-(trifluoromethyl)benzyl)acrylamide – SGA-138.

According to general procedure **B**, amide **22** (242 mg, 1.00 mmol, 1.0 eq.) in dry THF (10 mL), NaH (92.0 mg, 2.30 mmol, 2.3 eq.) and 3,5-dichlorobenzoyl chloride (230 mg, 1.10 mmol, 1.1 eq.) were used to give **SGA-138** as colorless solid (228 mg, 0.549 mmol, 55%). **R$_f$** = 0.70 (3:2 hexanes/acetone). **m.p.:** 173°C. **$^1$H NMR (500 MHz, (CD$_3$)$_2$SO)** δ/ppm = 9.57 (s, 1H, NH), 7.84 (t, $J$ = 1.8 Hz, 1H, 4''-H), 7.77 (d, $J$ = 1.8 Hz, 2H, 2''-H, 6''-H), 7.74–7.69 (m, 2H, 3'-H, 5'-H), 7.57–7.50 (m, 2H, 2'-H, 6'-H), 4.52 (s, 2H, CH$_2$). **$^{13}$C NMR (126 MHz, (CD$_3$)$_2$SO)** δ/ppm = 181.1 (C-3), 169.2 (C-1), 143.5 (C-

1'), 138.5 (C-1''), 134.2 (C-3'', C-5''), 130.8 (C-4''), 128.1 (C-2', C-6'), 127.7 (q, $J_{CF}$ = 31.7 Hz, C-4'), 126.6 (C-2'', C-6''), 125.3 (q, $J_{CF}$ = 3.8 Hz, C-3', C-5'), 124.3 (q, $J_{CF}$ = 271.9 Hz, CF$_3$), 118.1 (C-2), 78.0 (CN), 42.4 (CH$_2$). IR (ATR) $\tilde{V}_{max}$/cm$^{-1}$=3326, 2206, 1538, 1324, 1108, 1067, 805. **HRMS (ESI):** calcd. for C$_{18}$H$_{10}$$^{35}$Cl$_2$F$_3$N$_2$O$_2$ (M-H)$^-$ 413.00769; found 413.00780. **Purity (HPLC):**>96% ($\lambda$ = 210 nm), >96% ($\lambda$ = 254 nm).

## Synthesis of TPC2-A1-P and analogs

**Chemical structure 70.** 1-(5-Bromo-2-(trifluoromethoxy)phenyl)−2-chloroethan-1-one (23).

4-Bromo-2-iodo-1-(trifluoromethoxy)benzene (610 mg, 1.66 mmol, 1.0 eq.) was dissolved in dry THF (8.0 mL) and cooled to −78°C, then n-BuLi (0.670 mL, 1.66 mmol, 1.0 eq.) was added dropwise. The mixture was stirred for 20 min at −78°C and a solution of 2-chloro-N-methoxy-N-methylacetamide (700 mg, 4.99 mmol, 3.0 eq.) in dry THF (8.0 mL) was added slowly. The mixture was stirred for 1 hr at −78°C and then poured on sat. aq. NH$_4$Cl solution. The mixture was extracted with pentane, the organic layer was washed with sat. aq. NaCl solution, dried using hydrophobic phase separation filter papers and filtered through a short silica column (eluent: pentane). The product was carefully concentrated under ambient pressure to yield a colorless oil (**23**, 221 mg, 0.696 mmol, 42%). **R$_f$** = 0.65 (9:1 hexanes/EtOAc). **$^1$H NMR (500 MHz, CDCl$_3$)** $\delta$/ppm = 7.93 (d, J = 2.5 Hz, 1H, 6'-H), 7.71 (dd, J = 8.8, 2.5 Hz, 1H, 4'-H), 7.25–7.22 (m, 1H, 3'-H), 4.61 (s, 2H, 2 hr). **$^{13}$C NMR (126 MHz, CDCl$_3$)** $\delta$/ppm = 190.5 (C-1), 146.2 (C-2'), 137.1 (C-4'), 134.1 (C-6'), 130.8 (C-5'), 122.2 (C-3'), 120.8 (C-1'), 120.3 (q, $J_{CF}$ = 261 Hz, OCF$_3$), 49.0 (C-2). IR (ATR) $\tilde{V}_{max}$/cm$^{-1}$=1703, 1592, 1480, 1398, 1308, 1252, 1174, 1129, 1088, 822, 664. **HRMS (EI):** calcd. for C$_9$H$_5$$^{79}$Br$^{35}$ClF$_3$O$_2$ (M)$^+$ 315.9108; found 315.9106. **Purity (HPLC):**>96% ($\lambda$ = 210 nm),>96% ($\lambda$ = 254 nm).

**Chemical structure 71.** 2-Bromo-1-(2-(trifluoromethoxy)phenyl)ethan-1-one (24).

2'-(Trifluoromethoxy)acetophenone (779 µL, 4.90 mmol, 1.0 eq.) was dissolved in CH$_2$Cl$_2$ (10 mL), then p-toluenesulfonic acid (86.1 mg, 0.490 mmol, 0.10 eq.) and N-bromosuccinimide (872 mg, 4.90 mmol, 1.0 eq.) were added. The mixture was stirred for 24 hr at rt, then sat. aq. NaCl (10 mL) was added. The aqueous phase was extracted with CH$_2$Cl$_2$ (3 × 10 mL), dried using a hydrophobic filter paper and concentrated in vacuo. Purification by FCC (pentane/EtOAc 9:1) yielded bromoketone **24** (672 mg, 2.87 mmol, 49%) as light brown liquid. Analytical data are in accordance with literature (**Saruta et al., 2015**). **R$_f$** = 0.58 (9:1 hexanes/EtOAc). **$^1$H NMR (400 MHz, CDCl$_3$)** $\delta$/ppm = 7.81 (dd, J = 7.8, 1.8 Hz, 1H, 6'-H), 7.60 (ddd, J = 8.3, 7.6, 1.8 Hz, 1H, 4'-H), 7.41 (td, J = 7.6, 1.0 Hz, 1H, 5'-H), 7.37–7.32 (m, 1H, 3'-H), 4.47 (s, 2H, 2 hr). **$^{13}$C NMR (101 MHz, CDCl$_3$)** $\delta$/ppm = 191.5 (C-1), 147.1 (q, J = 1.7 Hz, C-2'), 134.3 (C-4'), 131.6 (C-6'), 129.3 (C-1'), 127.3 (C-5'), 120.6 (C-3'), 120.5 (q, J = 260.1 Hz, OCF$_3$), 35.1 (C-2). IR (ATR) $\tilde{V}_{max}$/cm$^{-1}$=1698, 1603, 1450, 1295, 1248, 1200, 1160. **HRMS (EI):** calcd. for C$_9$H$_6$$^{79}$BrF$_3$O$_2$ (M)$^+$ 281.9498; found 281.9494. **Purity (HPLC):**>96% ($\lambda$ = 210 nm),>96% ($\lambda$ = 254 nm).

**Chemical structure 72.** Ethyl 5-(5-bromo-2-(trifluoromethoxy)phenyl)−1-(cyclohexylmethyl)−2-methyl-1*H*-pyrrole-3-carboxylate. – SGA-140.

Following general procedure **C**, ethyl acetoacetate (88.0 µL, 0.693 mmol, 1.1 eq.) in dry THF (3.0 mL), NaH (37.8 mg, 0.945 mmol, 1.5 eq.) and a solution of ketone **23** (200 mg, 0.630 mmol, 1.0 eq.) and KI (209 mg, 1.26 mmol, 2.0 eq.) in dry THF (3.0 mL) were used. Then the residue was dissolved in acetic acid (6.0 mL) and cyclohexanemethanamine (0.160 mL, 1.26 mmol, 2.0 eq.) was added. FCC (hexanes/EtOAc 99:1) yielded **SGA-140** as colorless oil (140 mg, 0.287 mmol, 46%). $R_f$ = 0.30 (9:1 hexanes/EtOAc). **$^1$H NMR (400 MHz, CDCl$_3$)** δ/ppm = 7.54–7.50 (m, 2H, 4'-H, 6'-H), 7.20 (ddt, *J* = 7.6, 3.0, 1.5 Hz, 1H, 3'-H), 6.55 (s, 1H, 4 hr), 4.27 (q, *J* = 7.1 Hz, 2H, C$\underline{H}_2$CH$_3$), 3.59 (d, *J* = 7.1 Hz, 2H, C$\underline{H}_2$-cy), 2.59 (s, 3H, CH$_3$), 1.60–1.54 (m, 3H, cy), 1.38–1.32 (m, 6H, CH$_2$C$\underline{H}_3$, cy), 1.07–0.99 (m, 3H, cy), 0.68–0.59 (m, 2H, cy). **$^{13}$C NMR (121 MHz, CDCl$_3$)** δ/ppm = 165.6 ($\underline{C}$OOEt), 146.3 (C-2'), 137.6 (C-3), 135.7 (C-6'), 132.4 (C-4'), 129.0 (C-5'), 126.5 (C-5), 122.1 (C-3'), 120.3 (q, $J_{CF}$ = 260.2 Hz, OCF$_3$), 119.9 (C-1'), 112.6 (C-2), 112.2 (C-4), 59.6 ($\underline{C}$H$_2$CH$_3$), 50.8 ($\underline{C}$H$_2$-cy), 39.0 (cy), 30.6 (cy), 26.2 (cy), 25.7 (cy), 14.7 (CH$_2\underline{C}$H$_3$), 12.1 (CH$_3$). IR (ATR) $\bar{V}_{max}$/cm$^{-1}$=2976, 2925, 2854, 1699, 1254, 1240, 1206, 1190, 1169, 1080, 1064, 774. **HRMS (ESI):** calcd. for C$_{22}$H$_{26}$$^{79}$BrF$_3$NO$_3$ (M+H)$^+$ 488.10427; found 488.10459. **Purity (HPLC):**>96% (λ = 210 nm),>96% (λ = 254 nm).

**Chemical structure 73.** 5-(5-Bromo-2-(trifluoromethoxy)phenyl)−1-(cyclohexylmethyl)−2-methyl-1*H*-pyrrole-3-carboxylic acid – TPC2-A1-P.

According to general procedure **D**, LiOH (51.6 mg, 2.05 mmol, 10 eq.) and a solution of **SGA-140** (100 mg, 0.205 mmol, 1.0 eq.) in dioxane/H$_2$O (3.0 mL) were used. After 2 hr the reaction was completed and recrystallization from EtOH gave **TPC2-A1-P** as a colorless solid (51.2 mg, 0.111 mmol, 54%). $R_f$ = 0.14 (9:1 hexanes/EtOAc). **m.p.:** 202°C. **$^1$H NMR (500 MHz, CDCl$_3$)** δ/ppm = 11.34 (s, 1H, COOH), 7.56–7.51 (m, 2H, 3'-H, 4'-H), 7.23–7.19 (m, 1H, 6'-H), 6.61 (s, 1H, 4 hr), 3.60 (d, *J* = 7.1 Hz, 2H, C$\underline{H}_2$-cy), 2.60 (s, 3H, CH$_3$), 1.62–1.56 (m, 3H, cy), 1.40–1.33 (m, 3H, cy), 1.09–1.01 (m, 3H, cy), 0.68–0.61 (m, 2H, cy). **$^{13}$C NMR (126 MHz, CDCl$_3$)** δ/ppm = 170.0 (COOH), 146.4 (C-2'), 138.8 (C-2), 135.7 (C-6'), 132.6 (C-4'), 128.8 (C-5'), 126.8 (C-5), 122.2 (C-3'), 120.3 (q, $J_{CF}$ = 259.3 Hz, OCF$_3$), 119.9 (C-1'), 112.9 (C-4), 111.6 (C-3), 50.9 ($\underline{C}$H$_2$-cy), 39.0 (cy), 30.6 (cy), 26.2 (cy), 25.7 (cy), 12.3 (CH$_3$). IR (ATR) $\bar{V}_{max}$/cm$^{-1}$=2961, 2924, 2875, 2853, 2359, 2342, 1667, 1266, 1243, 1212, 1198, 1171, 925, 779, 658. **HRMS (ESI):** calcd. for C$_{20}$H$_{20}$$^{79}$BrF$_3$NO$_3$ (M-H)$^-$ 458.05841; found 458.05889. **Purity (HPLC):**>96% (λ = 210 nm),>96% (λ = 254 nm).

**SGA-43**

**Chemical structure 74.** Ethyl 1-(cyclohexylmethyl)−2-methyl-5-phenyl-1*H*-pyrrole-3-carboxylate – SGA-43.

Following general procedure **C**, ethyl acetoacetate (42.0 µL, 3.30 mmol, 1.1 eq.) in dry THF (12 mL), NaH (180 mg, 4.50 mmol, 1.5 eq.) and a solution of 2-bromo-1-phenylethan-1-one (405 µL, 3.00 mmol, 1.0 eq.) and KI (996 mg, 6.00 mmol, 2.0 eq.) in dry THF (10 mL) were used. Then the residue was dissolved in acetic acid (10 mL) and cyclohexanemethanamine (781 µL, 6.00 mmol, 2.0 eq.) was added. FCC (hexanes/EtOAc 9:1) yielded **SGA-43** as colorless solid (516 mg, 1.58 mmol, 53%). The compound is literature known, but no analytical data are available (*Kang et al., 2010*). $R_f$ = 0.49 (9:1 hexanes/EtOAc). **m.p.:** 91°C. **$^1$H NMR (400 MHz, CDCl$_3$)** δ/ppm = 7.42–7.36 (m, 2H, Ph), 7.35–7.29 (m, 3H, Ph), 6.53 (s, 1H, 4 hr), 4.27 (q, *J* = 7.1 Hz, 2H, C$\underline{H}_2$CH$_3$), 3.78 (d, *J* = 7.1 Hz, 2H, C$\underline{H}_2$-cy), 2.60 (s, 3H, CH$_3$), 1.58–1.49 (m, 3H, cy), 1.41–1.31 (m, 6H, cy, CH$_2$C$\underline{H}_3$), 1.06–0.95 (m, 3H, cy), 0.69–0.57 (m, 2H, cy). **$^{13}$C NMR (121 MHz, CDCl$_3$)** δ/ppm = 165.9 ($\underline{C}$OOEt), 137.0 (C-2), 134.1 (qPh), 133.7 (C-5), 129.6 (Ph), 128.5 (Ph), 127.4 (Ph), 112.0 (C-3), 110.0 (C-4), 59.4 ($\underline{C}$H$_2$CH$_3$), 50.2 ($\underline{C}$H$_2$-cy), 39.0 (cy), 30.6 (cy), 26.2 (cy), 25.8 (cy), 14.7 (CH$_2\underline{C}$H$_3$), 12.1 (CH$_3$). IR (ATR) $\tilde{V}_{max}$/cm$^{-1}$=2975, 2926, 2850, 1738, 1698, 1420, 1242, 1224, 1191, 1062, 772, 702. **HRMS (ESI):** calcd. for C$_{21}$H$_{28}$NO$_2$ (M+H)$^+$ 326.21146; found 326.21121. **Purity (HPLC):**>96% (λ = 210 nm),>96% (λ = 254 nm).

**SGA-53**

**Chemical structure 75.** 1-(Cyclohexylmethyl)−2-methyl-5-phenyl-1*H*-pyrrole-3-carboxylic acid – SGA-53.

According to general procedure **D**, LiOH (38.7 mg, 1.54 mmol, 10 eq.) and a solution of **SGA-43** (50.0 mg, 0.154 mmol, 1.0 eq.) in dioxane/H$_2$O (1.3 mL) were used. After 1 hr the reaction was completed and gave **SGA-53** as a colorless solid (40.4 mg, 0.136 mmol, 88%). The compound is literature known, but no analytical data are available (*Kang et al., 2010*). $R_f$ = 0.18 (6:1 hexanes/EtOAc). **m.p.:** 196°C. **$^1$H NMR (500 MHz, CDCl$_3$)** δ/ppm = 7.42–7.38 (m, 2H, Ph), 7.36–7.32 (m, 3H, Ph), 6.59 (s, 1H, 4 hr), 3.80 (d, *J* = 7.2 Hz, 2H, CH$_2$), 2.62 (s, 3H, CH$_3$), 1.59–1.52 (m, 3H, cy), 1.42–1.33 (m, 3H, cy), 1.06–0.97 (m, 3H, cy), 0.68–0.59 (m, 2H, cy). **$^{13}$C NMR (101 MHz, CDCl$_3$)** δ/ppm = 170.8 (COOH), 138.3 (C-2), 134.5 (C-5), 133.5 (qPh), 129.7 (Ph), 128.5 (Ph), 127.5 (Ph), 111.2 (C-3), 110.7 (C-4), 50.3 (CH$_2$), 39.0 (cy), 30.6 (cy), 26.2 (cy), 25.8 (cy), 12.3 (CH$_3$). IR (ATR) $\tilde{V}_{max}$/cm$^{-1}$=3030, 2971, 2921, 2848, 1738, 1660, 1533, 1435, 1364, 1267, 1227, 1205, 778, 768, 712, 703. **HRMS (ESI):** calcd. for C$_{19}$H$_{22}$NO$_2$ (M-H)$^-$ 296.16572; found 296.16560. **Purity (HPLC):**>96% (λ = 210 nm),>96% (λ = 254 nm).

**Chemical structure 76.** Ethyl 5-(5-bromo-2-methoxyphenyl)−1-(cyclohexylmethyl)−2-methyl-1*H*-pyrrole-3-carboxylate – SGA-54.

Following general procedure **C**, ethyl acetoacetate (143 µL, 1.10 mmol, 1.1 eq.) in dry THF (5.0 mL), NaH (60.0 mg, 1.50 mmol, 1.5 eq.) and a solution of 2-bromo-1-(5-bromo-2-methoxyphenyl) ethan-1-one (308 mg, 1.00 mmol, 1.0 eq.) in dry THF (1.0 mL) were used. Then the residue was dissolved in acetic acid (5.0 mL) and cyclohexanemethanamine (0.260 mL, 2.00 mmol, 2.0 eq.) was added. FCC (hexanes/EtOAc 9:1) yielded **SGA-54** as colorless solid (411 mg, 0.947 mmol, 95%). $R_f$ = 0.37 (6:1 hexanes/EtOAc). **m.p.:** 83°C. **$^1$H NMR (400 MHz, CDCl$_3$)** $\delta$/ppm = 7.45 (dd, $J$ = 8.7, 2.5 Hz, 1H, 4'-H), 7.37 (d, $J$ = 2.5 Hz, 1H, 6'-H), 6.81 (d, $J$ = 8.7 Hz, 1H, 3'-H), 6.48 (s, 1H, 4 hr), 4.25 (q, $J$ = 7.1 Hz, 2H, C$\underline{H}_2$CH$_3$), 3.76 (s, 3H, OCH$_3$), 3.56 (d, $J$ = 7.2 Hz, 2H, CH$_2$-cy), 2.58 (s, 3H, CH$_3$), 1.60–1.54 (m, 3H, cy), 1.41–1.29 (m, 6H, cy, CH$_2$CH$_3$), 1.08–0.97 (m, 3H, cy), 0.68–0.57 (m, 2H, cy). **$^{13}$C NMR (101 MHz, CDCl$_3$)** $\delta$/ppm = 165.7 (COOEt), 156.6 (C-2'), 136.9 (C-2), 135.2 (C-6'), 132.3 (C-4'), 128.9 (C-5), 124.8 (C-1'), 112.8 (C-5'), 112.6 (C-3'), 112.1 (C-3), 110.6 (C-4), 59.3 (C$\underline{H}_2$CH$_3$), 55.9 (OCH$_3$), 50.9 (CH$_2$-cy), 39.0 (cy), 30.7 (cy), 26.3 (cy), 25.8 (cy), 14.7 (CH$_2$C$\underline{H}_3$), 12.1 (CH$_3$). IR (ATR) $\tilde{V}_{max}$/cm$^{-1}$=2979, 2928, 2849, 1695, 1676, 1473, 1461, 1434, 1253, 1234, 1187, 1176, 1060, 1048, 1027, 774, 619. **HRMS (ESI):** calculated for C$_{22}$H$_{29}$$^{79}$BrNO$_3$ (M+H)$^+$ 434.13253; found 434.13229. **Purity (HPLC):**>96% ($\lambda$ = 210 nm),>96% ($\lambda$ = 254 nm).

**Chemical structure 77.** 5-(5-Bromo-2-methoxyphenyl)−1-(cyclohexylmethyl)−2-methyl-1*H*-pyrrole-3-carboxylic acid – SGA-55.

According to general procedure **D**, LiOH (88.4 mg, 3.51 mmol, 10 eq.) and a solution of **SGA-54** (152 mg, 0.351 mmol, 1.0 eq.) in dioxane/H$_2$O (1.3 mL) were used. After 1 hr the reaction was completed and gave **SGA-55** as a colorless solid (90.0 mg, 0.222 mmol, 63%). $R_f$ = 0.72 (1:1 hexanes/EtOAc). **m.p.:** 224°C. **$^1$H NMR (400 MHz, CDCl$_3$)** $\delta$/ppm = 7.46 (dd, $J$ = 8.8, 2.5 Hz, 1H, 4'-H), 7.38 (d, $J$ = 2.5 Hz, 1H, 6'-H), 6.82 (d, $J$ = 8.8 Hz, 1H, 3'-H), 6.54 (s, 1H, 4 hr), 3.77 (s, 3H, OCH$_3$), 3.57 (d, $J$ = 7.2 Hz, 2H, CH$_2$-cy), 2.59 (s, 3H, CH$_3$), 1.62–1.52 (m, 3H, cy), 1.43–1.33 (m, 3H, cy), 1.10–0.98 (m, 3H, cy), 0.69–0.58 (m, 2H, cy). **$^{13}$C NMR (101 MHz, CDCl$_3$)** $\delta$/ppm = 170.4 (COOH), 156.6 (C-2'), 138.2 (C-2), 135.2 (C-6'), 132.4 (C-4'), 129.3 (C-5), 124.6 (C-1'), 112.8 (C-5'), 112.6 (C-3'), 111.4 (C-4), 111.3 (C-3), 55.9 (OCH$_3$), 51.0 (CH$_2$-cy), 39.0 (cy), 30.7 (cy), 26.3 (cy), 25.8 (cy), 12.3 (CH$_3$). IR (ATR) $\tilde{V}_{max}$/cm$^{-1}$=3027, 2969, 2926, 2850, 1739, 1658, 1476, 1462, 1442, 1362, 1274, 1244, 1205, 1018, 808, 782, 619. **HRMS (ESI):** calculated for C$_{20}$H$_{23}$$^{79}$BrNO$_3$ (M-H)$^-$ 404.08668; found 404.08697. **Purity (HPLC):**>96% ($\lambda$ = 210 nm),>96% ($\lambda$ = 254 nm).

**Chemical structure 78.** Ethyl 1-(cyclohexylmethyl)−5-(2,5-dichlorophenyl)−2-methyl-1*H*-pyrrole-3-carboxylate − SGA-48.

Following general procedure **C**, ethyl acetoacetate (208 μL, 1.65 mmol, 1.1 eq.) in dry THF (5.0 mL), NaH (90.0 mg, 2.25 mmol, 1.5 eq.) and a solution of 2-bromo-1-(2,5-dichlorophenyl)ethan-1-one (402 mg, 1.50 mmol, 1.0 eq.) in dry THF (1.0 mL) were used. Then the residue was dissolved in acetic acid (5.0 mL) and cyclohexanemethanamine (390 μL, 3.00 mmol, 2.0 eq.) was added. FCC (hexanes/EtOAc 9:1), followed by recrystallization from EtOH yielded **SGA-48** as colorless solid (271 mg, 0.686 mmol, 46%). $R_f$ = 0.48 (9:1 hexanes/EtOAc). **m.p.:** 98°C. **$^1$H NMR (400 MHz, CDCl$_3$)** δ/ppm = 7.48 (d, *J* = 2.0 Hz, 1H, 6'-H), 7.29 (dd, *J* = 8.2, 2.0 Hz, 1H, 4'-H), 7.25 (d, *J* = 9.1 Hz, 1H, 3'-H, collapses with chloroform), 6.51 (s, 1H, 4 hr), 4.27 (q, *J* = 7.0 Hz, 2H, CH$_2$CH$_3$), 3.55 (d, *J* = 7.0 Hz, 2H, CH$_2$-cy), 2.59 (s, 3H, CH$_3$), 1.61–1.56 (m, 3H, cy), 1.40–1.31 (m, 6H, cy, CH$_2$CH$_3$), 1.08–0.99 (m, 3H, cy), 0.68–0.58 (m, 2H, cy). **$^{13}$C NMR (101 MHz, CDCl$_3$)** δ/ppm = 165.6 (COOEt), 137.0 (C-2), 135.9 (C-1' or C-5'), 134.8 (C-1' or C-5'), 134.0 (C-3'), 131.2 (C-2'), 129.6 (C-6'), 129.2 (C-5), 127.2 (C-4'), 112.2 (C-3), 111.1 (C-4), 59.5 (CH$_2$CH$_3$), 50.8 (CH$_2$-cy), 39.1 (cy), 30.6 (cy), 26.2 (cy), 25.8 (cy), 14.7 (CH$_2$CH$_3$), 12.1 (CH$_3$). IR (ATR) $\tilde{V}_{max}$/cm$^{-1}$=2981, 2923, 2845, 1739, 1723, 1695, 1565, 1454, 1262, 1238, 1201, 1159, 1076, 1066, 800, 771. **HRMS (ESI):** calculated for C$_{21}$H$_{26}$$^{35}$Cl$_2$NO$_2$ (M+H)$^+$ 394.13351; found 394.13343. **Purity (HPLC):**>96% (λ = 210 nm),>96% (λ = 254 nm).

**Chemical structure 79.** 1-(Cyclohexylmethyl)−5-(2,5-dichlorophenyl)−2-methyl-1*H*-pyrrole-3-carboxylic acid − SGA-52.

According to general procedure **D**, LiOH (63.9 mg, 2.54 mmol, 10 eq.) and a solution of **SGA-48** (100 mg, 0.254 mmol, 1.0 eq.) in dioxane/H$_2$O (1.3 mL) were used. After 1 hr the reaction was completed and gave **SGA-52** as a colorless solid (75.7 mg, 0.207 mmol, 81%). $R_f$ = 0.18 (6:1 hexanes/EtOAc). **m.p.:** 180°C. **$^1$H NMR (500 MHz, CDCl$_3$)** δ/ppm = 7.48 (d, *J* = 2.0 Hz, 1H, 6'-H), 7.30 (dd, *J* = 8.2, 2.0 Hz, 1H, 4'-H), 7.26 (d, *J* = 8.2 Hz, 1H, 3'-H, collapses with chloroform), 6.57 (s, 1H, 4 hr), 3.57 (d, *J* = 6.4 Hz, 2H, CH$_2$-cy), 2.60 (s, 3H, CH$_3$), 1.63–1.57 (m, 3H, cy), 1.42–1.36 (m, 3H, cy), 1.08–1.00 (m, 3H, cy), 0.68–0.59 (m, 2H, cy). **$^{13}$C NMR (126 MHz, CDCl$_3$)** δ/ppm = 170.7 (COOH), 138.3 (C-2), 136.0 (C-1' or C-5'), 135.0 (C-1' or C-5'), 134.0 (C-3'), 131.0 (C-2'), 129.7 (C-6'), 129.6 (C-5), 127.2 (C-4'), 111.8 (C-4), 111.4 (C-3), 50.9 (CH$_2$-cy), 39.0 (cy), 30.7 (cy), 26.2 (cy), 25.8 (cy), 12.3 (CH$_3$). IR (ATR) $\tilde{V}_{max}$/cm$^{-1}$=3014, 2970, 2926, 2851, 1739, 1659, 1449, 1365, 1270, 1228, 1217, 1204, 814, 776. **HRMS (ESI):** calculated for C$_{19}$H$_{20}$$^{35}$Cl$_2$NO$_2$ (M-H)$^-$ 364.08766; found 364.08783. **Purity (HPLC):**>96% (λ = 210 nm),>96% (λ = 254 nm).

**Chemical structure 80.** Ethyl 1-(cyclohexylmethyl)−5-(4-fluorophenyl)−2-methyl-1*H*-pyrrole-3-carboxylate – SGA-59.

Following general procedure **C**, ethyl acetoacetate (208 μL, 1.65 mmol, 1.1 eq.) in dry THF (5.0 mL), NaH (90.0 mg, 2.25 mmol, 1.5 eq.) and a solution of 2-chloro-1-(4-fluorophenyl)ethan-1-one (259 mg, 1.50 mmol, 1.0 eq.) and KI (249 mg, 1.50 mmol, 1.0 eq.) in dry THF (3.0 mL) were used. Then the residue was dissolved in acetic acid (5.0 mL) and cyclohexanemethanamine (0.390 mL, 3.00 mmol, 2.0 eq.) was added. FCC (hexanes/EtOAc 9:1) yielded **SGA-59** as yellow oil (499 mg, 1.45 mmol, 97%). $R_f$ = 0.43 (9:1 hexanes/EtOAc). **$^1$H NMR (500 MHz, CDCl$_3$)** δ/ppm = 7.28–7.23 (m, 2H, 2′-H, 6′-H), 7.10–7.01 (m, 2H, 3′-H, 5′-H), 6.48 (s, 1H, 4 hr), 4.25 (q, *J* = 7.1 Hz, 2H, C*H$_2$*CH$_3$), 3.71 (d, *J* = 7.2 Hz, 2H, C*H$_2$*-cy), 2.57 (s, 3H, CH$_3$), 1.58–1.48 (m, 3H, cy), 1.38–1.28 (m, 6H, cy, CH$_2$C*H$_3$*), 1.06–0.89 (m, 3H, cy), 0.68–0.53 (m, 2H, cy). **$^{13}$C NMR (121 MHz, CDCl$_3$)** δ/ppm = 165.7 (COOH), 162.2 (d, $J_{CF}$ = 247.0 Hz, C-4′), 136.9 (C-2), 132.8 (C-5), 131.3 (d, $J_{CF}$ = 8.0 Hz, C-2′, C-6′), 129.7 (d, $J_{CF}$ = 3.3 Hz, C-1′), 115.4 (d, $J_{CF}$ = 21.3 Hz, C-3′, C-5′), 111.9 (C-3), 110.0 (C-4), 59.3 (*C*H$_2$CH$_3$), 50.1 (*C*H$_2$-cy), 39.0 (cy), 30.5 (cy), 26.1 (cy), 25.7 (cy), 14.6 (CH$_2$*C*H$_3$), 12.0 (CH$_3$). **IR (ATR)** $\tilde{V}_{max}$/cm$^{-1}$=2977, 2925, 2853, 1693, 1242, 1227, 1218, 1195, 1152, 1062, 844, 811, 774. **HRMS (ESI):** calculated for C$_{21}$H$_{27}$FNO$_2$ (M+H)$^+$ 344.20203; found 344.20193. **Purity (HPLC):**>96% (λ = 210 nm), >96% (λ = 254 nm).

**Chemical structure 81.** 1-(Cyclohexylmethyl)−5-(4-fluorophenyl)−2-methyl-1*H*-pyrrole-3-carboxylic acid – SGA-66.

According to general procedure **D**, LiOH (169 mg, 6.71 mmol, 10 eq.) and a solution of **SGA-59** (230 mg, 0.671 mmol, 1.0 eq.) in dioxane/H$_2$O (3.0 mL) were used. After 2 hr the reaction was completed and gave **SGA-66** as a colorless solid (186 mg, 0.590 mmol, 88%). $R_f$ = 0.24 (6:1 hexanes/EtOAc). **m.p.:** 171°C. **$^1$H NMR (400 MHz, CDCl$_3$)** δ/ppm = 7.35–7.27 (m, 2H, 2′-H, 6′-H), 7.14–7.04 (m, 2H, 3′-H, 5′-H), 6.57 (s, 1H, 4 hr), 3.75 (d, *J* = 7.1 Hz, 2H, C*H$_2$*-cy), 2.61 (s, 3H, CH$_3$), 1.63–1.51 (m, 3H, cy), 1.41–1.32 (m, 3H, cy), 1.08–0.97 (m, 3H, cy), 0.70–0.57 (m, 2H, cy). **$^{13}$C NMR (121 MHz, CDCl$_3$)** δ/ppm = 171.2 (COOH), 162.37 (d, $J_{CF}$ = 247.1 Hz, C-4′), 138.3 (C-2), 133.3 (C-5), 131.42 (d, $J_{CF}$ = 8.1 Hz, C-2′, C-6′), 129.6 (d, $J_{CF}$ = 3.4 Hz, C-1′), 115.57 (d, $J_{CF}$ = 21.5 Hz, C-3′, C-5′), 111.2 (C-3), 110.8 (C-4), 50.3 (*C*H$_2$-cy), 39.0 (cy), 30.6 (cy), 26.2 (cy), 25.8 (cy), 12.3 (CH$_3$). **IR (ATR)** $\tilde{V}_{max}$/cm$^{-1}$=2927, 2854, 1739, 1652, 1568, 1494, 1449, 1265, 1223, 1203, 1158, 840, 776, 582. **HRMS (ESI):** calculated for C$_{19}$H$_{21}$FNO$_2$ (M-H)$^-$ 314.15618; found 314.15635. **Purity (HPLC):**>96% (λ = 210 nm),>96% (λ = 254 nm).

**Chemical structure 82.** Ethyl 1-(cyclohexylmethyl)−5-(4-methoxyphenyl)−2-methyl-1*H*-pyrrole-3-carboxylate – SGA-61.

Following general procedure **C**, ethyl acetoacetate (208 µL, 1.65 mmol, 1.1 eq.) in dry THF (5.0 mL), NaH (90.0 mg, 2.25 mmol, 1.5 eq.) and a solution of 2-bromo-1-(4-methoxyphenyl)ethan-1-one (344 mg, 1.50 mmol, 1.0 eq.) and KI (249 mg, 1.50 mmol, 1.0 eq.) in dry THF (3.0 mL) were used. Then the residue was dissolved in acetic acid (5.0 mL) and cyclohexanemethanamine (390 µL, 3.00 mmol, 2.0 eq.) was added. FCC (hexanes/EtOAc 9:1), followed by recrystallization from EtOH yielded **SGA-61** as colorless solid (274 mg, 0.770 mmol, 51%). $R_f$ = 0.37 (9:1 hexanes/EtOAc). **m.p.:** 88°C. **$^1$H NMR (500 MHz, CDCl$_3$)** δ/ppm = 7.25–7.21 (m, 2H, 2'-H, 6'-H), 6.95–6.89 (m, 2H, 3'-H, 5'-H), 6.47 (s, 1H, 4 hr), 4.27 (q, *J* = 7.1 Hz, 2H, C$\underline{H}_2$CH$_3$), 3.84 (s, 3H, OCH$_3$), 3.73 (d, *J* = 7.3 Hz, 2H, C$\underline{H}_2$-cy), 2.58 (s, 3H, CH$_3$), 1.60–1.51 (m, 3H, cy), 1.42–1.31 (m, 6H, cy, CH$_2$C$\underline{H}_3$), 1.07–0.96 (m, 3H, cy), 0.70–0.59 (m, 2H, cy). **$^{13}$C NMR (126 MHz, CDCl$_3$)** δ/ppm = 165.9 (COOH), 159.0 (C-4'), 136.6 (C-2), 133.8 (C-5), 130.9 (C-2', C-6'), 126.1 (C-1'), 113.9 (C-3', C-5'), 111.7 (C-3), 109.5 (C-4), 59.3 ($\underline{C}$H$_2$CH$_3$), 55.4 (OCH$_3$), 50.1 ($\underline{C}$H$_2$-cy), 39.0 (cy), 30.6 (cy), 26.2 (cy), 25.8 (cy), 14.7 (CH$_2$$\underline{C}$H$_3$), 12.1 (CH$_3$). IR (ATR) $\tilde{V}_{max}$/cm$^{-1}$=3016, 2970, 2928, 2847, 1739, 1693, 1568, 1532, 1496, 1443, 1424, 1373, 1243, 1226, 1195, 1175, 1064, 1031, 835, 817, 795, 774. **HRMS (ESI):** calculated for C$_{22}$H$_{30}$NO$_3$ (M+H)$^+$ 356.22202; found 356.22192. **Purity (HPLC):** >96% (λ = 210 nm), >96% (λ = 254 nm).

**Chemical structure 83.** 1-(Cyclohexylmethyl)−5-(4-methoxyphenyl)−2-methyl-1*H*-pyrrole-3-carboxylic acid – SGA-67.

According to general procedure **D**, LiOH (78.7 mg, 3.12 mmol, 10 eq.) and a solution of **SGA-61** (111 mg, 0.312 mmol, 1.0 eq.) in dioxane/H$_2$O (1.3 mL) were used. After 2 hr the reaction was completed and gave **SGA-67** as a colorless solid (95.3 mg, 0.291 mmol, 90%). $R_f$ = 0.20 (6:1 hexanes/EtOAc). **m.p.:** 198°C. **$^1$H NMR (400 MHz, CDCl$_3$)** δ/ppm = 7.27–7.23 (m, 2H, 2'-H, 6'-H), 6.95–6.91 (m, 2H, 3'-H, 5'-H), 6.54 (s, 1H, 4 hr), 3.85 (s, 3H, OCH$_3$), 3.75 (d, *J* = 7.2 Hz, 2H, C$\underline{H}_2$-cy), 2.60 (s, 3H, CH$_3$), 1.62–1.53 (m, 3H, cy), 1.43–1.33 (m, 3H, cy), 1.09–0.98 (m, 3H, cy), 0.70–0.59 (m, 2H, cy). **$^{13}$C NMR (101 MHz, CDCl$_3$)** δ/ppm = 171.3 (COOH), 159.0 (C-4'), 137.8 (C-2), 134.1 (C-5), 131.0 (C-2', C-6'), 125.9 (C-1'), 113.9 (C-3', C-5'), 111.0 (C-3), 110.2 (C-4), 55.3 (OCH$_3$), 50.2 ($\underline{C}$H$_2$-cy), 38.9 (cy), 30.5 (cy), 26.2 (cy), 25.8 (cy), 12.3 (CH$_3$). IR (ATR) $\tilde{V}_{max}$/cm$^{-1}$=3027, 3002, 2970, 2925, 2849, 1738, 1652, 1569, 1535, 1494, 1435, 1364, 1266, 1247, 1228, 1203, 840, 778. **HRMS (ESI):** calculated for C$_{20}$H$_{24}$NO$_3$ (M-H)$^-$ 326.17617; found 326.17633. **Purity (HPLC):** 93% (λ = 210 nm), >96% (λ = 254 nm).

**Chemical structure 84.** Ethyl 1-(cyclohexylmethyl)−5-(2,4-difluorophenyl)−2-methyl-1*H*-pyrrole-3-carboxylate – SGA-62.

Following general procedure **C**, ethyl acetoacetate (208 μL, 1.65 mmol, 1.1 eq.) in dry THF (5.0 mL), NaH (90.0 mg, 2.25 mmol, 1.5 eq.) and a solution of 2-chloro-1-(2,4-difluorophenyl)ethan-1-one (286 mg, 1.50 mmol, 1.0 eq.) and KI (249 mg, 1.50 mmol, 1.0 eq.) in dry THF (3.0 mL) were used. Then the residue was dissolved in acetic acid (5.0 mL) and cyclohexanemethanamine (390 μL, 3.00 mmol, 2.0 eq.) was added. FCC (hexanes/EtOAc 9:1) yielded **SGA-62** as yellow solid (424 mg, 1.17 mmol, 78%). $R_f$ = 0.43 (9:1 hexanes/EtOAc). **m.p.:** 94°C. **$^1$H NMR (500 MHz, CDCl$_3$)** δ/ppm = 7.27 (td, $J$ = 8.4, 6.5 Hz, 1H, 6'-H), 6.96–6.85 (m, 2H, 3'-H, 5'-H), 6.53 (s, 1H, 4 hr), 4.26 (q, $J$ = 7.1 Hz, 2H, C$\underline{H}_2$CH$_3$), 3.60 (d, $J$ = 7.1 Hz, 2H, C$\underline{H}_2$-cy), 2.58 (s, 3H, CH$_3$), 1.59–1.53 (m, 3H, cy), 1.38–1.30 (m, 6H, cy, CH$_2$C$\underline{H}_3$), 1.06–0.98 (m, 3H, cy), 0.68–0.58 (m, 2H, cy). **$^{13}$C NMR (121 MHz, CDCl$_3$)** δ/ppm = 165.5 (COOEt), 162.9 (dd, $J_{CF}$ = 250.1, 11.6 Hz, C-4'), 160.2 (dd, $J_{CF}$ = 248.9, 12.0 Hz, C-2'), 137.2 (C-2), 133.5 (dd, $J_{CF}$ = 9.4, 4.1 Hz, C-6'), 126.2 (C-5), 117.6 (dd, $J_{CF}$ = 15.7, 3.9 Hz, C-1'), 112.3 (C-3), 111.6 (dd, $J_{CF}$ = 21.1, 3.7 Hz, C-5'), 111.3 (C-4), 104.20 (t, $J_{CF}$ = 25.8 Hz, C-3'), 59.4 ($\underline{C}$H$_2$CH$_3$), 50.6 (d, $J_{CF}$ = 3.0 Hz, $\underline{C}$H$_2$-cy), 39.0 (cy), 30.5 (cy), 26.1 (cy), 25.7 (cy), 14.6 (CH$_2\underline{C}$H$_3$), 12.0 (CH$_3$). IR (ATR) $\tilde{V}_{max}$/cm$^{-1}$=2971, 2929, 2848, 1739, 1698, 1571, 1426, 1371, 1235, 1199, 1067, 851, 834. **HRMS (ESI):** calculated for C$_{21}$H$_{26}$F$_2$NO$_2$ (M+H)$^+$ 362.19261; found 362.19247. **Purity (HPLC):**>96% (λ = 210 nm),>96% (λ = 254 nm).

**Chemical structure 85.** 1-(Cyclohexylmethyl)−5-(2,4-difluorophenyl)−2-methyl-1*H*-pyrrole-3-carboxylic acid - SGA-68.

According to general procedure **D**, LiOH (167 mg, 6.61 mmol, 10 eq.) and a solution of **SGA-62** (239 mg, 0.661 mmol, 1.0 eq.) in dioxane/H$_2$O (1.3 mL) were used. After 1 hr the reaction was completed and gave **SGA-68** as a colorless solid (170 mg, 0.511 mmol, 77%). $R_f$ = 0.29 (6:1 hexanes/EtOAc). **m.p.:** 168°C. **$^1$H NMR (400 MHz, CDCl$_3$)** δ/ppm = 7.33–7.25 (m, 1H, 6'-H, collapses with chloroform), 6.98–6.86 (m, 2H, 3'-H, 5'-H), 6.60 (s, 1H, 4 hr), 3.62 (d, $J$ = 7.1 Hz, 2H, C$\underline{H}_2$-cy), 2.61 (s, 3H, CH$_3$), 1.65–1.51 (m, 3H, cy), 1.49–1.32 (m, 3H, cy), 1.14–0.94 (m, 3H, cy), 0.72–0.56 (m, 2H, cy). **$^{13}$C NMR (101 MHz, CDCl$_3$)** δ/ppm = 171.0 (COOH), 163.0 (dd, $J_{CF}$ = 250.3, 11.6 Hz, C-4'), 160.3 (dd, $J_{CF}$ = 249.0, 12.0 Hz, C-2'), 138.5 (C-2), 133.6 (dd, $J_{CF}$ = 9.5, 4.0 Hz, C-6'), 126.7 (C-5), 117.5 (dd, $J_{CF}$ = 15.8, 3.8 Hz, C-1'), 112.0 (C-3), 111.7 (dd, $J_{CF}$ = 21.3, 3.8 Hz, C-5'), 111.6 (C-4), 104.3 (t, $J_{CF}$ = 25.9 Hz, C-3'), 50.72 (d, $J_{CF}$ = 3.0 Hz, $\underline{C}$H$_2$-cy), 39.0 (cy), 30.6 (cy), 26.2 (cy), 25.7 (cy), 12.2 (CH$_3$). IR (ATR) $\tilde{V}_{max}$/cm$^{-1}$=2970, 2926, 2854, 1739, 1666, 1573, 1450, 1433, 1364, 1265, 1239, 1200, 1140, 778. **HRMS (ESI):** calculated for C$_{19}$H$_{20}$F$_2$NO$_2$ (M-H)$^-$ 332.14676; found 332.14697. **Purity (HPLC):**>96% (λ = 210 nm),>96% (λ = 254 nm).

**Chemical structure 86.** Ethyl 1-(cyclohexylmethyl)−2-methyl-5-(2-(trifluoromethoxy)phenyl)−1*H*-pyrrole-3-carboxylate (25).

Following general procedure **C**, ethyl acetoacetate (123 μL, 0.972 mmol, 1.1 eq.) in dry THF (4.0 mL), NaH (53.0 mg, 1.32 mmol, 1.5 eq.) and a solution of ketone **24** (250 mg, 0.883 mmol, 1.0 eq.) and KI (147 mg, 0.883 mmol, 1.0 eq.) in dry THF (2.0 mL) were used. Then the residue was dissolved in acetic acid (5.0 mL) and cyclohexanemethanamine (230 μL, 1.77 mmol, 2.0 eq.) was added. FCC (hexanes/EtOAc 9:1) yielded ester **25** as colorless solid (345 mg, 0.845 mmol, 95%). $R_f$ = 0.46 (9:1 hexanes/EtOAc). **m.p.:** 66°C. **$^1$H NMR (100 MHz, CDCl$_3$)** $\delta$/ppm = 7.44–7.29 (m, 4H, 3'-H, 4'-H, 5'-H, 6'-H), 6.53 (s, 1H, 4 hr), 4.27 (q, *J* = 7.1 Hz, 2H, C$\underline{H}_2$CH$_3$), 3.59 (d, *J* = 7.1 Hz, 2H, C$\underline{H}_2$-cy), 2.59 (s, 3H, CH$_3$), 1.59–1.52 (m, 3H, cy), 1.40–1.32 (m, 6H, cy, CH$_2$C$\underline{H}_3$), 1.06–0.94 (m, 3H, cy), 0.67–0.54 (m, 2H, cy). **$^{13}$C NMR (101 MHz, CDCl$_3$)** $\delta$/ppm = 165.8 (COOEt), 147.4 (C-2'), 137.1 (C-2), 133.4 (C-4', C-5' or C-6'), 129.6 (C-4', C-5' or C-6'), 127.9 (C-5), 126.9 (C-1'), 126.7 (C-4', C-5' or C-6'), 120.5 (q, $J_{CF}$ = 257.4 Hz, OCF$_3$), 120.4 (C-3'), 112.2 (C-3), 111.4 (C-4), 59.5 (C$\underline{H}_2$CH$_3$), 50.7 (C$\underline{H}_2$-cy), 39.0 (cy), 30.6 (cy), 26.2 (cy), 25.8 (cy), 14.7 (CH$_2$C$\underline{H}_3$), 12.1 (CH$_3$). IR (ATR) $\tilde{V}_{max}$/cm$^{-1}$=1934, 1692, 1447, 1422, 1242, 1192, 1155, 1059, 769. **HRMS (ESI):** calculated for C$_{22}$H$_{27}$F$_3$NO$_3$ (M+H)$^+$ 410.19375; found 410.19336. **Purity (HPLC):**>96% ($\lambda$ = 210 nm),>96% ($\lambda$ = 254 nm).

**Chemical structure 87.** 1-(Cyclohexylmethyl)−2-methyl-5-(2-(trifluoromethoxy)phenyl)−1*H*-pyrrole-3-carboxylic acid – SGA-162.

According to general procedure **D**, LiOH (142 mg, 5.62 mmol, 10 eq.) and a solution of ester **25** (230 mg, 0.562 mmol, 1.0 eq.) in dioxane/H$_2$O (3.0 mL) were used. After 2 hr the reaction was completed and gave **SGA-162** as a colorless solid (199 mg, 0.523 mmol, 93%). $R_f$ = 0.11 (9:1 hexanes/EtOAc). **m.p.:** 170°C. **$^1$H NMR (400 MHz, CDCl$_3$)** $\delta$/ppm = 11.13 (s, 1H, COOH), 7.46–7.30 (m, 4H, 3'-H, 4'-H, 5'-H, 6'-H), 6.59 (s, 1H, 4 hr), 3.60 (d, *J* = 7.1 Hz, 2H, C$\underline{H}_2$-cy), 2.61 (s, 3H, CH$_3$), 1.61–1.52 (m, 3H, cy), 1.41–1.31 (m, 3H, cy), 1.09–0.96 (m, 3H, cy), 0.67–0.55 (m, 2H, cy). **$^{13}$C NMR (101 MHz, CDCl$_3$)** $\delta$/ppm = 169.9 (COOH), 147.4 (C-2'), 138.3 (C-2), 133.4 (C-4', C-5' or C-6'), 129.8 (C-4', C-5' or C-6'), 128.3 (C-5), 126.8 (C-4', C-5' or C-6'), 126.7 (C-1'), 120.5 (q, $J_{CF}$ = 260.2 Hz, OCF$_3$), 120.4 (C-4'), 112.1 (C-4), 111.2 (C-3), 50.8 (C$\underline{H}_2$-cy), 38.9 (cy), 30.6 (cy), 26.2 (cy), 25.7 (cy), 12.3 (CH$_3$). IR (ATR) $\tilde{V}_{max}$/cm$^{-1}$=2927, 1642, 1473, 1444, 1249, 1199, 1156, 776, 759. **HRMS (ESI):** calculated for C$_{20}$H$_{21}$F$_3$NO$_3$ (M-H)$^-$ 380.14790; found 380.14805. **Purity (HPLC):**>96% ($\lambda$ = 210 nm).

**Chemical structure 88.** 1-Benzyl-5-(5-bromo-2-(trifluoromethoxy)phenyl)−2-methyl-1*H*-pyrrole-3-carboxylic acid − SGA-150.

Following general procedure **C**, ethyl acetoacetate (104 µL, 0.821 mmol, 1.1 eq.) in dry THF (4.0 mL), NaH (44.8 mg, 1.12 mmol, 1.5 eq.) and a solution of ketone **23** (237 mg, 0.746 mmol, 1.0 eq.) and KI (124 mg, 0.746 mmol, 1.0 eq.) in dry THF (2.0 mL) were used. Then the residue was dissolved in acetic acid (5.0 mL) and benzylamine (204 µL, 1.87 mmol, 2.5 eq.) was added. FCC (hexanes/EtOAc 97:3) yielded ethyl 5-(5-bromo-2-(trifluoromethoxy)phenyl)−1-isopropyl-2-methyl-1*H*-pyrrole-3-carboxylate (**26**) as colorless oil (107 mg, 0.222 mmol). This product was used without further purification or characterization for the next step. $R_f$ = 0.44 (9:1 hexanes/EtOAc). According to general procedure **D**, LiOH (55.9 mg, 2.22 mmol, 10 eq.) and a solution of ester **26** (107 mg, 0.222 mmol, 1.0 eq.) in dioxane/$H_2O$ (3.0 mL) were used. After 16 hr the reaction was completed and gave **SGA-150** as a colorless solid (34.4 mg, 0.0757 mmol, 10% over two steps). $R_f$ = 0.27 (9:1 hexanes/EtOAc). **m.p.:** 185°C. **[1]H NMR (400 MHz, CD$_2$Cl$_2$)** $\delta$/ppm = 11.12 (s, 1H, COOH), 7.51 (dd, *J* = 8.8, 2.5 Hz, 1H, 4'-H), 7.39 (d, *J* = 2.5 Hz, 1H, 6'-H), 7.30–7.18 (m, 4H, 3'-H, Ph), 6.83–6.77 (m, 2H, Ph), 6.70 (s, 1H, 4 hr), 5.00 (s, 2H, CH$_2$), 2.49 (s, 3H, CH$_3$). **[13]C NMR (121 MHz, CD$_2$Cl$_2$)** $\delta$/ppm = 169.9 (COOH), 147.0 (C-2'), 139.6 (C-2), 137.4 (C-1'), 136.3 (C-6'), 133.4 (C-4'), 129.3 (Ph), 128.5 (qPh), 128.0 (Ph), 127.3 (C-5), 126.2 (Ph), 122.8 (C-3'), 120.8 (q, $J_{CF}$ = 258.7 Hz, OCF$_3$), 120.3 (C-5'), 113.1 (C-4), 112.3 (C-3), 48.7 (CH$_2$), 12.2 (CH$_3$). IR (ATR) $\tilde{V}_{max}$/cm$^{-1}$=2925, 2360, 1670, 1249, 1223, 1198, 1171, 733. **HRMS (ESI):** calculated for C$_{20}$H$_{14}$$^{79}$BrF$_3$NO$_3$ (M-H)$^-$ 452.01146; found 452.01168. **Purity (HPLC):** >96% ($\lambda$ = 210 nm),>96% ($\lambda$ = 254 nm).

**Chemical structure 89.** 5-(5-Bromo-2-(trifluoromethoxy)phenyl)−1-isopropyl-2-methyl-1*H*-pyrrole-3-carboxylic acid − SGA-153.

Following general procedure **C**, ethyl acetoacetate (87.6 µL, 0.693 mmol, 1.1 eq.) in dry THF (4.0 mL), NaH (37.8 mg, 0.945 mmol, 1.5 eq.) and a solution of ketone **23** (200 mg, 0.630 mmol, 1.0 eq.) and KI (105 mg, 0.630 mmol, 1.0 eq.) in dry THF (2.0 mL) were used. Then the residue was dissolved in acetic acid (5.0 mL) and isopropylamine (108 µL, 1.26 mmol, 2.0 eq.) was added. FCC (hexanes/EtOAc 99:1) yielded ethyl 5-(5-bromo-2-(trifluoromethoxy)phenyl)−1-isopropyl-2-methyl-1*H*-pyrrole-3-carboxylate (**27**) as colorless oil (65.9 mg, 0.152 mmol). This product was used without further purification or characterization for the next step. $R_f$ = 0.55 (9:1 hexanes/EtOAc). According to general procedure **D**, LiOH (38.3 mg, 1.52 mmol, 10 eq.) and a solution of ester **27** (65.9 mg, 0.152 mmol, 1.0 eq.) in dioxane/$H_2O$ (3.0 mL) were used. After 18 hr the reaction was completed and gave **SGA-153** as a colorless solid (32.4 mg, 0.0798 mmol, 12% over two steps). $R_f$ = 0.11 (9:1 hexanes/EtOAc). **m.p.:** 192°C. **[1]H NMR (400 MHz, C$_3$D$_6$O)** $\delta$/ppm = 7.75 (dd, *J* = 8.8, 2.6 Hz, 1H, 4'-H), 7.65 (d,

$J$ = 2.6 Hz, 1H, 6'-H), 7.44 (dq, $J$ = 8.8, 1.5 Hz, 1H, 3'-H), 6.47 (s, 1H, 4 hr), 4.27 (p, $J$ = 7.0 Hz, 1H, CH), 2.72 (s, 3H, CH$_3$), 1.45 (d, $J$ = 7.0 Hz, 6H, CH(CH$_3$)$_2$). **$^{13}$C NMR (101 MHz, C$_3$D$_6$O)** $\delta$/ ppm = 166.4 (COOH), 147.7 (C-2'), 137.5 (C-2), 137.0 (C-6'), 134.0 (C-4'), 130.5 (C-1'), 126.4 (C-5), 123.7 (q, $J$ = 257.6 Hz, OCF$_3$), 123.5 (C-3'), 120.6 (C-5'), 113.9 (C-3), 113.0 (C-4), 50.2 (CH), 22.0 (CH (CH$_3$)$_2$), 13.0 (CH$_3$). **IR (ATR)** $\tilde{V}_{max}$/cm$^{-1}$=2936, 2358, 1672, 1248, 1214, 1200, 1166. **HRMS (ESI):** calculated for C$_{16}$H$_{14}$$^{79}$BrF$_3$NO$_3$ (M-H)$^-$ 404.01146; found 404.01164. **Purity (HPLC):**>96% ($\lambda$ = 210 nm), >96% ($\lambda$ = 254 nm).

**Chemical structure 90.** 5-(5-Bromo-2-(trifluoromethoxy)phenyl)−2-methyl-1-pentyl-1$H$-pyrrole-3-carboxylic acid − SGA-149.

Following general procedure **C**, ethyl acetoacetate (87.6 µL, 0.693 mmol, 1.1 eq.) in dry THF (4.0 mL), NaH (37.8 mg, 0.945 mmol, 1.5 eq.) and a solution of ketone **23** (200 mg, 0.630 mmol, 1.0 eq.) and KI (105 mg, 0.630 mmol, 1.0 eq.) in dry THF (2.0 mL) were used. Then the residue was dissolved in acetic acid (5.0 mL) and $n$-pentylamine (146 µL, 1.26 mmol, 2.0 eq.) was added. FCC (hexanes/ EtOAc 99:1) yielded ethyl 5-(5-bromo-2-(trifluoromethoxy)phenyl)−2-methyl-1-pentyl-1$H$-pyrrole-3-carboxylate (**28**) as colorless oil (81.5 mg, 0.176 mmol). The product was used without further purifi-cation or characterization for the next step. **R$_f$** = 0.51 (9:1 hexanes/EtOAc). According to general procedure **D**, LiOH (44.4 mg, 1.76 mmol, 10 eq.) and a solution of ester **28** (81.5 mg, 0.176 mmol, 1.0 eq.) in dioxane/H$_2$O (3.0 mL) were used. After 18 hr the reaction was completed and gave **SGA-149** as a colorless solid (40.5 mg, 0.0933 mmol, 15% over two steps). **R$_f$** = 0.06 (9:1 hexanes/EtOAc). **m.p.:** 121°C. **$^1$H NMR (400 MHz, CDCl$_3$)** $\delta$/ppm = 7.57–7.51 (m, 2H, 4''-H, 6''-H), 7.24–7.20 (m, 1H, 3''-H), 6.61 (s, 1H, 4 hr), 3.74–3.68 (m, 2H, 1'-H), 2.61 (s, 3H, CH$_3$), 1.49–1.43 (m, 2H, 2'-H), 1.19–1.06 (m, 4H, 3'-H, 4'-H), 0.80 (t, $J$ = 7.2 Hz, 3H, 5'-H). **$^{13}$C NMR (101 MHz, CDCl$_3$)** $\delta$/ppm = 169.4 (COOH), 146.6 (C-2''), 138.3 (C-2), 135.9 (C-6''), 132.8 (C-4''), 128.7 (C-5''), 126.1 (C-5), 122.5 (q, $J$ = 279.8 Hz, OCF$_3$), 122.4 (C-3''), 120.0 (C-1''), 112.7 (C-4), 111.4 (C-3), 44.7 (C-1'), 30.2 (C-2'), 28.8 (C-3'), 22.1 (C-4'), 13.9 (C-5'), 11.9 (CH$_3$). **IR (ATR)** $\tilde{V}_{max}$/cm$^{-1}$=2929, 1663, 1471, 1436, 1247, 1212, 1194, 1160, 781. **HRMS (ESI):** calculated for C$_{18}$H$_{20}$$^{79}$BrF$_3$NO$_3$ (M+H)$^+$ 434.05732; found 434.05765. **Purity (HPLC):**>96% ($\lambda$ = 210 nm),>96% ($\lambda$ = 254 nm).

**Chemical structure 91.** 5-(5-Bromo-2-(trifluoromethoxy)phenyl)−1-(cyclohexylmethyl)−2-ethyl-1$H$-pyrrole-3-carboxylic acid − SGA-152.

Following general procedure **C**, ethyl propionylacetate (99.9 µL, 0.693 mmol, 1.1 eq.) in dry THF (4.0 mL), NaH (37.8 mg, 0.945 mmol, 1.5 eq.) and a solution of ketone **23** (200 mg, 0.630 mmol, 1.0 eq.) and KI (105 mg, 0.630 mmol, 1.0 eq.) in dry THF (2.0 mL) were used. Then the residue was

dissolved in acetic acid (5.0 mL) and cyclohexanemethanamine (164 µL, 1.26 mmol, 2.0 eq.) was added. FCC (hexanes/EtOAc 99:1) yielded ethyl 5-(5-bromo-2-(trifluoromethoxy)phenyl)−1-(cyclo-hexylmethyl)−2-ethyl-1$H$-pyrrole-3-carboxylate (**29**) as colorless oil (98.0 mg, 0.195 mmol). This product was used without further purification or characterization for the next step. $R_f$ = 0.58 (9:1 hexanes/EtOAc). According to general procedure **D**, LiOH (49.2 mg, 1.95 mmol, 10 eq.) and a solution of ester **29** (98.0 mg, 0.195 mmol, 1.0 eq.) in dioxane/$H_2O$ (3.0 mL) were used. After 18 hr the reaction was completed and gave **SGA-152** as yellow solid (30.8 mg, 0.0649 mmol, 9% over two steps). $R_f$ = 0.24 (9:1 hexanes/EtOAc). **m.p.:** 152°C. **$^1$H NMR (500 MHz, CDCl$_3$)** δ/ppm = 11.35 (s, 1H, COOH), 7.57–7.50 (m, 2H, 4'-H, 6'-H), 7.20 (dd, $J$ = 8.6, 1.4 Hz, 1H, 3'-H), 6.61 (s, 1H, 4 hr), 3.61 (d, $J$ = 7.2 Hz, 2H, CH$_2$-cy), 3.04 (q, $J$ = 7.4 Hz, 2H, CH$_2$CH$_3$), 1.64–1.54 (m, 3H, cy), 1.39–1.31 (m, 3H, cy), 1.23 (t, $J$ = 7.4 Hz, 3H, CH$_2$CH$_3$), 1.08–0.99 (m, 3H, cy), 0.70–0.60 (m, 2H, cy). **$^{13}$C NMR (121 MHz, CDCl$_3$)** δ/ppm = 169.8 (COOH), 146.3 (C-2'), 145.1 (C-2), 135.5 (C-4' or C-6'), 132.5 (C-4' or C-6'), 129.1 (C-1'), 126.7 (C-5), 122.3 (C-3'), 120.3 (q, $J_{CF}$ = 259.2 Hz, OCF$_3$), 120.0 (C-5'), 113.2 (C-4), 110.8 (C-3), 50.9 (CH$_2$-cy), 39.6 (cy), 30.7 (cy), 26.2 (cy), 25.8 (cy), 19.2 (CH$_2$CH$_3$), 14.5 (CH$_2$CH$_3$). IR (ATR) $\tilde{V}_{max}$/cm$^{-1}$=2927, 2359, 1659, 1469, 1441, 1250, 1212, 1192, 1172. **HRMS (ESI):** calculated for C$_{21}$H$_{22}$$^{79}$BrF$_3$NO$_3$ (M-H)$^-$ 472.07406; found 472.07418. **Purity (HPLC):**>96% (λ = 210 nm),>96% (λ = 254 nm).

**Chemical structure 92.** 5-(5-Bromo-2-(trifluoromethoxy)phenyl)−1-(cyclohexylmethyl)−2-phenyl-1$H$-pyrrole-3-carboxylic acid – SGA-154.

Following general procedure **C**, ethyl benzoylacetate (134 µL, 0.693 mmol, 1.1 eq.) in dry THF (4.0 mL), NaH (37.8 mg, 0.945 mmol, 1.5 eq.) and a solution of ketone **23** (200 mg, 0.630 mmol, 1.0 eq.) and KI (105 mg, 0.630 mmol, 1.0 eq.) in dry THF (2.0 mL) were used. Then the residue was dissolved in acetic acid (5.0 mL) and cyclohexanemethanamine (328 µL, 2.52 mmol, 4.0 eq.) was added. FCC (hexanes/EtOAc 99:1) yielded ethyl 5-(5-bromo-2-(trifluoromethoxy)phenyl)−1-(cyclo-hexylmethyl)−2-phenyl-1$H$-pyrrole-3-carboxylate (**30**) as yellow solid (151 mg, 0.274 mmol). This product was used without further purification or characterization for the next step. $R_f$ = 0.50 (9:1 hexanes/EtOAc). According to general procedure **D**, LiOH (69.2 mg, 2.74 mmol, 10 eq.) and a solution of ester **30** (151 mg, 0.274 mmol, 1.0 eq.) in dioxane/$H_2O$ (3.0 mL) were used. After 18 hr the reaction was completed and gave **SGA-154** as yellow solid (91.5 mg, 0.175 mmol, 28% over two steps). $R_f$ = 0.12 (9:1 hexanes/EtOAc). **m.p.:** 170°C. **$^1$H NMR (400 MHz, CDCl$_3$)** δ/ppm = 7.61 (d, $J$ = 2.5 Hz, 1H, 6'-H), 7.55 (dd, $J$ = 8.7, 2.5 Hz, 1H, 4'-H), 7.47–7.41 (m, 3H, Ph), 7.39–7.35 (m, 2H, Ph), 7.23 (dq, $J$ = 8.7, 1.3 Hz, 1H, 3'-H), 6.74 (s, 1H, 4 hr), 3.56 (d, $J$ = 7.4 Hz, 2H, CH$_2$-cy), 1.48–1.42 (m, 3H, cy), 1.11–1.05 (m, 2H, cy), 1.03–0.95 (m, 1H, cy), 0.90–0.82 (m, 3H, cy), 0.43–0.32 (m, 2H, cy). **$^{13}$C NMR (101 MHz, CDCl$_3$)** δ/ppm = 167.7 (COOH), 146.1 (C-2'), 141.7 (C-2), 135.2 (C-6'), 132.7 (C-4'), 131.7 (C-5, C-1', C-5' or qPh), 131.0 (Ph), 128.8 (C-5, C-1', C-5' or qPh), 128.7 (Ph), 128.3 (Ph), 122.4 (C-3'), 120.4 (q, $J_{CF}$ = 266.3 Hz, OCF$_3$), 120.2 (C-5, C-1', C-5' or qPh), 113.8 (C-4), 112.7 (C-3), 52.0 (CH$_2$-cy), 38.7 (cy), 30.3 (cy), 26.1 (cy), 25.6 (cy). One quaternary carbon is missing. IR (ATR) $\tilde{V}_{max}$/cm$^{-1}$=2915, 2335, 1667, 1487, 1248, 1208, 1169, 1127, 796, 697. **HRMS (ESI):** calculated for C$_{25}$H$_{22}$$^{79}$BrF$_3$NO$_3$ (M-H)$^-$ 520.07406; found 520.07418. **Purity (HPLC):**>96% (λ = 210 nm),>96% (λ = 254 nm).

## Statistical analysis

All error bars are depicted as SEM. Statistical significance was determined via Student's t-test, one-way ANOVA, or two-way ANOVA followed by either Tukey's or Bonferroni's post hoc test.

Significance is denoted on figures with asterisks as outlined in the legends. All data presented are representative of three or more independent experiments.

## Acknowledgements

We thank Christopher Wolf and Elisabeth Butz for their assistance with pilot studies, Dr. Youxing Jiang (University of Texas Southwestern Medical Center) for plasmid gifts, and Dr. Herman van der Putten (NCL Foundation) for valuable discussions. This work was supported, in part, by funding from the German Research Foundation, DFG (project number 239283807, SFB/TRR152 projects P04 to CG, P06 to CW-S, P12 to MB, and P14 to SZ; BR 1034/7–1 to FB; GR 4315/2–1 to CG), the University of Pennsylvania Orphan Disease Center and the Mucolipidosis IV Foundation grant MDBR-17–120 ML4 to CG, the Mucolipidosis IV Foundation grant (project number 404.510.2577) to CG, the NCL Foundation grant (project 2018-Grimm-Paquet) to CG, and the BBSRC grant (BB/N01524X/1) to SP.

## Additional information

### Funding

| Funder | Grant reference number | Author |
| --- | --- | --- |
| Mucolipidosis IV Foundation | MDBR-17-120- ML4 | Christian Grimm |
| Deutsche Forschungsgemeinschaft | SFB/TRR152 P04 | Christian Grimm |
| Deutsche Forschungsgemeinschaft | SFB/TRR152 P06 | Christian Wahl-Schott |
| Deutsche Forschungsgemeinschaft | SFB/TRR152 P12 | Martin Biel |
| Deutsche Forschungsgemeinschaft | BR 1034/7-1 | Franz Bracher |
| Biotechnology and Biological Sciences Research Council | BB/N01524X/1 | Sandip Patel |
| Deutsche Forschungsgemeinschaft | 239283807 | Christian Grimm |
| Deutsche Forschungsgemeinschaft | SFB/TRR152 P14 | Christian Grimm |
| Deutsche Forschungsgemeinschaft | GR 4315/2-1 | Christian Grimm |
| University of Pennsylvania | Orphan Disease Center | Christian Grimm |
| Mucolipidosis IV Foundation | 404.510.2577 | Christian Grimm |
| NCL Foundation | 2018-Grimm-Paquet | Christian Grimm |

The funders had no role in study design, data collection and interpretation, or the decision to submit the work for publication.

### Author contributions

Susanne Gerndt, Data curation, Formal analysis, Validation, Visualization; Cheng-Chang Chen, Formal analysis, Validation, Visualization, Methodology; Yu-Kai Chao, Data curation, Formal analysis, Validation, Visualization, Methodology; Yu Yuan, Anna Scotto Rosato, Katharina Jacob, Data curation, Formal analysis, Validation, Investigation, Visualization, Methodology; Sandra Burgstaller, Conceptualization, Data curation, Formal analysis, Validation, Investigation, Visualization, Methodology; Einar Krogsaeter, Data curation, Formal analysis, Investigation, Methodology; Nicole Urban, Ong Nam Phuong Nguyen, Data curation, Formal analysis, Validation, Investigation, Methodology; Meghan T Miller, Data curation, Validation; Marco Keller, Conceptualization, Resources, Validation; Angelika M Vollmar, Resources, Funding acquisition, Project administration; Thomas Gudermann, Resources, Funding acquisition; Susanna Zierler, Conceptualization, Resources; Johann

Schredelseker, Conceptualization; Michael Schaefer, Resources, Methodology; Martin Biel, Resources; Roland Malli, Conceptualization, Resources, Validation, Methodology; Christian Wahl-Schott, Conceptualization, Resources, Methodology; Franz Bracher, Conceptualization, Resources, Data curation, Formal analysis, Supervision, Funding acquisition, Validation, Methodology; Sandip Patel, Conceptualization, Resources, Data curation, Supervision, Funding acquisition, Investigation, Methodology, Writing - original draft, Project administration, Writing - review and editing; Christian Grimm, Conceptualization, Resources, Data curation, Formal analysis, Supervision, Funding acquisition, Validation, Investigation, Visualization, Methodology, Writing - original draft, Project administration, Writing - review and editing

### Author ORCIDs
Cheng-Chang Chen  http://orcid.org/0000-0003-1282-4026
Yu-Kai Chao  http://orcid.org/0000-0002-1202-2448
Einar Krogsaeter  http://orcid.org/0000-0001-8232-5498
Susanna Zierler  https://orcid.org/0000-0002-4684-0385
Johann Schredelseker  http://orcid.org/0000-0002-6657-0466
Roland Malli  http://orcid.org/0000-0001-6327-8729
Christian Grimm  https://orcid.org/0000-0002-0177-5559

### Ethics
Animal experimentation: This study was performed in strict accordance with the recommendations in the Guide for the Care and Use of Laboratory Animals of the Bavarian Government and the European Union. All of the animals were handled according to approved institutional animal care protocols of the University of Munich. The protocol was approved by the Bavarian Government (AZ55.2-1-54-2532-170-17).

### Decision letter and Author response
Decision letter https://doi.org/10.7554/eLife.54712.sa1
Author response https://doi.org/10.7554/eLife.54712.sa2

## Additional files
### Supplementary files
• Transparent reporting form

### Data availability
All data generated or analysed during this study are included in the manuscript and supporting files.

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
