## [Decision Letter]

**Acceptance summary:**

While ion selectivity is generally considered a constant and characteristic feature of ion channels, this paper demonstrates that the ion selectivity of "two-pore" TPC2 Ion channels in endolysosomes is instead a mutable property that depends on the nature of the activating agonist. The authors describe two new agonists that mimic endogenous activators NAADP and PI(3,5)P_2_ and make the channels Na^+^/Ca^2+^-selective or Na^+^-selective, respectively. In addition to resolving conflicting reports of TPC2 ion selectivity, these compounds offer powerful new tools to study the consequences of altered TPC2 selectivity on endolysosome function.

**Decision letter after peer review:**

Thank you for submitting your article "Agonist-mediated switching of ion selectivity in TPC2 differentially promotes lysosomal function" for consideration by *eLife*. Your article has been reviewed by three peer reviewers, and the evaluation has been overseen by a Reviewing Editor and Richard Aldrich as the Senior Editor. The following individuals involved in review of your submission have agreed to reveal their identity: Antony Galione (Reviewer #1); Dejian Ren (Reviewer #2); Youxing Jiang (Reviewer #3).

The reviewers have discussed the reviews with one another and the Reviewing Editor has drafted this decision to help you prepare a revised submission. Please aim to submit the revised version within two months.

Summary:

This paper reports the isolation of novel compounds that activate "two-pore" TPC2 channels present in endolysosomal membranes and their effects on the channels' ion selectivity. The agonists differ markedly in that one (TPC2-A1-N) activates a non-selective (Na^+^- and Ca^2+^-permeable) current and proton flux while the other (TPC2-A1-P) activates a selective Na^+^ current and promotes endosome fusion with the plasma membrane. Interestingly, the two compounds mimic the different effects of the endogenous agonists of the channel (NAADP and PI(3,5)P_2_). Strengths of the paper include the combination of Ca^2+^ imaging and patch-clamp recordings from channels in endolysosomes as well as the plasma membrane. Significant conclusions of the paper include the demonstration that the ion selectivity of a channel can be modulated by different agonists, which challenges the dogma that ion selectivity is a fixed property of the channel protein. In addition, this may help resolve conflicts in the literature regarding the cation selectivity of TPC2, namely that the channels are Na^+^-selective in some studies and Ca^2+^-permeable in others. Finally, the study provides a powerful new tool, the first membrane-permeant agonists that can be used to probe different functions of TPC2 channels in intact cells.

All three reviewers agree that the study is well-designed and extensive and that the data are of high quality and clearly presented. No major new experiments are required, but some straightforward measurements described below are needed to bolster the conclusions regarding the selectivity effects and specificity of the novel agonists.

Essential revisions:

1) The currents produced by TPC2-A1-N and NAADP are quite small, which makes V_rev_ particularly prone to errors from leak current contamination. For example, TPC2-A1-N activates a H^+^ current (Figure 4L). If this current is subtracted from the total current in Figure 2K, the V_rev_ will shift to a more negative value and the calculated *P*_Ca_/*P*_Na_ will be much smaller than that shown in Figure 2Q. A similar argument applies to the NAADP-activated current. This problem could be ameliorated by repeating the TPC2-A1-N and NAADP measurements using larger currents. This may be achievable by recording from L11A/L12A mutant channels in the plasma membrane. If this is not possible, the authors should qualify their estimates of the selectivity for TPC2-A1-N and NAADP accordingly. To strengthen the statistics, it would also be useful to increase the numbers of recordings used to calculate *P*_Ca_/*P*_Na_ (currently only 4 in Figure 2Q). Finally, it would be informative (though not absolutely essential) to show the time course of current activation.

2) That the compounds fail to activate TRPML channels shows a degree of selectivity, but to use these compounds to probe TPC channel function in vivo it is important to know whether the compounds are specific for TPC2 over TPC1. This is especially true given the use of TPC2 in the names of these compounds. This should be tested.

[Editors' note: further revisions were suggested prior to acceptance, as described below.]

Thank you for resubmitting your work entitled "Agonist-mediated switching of ion selectivity in TPC2 differentially promotes lysosomal function" for further consideration by *eLife*. Your revised article has been evaluated by Richard Aldrich as the Senior Editor, and a Reviewing Editor.

The manuscript has been improved but there are some remaining issues that need to be addressed before acceptance, as outlined below:

1) Essential revision #1: The point the reviewers were making is that contamination of the TPC2-A1-N-induced current by H^+^ current may have shifted the reversal potential in the positive direction, which would overestimate the relative *P*_Ca_/*P*_Na_. Since the currents are quite small, the effect of H^+^ current contamination could be significant. You have explained that it is not possible to record larger currents from PM-targeted channels, which would have minimized the contamination effect. That is fine, but there should be some estimate in the paper about the possible size of the error from H^+^ current contamination. This could be done by subtracting the average H^+^ current I/V from the average TPC2-A1-N and NAADP current I/Vs, and recalculating Vrev and *P*_Ca_/*P*_Na_. The question is, how much would it affect the result?

2) The equation that was added to the Materials and methods for calculating *P*_Ca_/*P*_Na_ under non-bi-ionic conditions should have correction factors for Na^+^ and Ca^2+^ activity coefficients (as in the bi-ionic equation). The *P*_Ca_/*P*_Na_ values in Figure 2—figure supplement 1F should be replotted using the corrected values). Please provide a reference for the non-bi-ionic equation. Also, use a consistent term for reversal potential in the two equations (e.g., E_rev_).

3) The new values of *P*_Ca_/*P*_Na_ for TPC2-A1-N under bi-ionic conditions (Figure 2—figure supplement 1F) do not agree with results of the bi-ionic equation. With Vrev=-17 mV, Nai=160 mM, Cao=105 mM, *P*_Ca_/*P*_Na_ is 0.42, not 0.65 as in the figure. Also, the text refers to 140 mM Na^+^ but the Figure 2 shows 160 mM Na^+^. In Figure 2P Vrev for TPC2-A1-N is -12 mV, but -17 mV in Figure 2—figure supplement 1E. Are these different datasets, and if so, why? To avoid confusion, it would be best to consolidate all the bi-ionic measurements and make sure the values in Figure 2P, Q and Figure 2—figure supplement 1E, F are the same. Text should be edited accordingly.

4) The error bar in Figure 2—figure supplement 1F for TPC2-A1-P goes below zero, which is not physically possible. Please check the calculations.

---

## [Author Response]

Essential revisions:1) The currents produced by TPC2-A1-N and NAADP are quite small, which makes Vrev particularly prone to errors from leak current contamination. For example, TPC2-A1-N activates a H^+^ current (Figure 4L). If this current is subtracted from the total current in Figure 2K, the Vrev will shift to a more negative value and the calculated P_Ca_/P_Na_ will be much smaller than that shown in Figure 2Q. A similar argument applies to the NAADP-activated current. This problem could be ameliorated by repeating the TPC2-A1-N and NAADP measurements using larger currents. This may be achievable by recording from L11A/L12A mutant channels in the plasma membrane. If this is not possible, the authors should qualify their estimates of the selectivity for TPC2-A1-N and NAADP accordingly. To strengthen the statistics, it would also be useful to increase the numbers of recordings used to calculate P_Ca_/P_Na_ (currently only 4 in Figure 2Q). Finally, it would be informative (though not absolutely essential) to show the time course of current activation.

We want to clarify that the measurements in Figure 2K cannot be directly compared with the measurements shown in Figure 4L. In Figure 2K endolysosomal patch-clamp experiments were performed using bi-ionic conditions. While the conditions in Figure 4L are not bi-ionic. But we agree with the concern regarding the proton effect. Based on the data that we showed in old Figure 2—figure supplement 1A-D, where we used a pipette solution with mixed Ca/Na, we can conclude that it is indeed the calcium leading to the shift in E_rev_ not protons. See also new Figure 2—figure supplement 1D-F.

**Author response image 1. respfig1:** Agonist-evoked TPC2 cation currents from enlarged endo-lysosomes under different bi-ionic conditions. (**A**) Agonist-evoked cation currents from enlarged endo-lysosomes isolated from HEK293 cells stably expressing human TPC2 using the following conditions: luminal solution containing 114 mM Na^+^ and 30 mM Ca^2+^, pH 4.6; bath solution containing 160 mM Na^+^, pH 7.2 (n = 3, each). (**B**) Expanded view of A. (C-D) Statistical analyses of E_rev_ (**C**) and permeability ratio (P_Ca_/P_Na_) (**D**) using either bi-ionic conditions as shown in Figure 2K-S or conditions as used in A and B.

We appreciate the suggestion to measure NAADP currents in the PM. Unfortunately however, NAADP measurements using the plasma membrane variant of TPC2 consistently failed in our hands, possibly due to lack of an accessory protein necessary for the NAADP effect.

To strengthen the statistics, we have now increased the numbers of recordings used to calculate *P*_Ca_/*P*_Na_ in Figure 2K-Q from n = 4 and 6 to n = 9 and 10.

2) That the compounds fail to activate TRPML channels shows a degree of selectivity, but to use these compounds to probe TPC channel function in vivo it is important to know whether the compounds are specific for TPC2 over TPC1. This is especially true given the use of TPC2 in the names of these compounds. This should be tested.

We absolutely agree with the reviewers and now provide new data using endolysosomal patch-clamp showing that neither TPC2-A1-P nor TPC2-A1-N has an activating effect on TPC1. See new Figure 1—figure supplement 7.

[Editors' note: further revisions were suggested prior to acceptance, as described below.]The manuscript has been improved but there are some remaining issues that need to be addressed before acceptance, as outlined below:1) Essential revision #1: The point the reviewers were making is that contamination of the TPC2-A1-N-induced current by H^+^ current may have shifted the reversal potential in the positive direction, which would overestimate the relative P_Ca_/P_Na_. Since the currents are quite small, the effect of H^+^ current contamination could be significant. You have explained that it is not possible to record larger currents from PM-targeted channels, which would have minimized the contamination effect. That is fine, but there should be some estimate in the paper about the possible size of the error from H^+^ current contamination. This could be done by subtracting the average H^+^ current I/V from the average TPC2-A1-N and NAADP current I/Vs, and recalculating Vrev and P_Ca_/P_Na_. The question is, how much would it affect the result?

We have updated the representative I-V plots of Figure 2K, L and O based on the increased n numbers that are now available in this dataset. Please note, the luminal pH of the bi-ionic measurements was 4.6 while the luminal pH of the H^+^ conductance measurements was 4.4. Hence, there is an almost two-fold difference in proton concentration. The composition of the applied solutions in these two series of experiments are also different: high concentration of Na^+^ and Ca^2+^ were applied for bi-ionic measurements versus Na^+^/Ca^2+^-free in NMDG^+^/H^+^ experiments.

Nevertheless, we now account for the H^+^ current “contamination” by subtracting the average H^+^ current I/V from the average current IVs for TPC2-A1-N and TPC2-A1-P as requested (see Author response image 2). The difference of *P*_Ca_/*P*_Na_ between TPC2-A1-P and TPC2-A1-N is not affected (0.44 for TPC2-A1-N and 0.06 for TPC2-A1-P). To clarify this we have now added the following sentence:

“Proton permeability did not substantially change our estimates of relative Ca^2+^ and Na^+^ permeability as *P*_Ca_/*P*_Na_ values were similar when proton currents were subtracted from currents obtained under bi-ionic conditions (0.44 for TPC2-A1-N and 0.06 for TPC2-A1-P).”

**Author response image 2. respfig2:** Proton permeablity does not substantially change estimates of relative Ca/Na permeabilities. (**A**) I-Vs from Figure 2K/L (Na^+^/Ca^2+^ bi-ionic condition) and Figure 4L/M (NMDG^+^/H^+^ condition). (**B**) Expanded views of I-Vs after subtracting the average H^+^ current from the average Na^+^/Ca^2+^ currents from A. (**C**) Relative cationic permeability ratios (P_Ca_/P_Na_) calculated from data in B.

2) The equation that was added to the Materials and methods for calculating P_Ca_/P_Na_ under non-bi-ionic conditions should have correction factors for Na^+^ and Ca^2+^ activity coefficients (as in the bi-ionic equation). The P_Ca_/P_Na_ values in Figure 2—figure supplement 1F should be replotted using the corrected values). Please provide a reference for the non-bi-ionic equation. Also, use a consistent term for reversal potential in the two equations (e.g., E_rev_).

We corrected the equation as follows:

PCaPNa=γNaγCa∙Nai∙expErevFRT-Nao4Cao∙expErevFRT+1

A reference is cited now: Jackson, 2006.

3) The new values of P_Ca_/P_Na_ for TPC2-A1-N under bi-ionic conditions (Figure 2—figure supplement 1F) do not agree with results of the bi-ionic equation. With V_rev_=-17 mV, Nai=160 mM, Cao=105 mM, P_Ca_/P_Na_ is 0.42, not 0.65 as in the figure.

We have increased the n numbers to 9 and 10, respectively. In the supplementary figure this had not been adapted yet. We have now corrected and updated this figure. 0.65 is correct.

Also, the text refers to 140 mM Na^+^ but the Figure 2 shows 160 mM Na^+^.

We have corrected this in the text. 160 mM Na^+^ is correct.

In Figure 2P Vrev for TPC2-A1-N is -12 mV, but -17 mV in Figure 2—figure supplement 1E. Are these different datasets, and if so, why? To avoid confusion, it would be best to consolidate all the bi-ionic measurements and make sure the values in Figure 2P, Q and Figure 2—figure supplement 1E, F are the same. Text should be edited accordingly.

We have increased the n numbers to 9 and 10, respectively. In the supplementary figure this had not been adapted yet. We have now corrected and updated this figure.

4) The error bar in Figure 2—figure supplement 1F for TPC2-A1-P goes below zero, which is not physically possible. Please check the calculations.

We have recalculated this and corrected accordingly.